# Generation and Evaluation of Synthetic Data Containing Treatments

## Abstract

Causal inference on medical data containing treatments, such as estimation of treatment effects, is crucial to ensure the efficacy and safety of interventions. However, privacy concerns can often limit access to the patient data necessary for such analyses. Generative models can produce synthetic data that preserve privacy and closely approximate the real data distribution, yet existing methods do not consider downstream tasks for data containing treatments, nor the unique challenges these pose. With our work we establish a set of desiderata that synthetic data containing treatments should satisfy to maximise downstream utility: preservation of (i) the covariate distribution, (ii) the treatment assignment mechanism, and (iii) the outcome generation mechanism. Based on these desiderata, we propose a set of evaluation metrics to assess such synthetic data. Finally, we present STEAM: a novel method for generating *Synthetic data for Treatment Effect Analysis in Medicine*. STEAM mimics the data-generating process of data containing treatments and optimises for our desiderata, while allowing differentially private generation. We empirically demonstrate that STEAM achieves state-of-the-art performance across our metrics as compared to existing generative models, particularly as the complexity of the generative task increases.

## 1 Introduction

Medical data sharing is crucial for advancing research in healthcare, enabling the replication of results to establish validity, and the discovery of new insights through alternative analyses (Bauchner et al., 2016; Wirth et al., 2021). However, such sharing is hindered by stringent regulations which restrict access to patient data for research purposes (Annas, 2003; Voigt & Von dem Bussche, 2017).

A potential solution is offered by synthetic data, which has gained increasing recognition in medical literature (Jadon & Kumar, 2023). Generative models can produce synthetic copies of sensitive data, which can be shared more freely (Jordon et al., 2018). If necessary, generation can satisfy formal definitions of privacy, such as differential privacy (DP) (Dwork et al., 2006), ensuring provable guarantees (Pan et al., 2024). Importantly, the promise of synthetic data hinges on its ability to preserve information critical to relevant downstream tasks. Among the existing synthetic data literature (Bauer et al., 2024), most works focus on downstream *predictive* (supervised) tasks, shaping standard evaluation and generation practices to this setting.

However, medical data typically contain *treatment assignment variables*, which invite unique downstream analysis. Data containing treatments are typically analysed via *causal inference* methods (e.g. treatment effect estimation methods) which examine the causal relationships between covariates, treatments, and outcomes, in a manner distinct from associative prediction (Feuerriegel et al., 2024). Despite this, synthetic data papers which use medical data containing treatments for motivation and validation (Choi et al., 2018; Kotelnikov et al., 2022; Yan et al., 2022; Borisov et al., 2023) generally employ standard, prediction-oriented, generation and evaluation techniques (see examples in Appendix A). Failure to acknowledge the likely downstream use of synthetic data containing treatments leads to low-quality generation, which is masked by misaligned evaluation metrics.

**Evaluation.** Standard synthetic data evaluation involves statistical comparison of synthetic and real data (Table 1), and assessing the accuracy of synthetically-trained models in predicting a target variable. In this evaluation paradigm, causal inference tasks are not considered, as treatments are handled like any other feature, limiting the relevance of such assessment for synthetic data containing

treatments. To illustrate this, consider the following key questions that an analyst working with a synthetic dataset containing treatments, $\mathcal{D}_{\text{synth}}$, may ask: *Q1 How representative are the patient covariates in $\mathcal{D}_{synth}$?*; *Q2 How accurate are the treatment assignment decisions in $\mathcal{D}_{synth}$?*; and *Q3 How much error might be introduced in treatment effect estimates derived from $\mathcal{D}_{synth}$?* These questions require differentiation between covariates, treatments, and outcomes, and they cannot be accurately answered with current evaluation protocols.

**Generation.** *Generic synthetic data generation* (Table 2) seeks to minimise the difference in synthetic and real joint distributions, and all variables are generated simultaneously from this distribution. This overlooks the data-generating process (DGP) from which data containing treatments arise, where covariates drive treatment assignments, and both covariates and treatments influence outcomes (Pearl, 2009). In doing so, such generation fails to capitalise on a valuable inductive bias, producing synthetic data which poorly preserve these important relationships for causal inference. Existing *causal generative models*, on the other hand, generally assume access to the full causal graph $\mathcal{G}$, which is overly restrictive in complex settings, such as medicine, where $\mathcal{G}$ is unlikely to be known.

In this work we address these limitations by conducting an analysis of synthetic data containing treatments, proposing novel approaches to evaluation and generation which operate under reasonable assumptions and explicitly consider the likely downstream use of such data. In doing so, we make the following contributions:

1. **Desiderata:** By examining the typical analysis conducted on data containing treatments, we establish a set of desiderata that synthetic data should satisfy in this context (Section 4).

2. **Evaluation:** We show that existing evaluation metrics for synthetic data are inadequate in this setting, as they do not measure how well these desiderata are respected. As a remedy, we propose a principled set of metrics derived from our desiderata, allowing meaningful evaluation of synthetic data containing treatments (Section 5).

3. **Generation:** We propose STEAM, a novel method for synthetic data generation that contains inductive biases to optimise for our desiderata and mimic the real DGP of data containing treatments. Furthermore, STEAM can satisfy DP if desired (Section 6).

4. **Empirical Analysis:** Using our newly established metrics, we demonstrate that STEAM exhibits state-of-the-art performance in generating synthetic data containing treatments, particularly as the real DGP grows in complexity, and in high-dimensional scenarios (Sections 7). Our code is available via `https://anonymous.4open.science/r/STEAM-35EC`.

## 2 PROBLEM FORMULATION

**Setup.** We consider a *data owner* with access to observational or experimental real data $\mathcal{D}_{\text{real}} = \{(\mathbf{X}_{\text{real}}^{(i)}, W_{\text{real}}^{(i)}, Y_{\text{real}}^{(i)})\}_{i=1}^{n}$ sampled from a population $P_{\mathbf{X},W,Y}$, where $\mathbf{X}_{\text{real}}^{(i)} = \{X_j\}_{j=1}^{d} \in \mathcal{X}^{(d)}$ is a vector of $d$ binary or continuous covariates, $W_{\text{real}}^{(i)} \in \{0,1\}$ is a binary treatment assignment, and $Y_{\text{real}}^{(i)} \in \mathcal{Y}$ is a binary or continuous outcome. We refer to the set of all variables in $\mathcal{D}_{\text{real}}$ as $\mathcal{V} = \{X_1, ..., X_d, W, Y\}$. We denote the propensity score with $\pi(\mathbf{x}) = P_{W|\mathbf{X}}(W = 1|\mathbf{X} = \mathbf{x})$.

**Objective.** We wish to enable the release of synthetic data to downstream users with various analysis goals, such as estimation of propensity scores, average treatment effects (ATEs), and conditional average treatment effects (CATEs).[1] To do so, we aim to generate synthetic data $\mathcal{D}_{\text{synth}} = \{(\mathbf{X}_{\text{synth}}^{(i)}, W_{\text{synth}}^{(i)}, Y_{\text{synth}}^{(i)})\}_{i=1}^{n}$ from a distribution $Q$ and evaluate how well $\mathcal{D}_{\text{synth}}$ captures information relevant to likely downstream tasks with a set of metrics $\mathcal{M}(\mathcal{D}_{\text{real}}, \mathcal{D}_{\text{synth}})$.

**Terminology.** To avoid confusion, we clarify that 'real data' refers to the specific data of the data owner (rather than simply any real-world observational data). Further, 'synthetic data' refers strictly to data that serve as synthetic copies of real data. This should not be confused with *simulated* or *semi-simulated* data, which are often used to benchmark CATE learners (Curth et al., 2021).

## 3 RELATED WORK

**Evaluation.** Evaluation of synthetic tabular data, the modality we focus on in this paper, is diverse, although there are two common themes: *resemblance* and *predictive utility* (Murtaza et al., 2023).

---

[1] Denoting the potential outcomes as Y(0) and Y(1), ATE is defined as ATE $= \mathbb{E}_P[Y(1) - Y(0)]$ and CATE is $\tau(\mathbf{x}) = \mathbb{E}_P[Y(1) - Y(0)|\mathbf{X} = \mathbf{x}]$.

Table 1: Tabular synthetic data evaluation methods applied to data containing treatments. $d(\cdot)$ is an abstract distance function. For 'Prec., Rec.', we use $\mathcal{S}_P$ to denote the support of distribution $P$. For 'Discriminator', $\mathcal{D}_{\text{comb}} = \mathcal{D}_{\text{real}} \cup \mathcal{D}_{\text{synth}}$, and $c^{(i)}$ is the dataset label for instance $i$. '**Q. addressed?**': which, if any, of the key questions from Section 1 does the method answer?

| | Method | Formula | Differentiates between $\mathbf{X}, W, Y$? | Q. addressed? |
|---|---|---|---|---|
| *Existing* | Marginal | $\frac{1}{|\mathcal{V}|}\sum_{i\in\mathcal{V}} d(P_i, Q_i)$ | ✗ | - |
| | Correlation | $\frac{1}{|\mathcal{V}|(|\mathcal{V}|-1)}\sum_{\substack{i,j\in\mathcal{V}\\i\neq j}} d(\text{corr}(P_i, P_j), \text{corr}(Q_i, Q_j))$ | ✗ | - |
| | Joint | $d(P_{\mathcal{V}}, Q_{\mathcal{V}})$ | ✗ | - |
| | Prec., Rec. | $|\mathcal{S}_P \cap \mathcal{S}_Q|/|\mathcal{S}_Q|, |\mathcal{S}_P \cap \mathcal{S}_Q|/|\mathcal{S}_P|$ | ✗ | - |
| | Discriminator | $\frac{1}{2n}\sum_{i=1}^{2n} \mathbb{1}\{C(\mathcal{D}_{\text{comb}}^{(i)}) = c^{(i)}\}$ | ✗ | - |
| *Ours* | $P_{\alpha,\mathbf{X}}, R_{\beta,\mathbf{X}}$ | Equations 3 and 4 | ✓ | *Q1* |
| | $\text{JSD}_\pi$ | Equation 5 | ✓ | *Q2* |
| | $U_{\text{PEHE}}$ | Equation 6 | ✓ | *Q3* |

Assessing *predictive utility* involves training a predictive model on $\mathcal{D}_{\text{synth}}$ and measuring its accuracy. Since key causal inference tasks, such as treatment effect estimation, do not have observable ground-truths (Holland, 1986), model validation becomes a non-trivial task (Curth & Van Der Schaar, 2023) and such utility assessment does not apply to our setting. On the other hand, methods for assessing *resemblance*, which can involve comparison of synthetic and real marginals, 2-way correlation matrices, joint distributions, supports via precision and recall, and assessment of a discriminator model $C$ in separating real and synthetic data, are summarised in Table 1. Importantly, these methods handle all features within a dataset similarly, not differentiating between the variable classes $\mathbf{X}$, $W$, and $Y$, and they therefore cannot directly answer any of the key questions *Q1-3* posed in Section 1. Our metrics, proposed in Section 5, remedy this.

**Generation.** We consider two related approaches to generative modeling, with a high-level comparison in Table 2. Firstly, *generic generative models* are those which make minimal assumptions on $\mathcal{D}_{\text{real}}$, and seek to minimise the difference between real and synthetic joint distributions. *Causal generative models*, on the other hand, assume access to the full causal graph $\mathcal{G}$, and then seek to minimize the difference between each real and synthetic conditional distribution, as dictated by the causal relationships in $\mathcal{G}$. The assumptions for our proposed method, STEAM, fall in between these two. While we assume that the underlying DGP of $\mathcal{D}_{\text{real}}$ is of the form $X \sim P_{\mathbf{X}}, W \sim P_{W|\mathbf{X}}, Y \sim P_{Y|W,\mathbf{X}}$, this will hold for a wide array of datasets containing treatments, and it is generally less restrictive than assuming complete knowledge of $\mathcal{G}$, as we do not need the individual causal relationships between variables. Therefore, STEAM is more applicable in complex settings. Furthermore, neither of these existing approaches are tailored to downstream treatment effect estimation, and they do not target our desiderata for synthetic data containing treatments (Section 4). For empirical comparisons against generic generative models, see Section 7, and see Appendix O for comparison with causal generative models. For further elaboration on specific evaluation and generation methods see Appendix B.

## 4 DESIDERATA FOR SYNTHETIC DATA CONTAINING TREATMENTS

We now consider three distributions that are critical in downstream analysis of data containing treatments: (i) the covariate distribution $P_{\mathbf{X}}$, (ii) the treatment assignment mechanism $P_{W|\mathbf{X}}$, and (iii) the outcome generation mechanism $P_{Y|W,\mathbf{X}}$. Their importance, which we describe below, is clear in the causal inference community, giving rise to the key questions *Q1-3* which downstream analysts may ask when using $\mathcal{D}_{\text{synth}}$. However, this has not been addressed by the synthetic data community, and methods designed to specifically preserve them are missing. To bridge this gap, we establish our desiderata for synthetic data containing treatments based on these distributions.

> **(i) The covariate distribution $P_{\mathbf{X}}$**
>
> $P_{\mathbf{X}}$ describes the population of interest and, in medical practice, it is standard to report its characteristics (Wolff et al., 2019), as it determines to whom analysis will be relevant.
>
> *Why is its preservation important?* Failure to cover covariate levels in $\mathcal{D}_{\text{synth}}$ can result in exclusion from downstream analysis of members of the population whose covariates are not well

Table 2: Tabular synthetic data generation methods. $d(\cdot)$ is an abstract distance function. $\text{PA}_{\mathcal{G}}(\mathcal{V}_i)$ refers to the set of parents of node $\mathcal{V}_i$ in the causal graph $\mathcal{G}$. [1]: Rezende & Mohamed (2016), [2]: Xu et al. (2019), [3]: Kotelnikov et al. (2022), [4]: Watson et al. (2023), [5]: ANM (Hoyer et al., 2008), [6]: Sánchez-Martin et al. (2022), [7]: Chao et al. (2024)

| | Methods | Distributional target | Assumptions on $\mathcal{D}_{\text{real}}$ | $\mathcal{D}_{\text{synth}}$ application |
|---|---|---|---|---|
| *Generic gen. models* | NFlow [1] CTGAN [2] TVAE [2] TabDDPM [3] ARF [4] | $\min d(Q_{\mathcal{V}}, P_{\mathcal{V}})$ | None | Prediction |
| *Causal gen. models* | ANM [5] VACA [6] CGM [7] | $\min d\big(Q_{\mathcal{V}_1\mid\text{PA}_{\mathcal{G}}(\mathcal{V}_1)}, P_{\mathcal{V}_1\mid\text{PA}_{\mathcal{G}}(\mathcal{V}_1)}\big)$ $\vdots$ $\min d\big(Q_{\mathcal{V}_{\lvert\mathcal{V}\rvert}\mid\text{PA}_{\mathcal{G}}(\mathcal{V}_{\lvert\mathcal{V}\rvert})}, P_{\mathcal{V}_{\lvert\mathcal{V}\rvert}\mid\text{PA}_{\mathcal{G}}(\mathcal{V}_{\lvert\mathcal{V}\rvert})}\big)$ | Known $\mathcal{G}$ | Interventional and counterfactual queries on $\mathcal{G}$ |
| *Ours* | STEAM | $\min d(Q_{\mathbf{X}}, P_{\mathbf{X}})$ $\min d(Q_{W\mid\mathbf{X}}, P_{W\mid\mathbf{X}})$ $\min d(Q_{Y\mid W,\mathbf{X}}, P_{Y\mid W,\mathbf{X}})$ | Valid DGP | Treatment effect estimation |

explored, as making reliable inferences can become infeasible (Petersen et al., 2010; Rudolph et al., 2022). On the other hand, generating out-of-distribution covariates in $\mathcal{D}_{\text{synth}}$ can cause groundless extrapolation by synthetically-trained models, leading to potential misuse.

**(ii) The treatment assignment mechanism $P_{W\mid\mathbf{X}}$**

$P_{W\mid\mathbf{X}}$ is often used as a nuisance parameter in treatment effect models (Austin, 2011; Curth & van der Schaar, 2021), and it can be a target for analysis itself when examining treatment protocols.

*Why is its preservation important?* Given the use of $P_{W\mid\mathbf{X}}$ as a nuisance parameter, errors in its modelling will propagate to errors in treatment effect estimates derived from $\mathcal{D}_{\text{synth}}$. Furthermore, $P_{W\mid\mathbf{X}}$ can guide the difficult task of CATE model selection (Hüyük et al., 2024), so poor preservation may lead to inconsistency in this area between $\mathcal{D}_{\text{real}}$ and $\mathcal{D}_{\text{synth}}$, which is unideal (Hansen et al., 2023). Finally, misrepresenting $P_{W\mid\mathbf{X}}$ can lead to misreporting of treatment protocols. Given that extreme propensities of $\pi(\mathbf{x}) \approx 0$ (or $\pi(\mathbf{x}) \approx 1$) are common in high-dimensional data, such as electronic health records (Li et al., 2018), an inaccurate $Q_{W\mid\mathbf{X}}$ could lead to subsequent exploration of treatments in patient subgroups for which they are unsafe.

**(iii) The outcome generation mechanism $P_{Y\mid W,\mathbf{X}}$**

$P_{Y\mid W,\mathbf{X}}$ is the distribution through which treatment effects can be estimated by comparing the statistical functionals of $P_{Y\mid W=1,\mathbf{X}}$ and $P_{Y\mid W=0,\mathbf{X}}$.

*Why is its preservation important?* $P_{Y\mid W,\mathbf{X}}$ must be preserved, so that $\mathcal{D}_{\text{synth}}$ can permit accurate estimation of treatment effects. If $Q_{Y\mid W,\mathbf{X}}$ is inaccurate, then even a perfect model could not estimate correct treatment effects from $\mathcal{D}_{\text{synth}}$, and the worse this relationship is preserved, the less useful it becomes.

Preserving (i)-(iii) is *necessary* and *sufficient* for $Q$ to be a high quality approximation of $P$. Modelling each distribution well is evidently *necessary* given the above reasons, and it is also *sufficient*, which is clear from the following decomposition of $P_{\mathbf{X},W,Y}$:

$$P_{\mathbf{X},W,Y}(\mathbf{X}, W, Y) = \underbrace{P_{\mathbf{X}}(\mathbf{X})}_{(i)} \underbrace{P_{W\mid\mathbf{X}}(W\mid\mathbf{X})}_{(ii)} \underbrace{P_{Y\mid W,\mathbf{X}}(Y\mid W,\mathbf{X})}_{(iii)} \tag{1}$$

The components (i)-(iii) offer a complete factorisation of the joint distribution, and therefore $Q$ matching $P$ in each component is sufficient for $Q$ to match $P$ entirely. As such, accurate modelling of (i)-(iii) forms our desiderata for synthetic data containing treatments. Generation methods should seek to *maximise adherence to these desiderata*, and evaluation metrics should *assess how successful $\mathcal{D}_{synth}$ is in this regard* (and therefore answer **Q1-3**). In the following sections, we show that existing metrics (Section 5.1), and generation methods (Section 7) perform poorly in this regard.

**On causal assumptions.** Even if these desiderata are satisfied, $\mathcal{D}_{\text{synth}}$ may not permit correct causal inference. Required assumptions, such as typical identifiability assumptions,[2] must still be critically examined, since any violations in $\mathcal{D}_{\text{real}}$ will almost surely be violated in a faithful $\mathcal{D}_{\text{synth}}$ as well. Identifying and accounting for such violated assumptions is a task orthogonal to synthetic data generation, with existing literature (Kallus et al., 2019; Frauen & Feuerriegel, 2022), and we do not consider it necessary for $Q$ to improve upon such factors. Instead, any biases in $P$ should be maintained in $Q$, allowing post-generation methods to rectify them if necessary.

## 5 How to evaluate synthetic data containing treatments

With our desiderata established, we now investigate how to evaluate the adherence of $\mathcal{D}_{\text{synth}}$.

### 5.1 Inadequacy of existing metrics

Existing evaluation metrics, discussed in Section 3, do not offer a clear sense of how well $\mathcal{D}_{\text{synth}}$ satisfies our desiderata. These metrics do not differentiate between $\mathbf{X}$, $W$, and $Y$, and they therefore cannot directly assess any of $Q_{\mathbf{X}}$, $Q_{W|\mathbf{X}}$, or $Q_{Y|W,\mathbf{X}}$. Within these existing metrics, joint-distribution-level metrics, such as Kullback–Leibler divergence (KL) (Kullback & Leibler, 1951), are most popular, since they offer a complete, holistic assessment of how well $Q$ models $P$. However these are, at best, loosely related to our desiderata, and they do not allow a user to disentangle how each of (i)-(iii) is preserved, limiting the depth of information offered on $\mathcal{D}_{\text{synth}}$. Furthermore, we argue that these metrics will tend to be dominated by the covariate distribution as $\mathbf{X}$ grows in dimensionality, and they will lose *sensitivity* to the treatment assignment and outcome generation mechanisms. In this sense, *sensitivity* refers to the effect that differences in the modelling of $P_{W|\mathbf{X}}$ or $P_{Y|W,\mathbf{X}}$ by a proposal distribution $Q$ have on a metric $\mathcal{M}$.

To demonstrate this more formally, consider a simple $P_{\mathbf{X},W,Y}$ which can be factorized as $P_{\mathbf{X},W,Y} = \prod_{i=1}^{d} P_{\mathbf{X}_i} \, P_{W|\mathbf{X}} \, P_{Y|W,\mathbf{X}}$. Let there be two learnable distributions $Q_{\mathbf{X},W,Y}^{\theta_1}$ and $Q_{\mathbf{X},W,Y}^{\theta_2}$, which estimate $P_{\mathbf{X},W,Y}$, with the same form $Q_{\mathbf{X},W,Y}^{\theta_k} = \prod_{i=1}^{d} Q_{\mathbf{X}_i}^{\theta_{\mathbf{X}}} \, Q_{W|\mathbf{X}}^{\theta_{W,k}} \, Q_{Y|W,\mathbf{X}}^{\theta_{Y,k}}$, and which only differ in either $\theta_{W,k}$ or $\theta_{Y,k}$ (i.e. they either model $P_{W|\mathbf{X}}$ or $P_{Y|W,\mathbf{X}}$ differently). In this setting, the following holds:

**Theorem 1.** *Let $P$, $Q^{\theta_1}$, $Q^{\theta_2}$ be of the above form, and $\mathcal{M}$ be KL divergence. If we assume that $Q^{\theta_1}$ and $Q^{\theta_2}$ have sufficient capacity to have bounded error on each component, i.e. $\forall i$, $0 < \mathcal{M}(P_{\mathbf{X}_i}, Q_{\mathbf{X}_i}^{\theta_{\mathbf{X}}}) < \varepsilon_{\mathbf{X}}$, and $0 < \mathcal{M}(P_{W|\mathbf{X}}, Q_{W|\mathbf{X}}^{\theta_{W,k}}) < \varepsilon_{W,k}$, and $0 < \mathcal{M}(P_{Y|W,\mathbf{X}}, Q_{Y|W,\mathbf{X}}^{\theta_{Y,k}}) < \varepsilon_{Y,k}$, then:*

$$\frac{\mathcal{M}(P_{\mathbf{X},W,Y}, Q_{\mathbf{X},W,Y}^{\theta_1})}{\mathcal{M}(P_{\mathbf{X},W,Y}, Q_{\mathbf{X},W,Y}^{\theta_2})} \to 1, \text{ as } d \to \infty \tag{2}$$

*Proof.* See Appendix B. $\qquad\square$

Theorem 1 shows that KL divergence loses sensitivity to $W|\mathbf{X}$ and $Y|W,\mathbf{X}$ as $d$ grows, suggesting that this metric will struggle in selecting between $Q_{\mathbf{X},W,Y}^{\theta_1}$ and $Q_{\mathbf{X},W,Y}^{\theta_2}$, since their scores will converge to the same value despite any difference in their modelling of $P_{W|\mathbf{X}}$ or $P_{Y|W,\mathbf{X}}$. For an empirical example of this phenomena, with an extended array of joint-distribution-level metrics, see Appendix D.

### 5.2 Metrics tailored to synthetic data containing treatments

These formal and empirical findings motivate us to design our own metrics for synthetic data containing treatments. We now propose an appropriate set of metrics $\mathcal{M} = (P_{\alpha,\mathbf{X}}, R_{\beta,\mathbf{X}}, JSD_{\pi}, U_{\text{PEHE}})$ which directly measure performance in line with desiderata (i)-(iii), and can offer answers to *Q1-3*.

#### 5.2.1 The covariate distribution $P_{\mathbf{X}}$

Evaluation of the preservation of $P_{\mathbf{X}}$ requires direct comparison of the generally high-dimensional covariate distributions of $\mathcal{D}_{\text{real}}$ and $\mathcal{D}_{\text{synth}}$, which is non-trivial. Nevertheless this is a standard synthetic data evaluation task, as $\mathbf{X}_{\text{real}}$ and $\mathbf{X}_{\text{synth}}$ do not contain treatments. We see precision/recall analysis as the most useful evaluation practice in this context. There is typically a trade-off between these two qualities, which generative models approach differently (Sajjadi et al., 2018; Bayat, 2023),

---

[2]Consistency: $Y^{(i)} = Y(W^{(i)})$, overlap: $0 < \pi(\mathbf{x}) < 1$, and unconfoundedness: $Y(0), Y(1) \perp\!\!\!\perp W|\mathbf{X}$

and by measuring them both a data holder can guide generation towards their preferences of covariate realism and diversity. If the data holder has no strong preference, balancing the two is recommended to achieve the best downstream results (Jordon et al., 2022).

We propose the use of the integrated $P_\alpha$ and $R_\beta$ scores, introduced by Alaa et al. (2022). Intuitively, $P_\alpha$ captures how much of the synthetic data falls within the support of the real data, and $R_\beta$ reflects how much of the real data is covered by the support of synthetic data. We denote the covariate precision and recall with $P_{\alpha,\mathbf{X}}$ and $R_{\beta,\mathbf{X}}$ respectively, which are calculated by applying integrated $P_\alpha$ and $R_\beta$ to the covariate distribution only, as in (3) and (4).

**To assess the preservation of $P_\mathbf{X}$**

$$P_{\alpha,\mathbf{X}}(\mathcal{D}_{\text{real}}, \mathcal{D}_{\text{synth}}) = 1 - 2 \int_0^1 |\mathbb{P}(\tilde{\mathbf{X}}_{\text{synth}} \in \mathcal{S}_{\text{real}}^\alpha) - \alpha| \, d\alpha \tag{3}$$

$$R_{\beta,\mathbf{X}}(\mathcal{D}_{\text{real}}, \mathcal{D}_{\text{synth}}) = 1 - 2 \int_0^1 |\mathbb{P}(\tilde{\mathbf{X}}_{\text{real}} \in \mathcal{S}_{\text{synth}}^\beta) - \beta| \, d\beta \tag{4}$$

where $\tilde{\mathbf{X}}_\diamond$ and $\mathcal{S}_\diamond^\square$ are the embedding $\tilde{\mathbf{X}}_\diamond = \Phi(\mathbf{X}_\diamond)$ and $\square$-support as defined by Alaa et al. (2022), respectively.

We have $0 < P_{\alpha,\mathbf{X}}, R_{\beta,\mathbf{X}} < 1$, and scores near 1 indicate a realistic and diverse $Q_\mathbf{X}$. Together, these metrics can be used to answer **Q1**.

### 5.2.2 THE TREATMENT ASSIGNMENT MECHANISM $P_{W|\mathbf{X}}$

While in general we do not have access to $P_{W|\mathbf{X}}$ and $Q_{W|\mathbf{X}}$, we know that, for each $\mathbf{X} = \mathbf{x}$, they are Bernoulli distributions, since $W$ is a binary variable. The success probabilities can be estimated from $\mathcal{D}_{\text{real}}$ and $\mathcal{D}_{\text{synth}}$ with a probabilistic classifier, which can be used to form approximations of $P_{W|\mathbf{X}}$ and $Q_{W|\mathbf{X}}$. There is then an array of valid options to compare these approximations. We propose the use of Jensen-Shannon distance[3] given its desirable properties of symmetry, smoothness, and boundedness (we discuss alternatives in Appendix E). For a given probabilistic classifier $\hat{\pi}$, we define $\hat{P}_{W|\mathbf{X}=\mathbf{x}} = \text{Bern}(\hat{\pi}_{\text{real}}(\mathbf{x}))$ and $\hat{Q}_{W|\mathbf{X}=\mathbf{x}} = \text{Bern}(\hat{\pi}_{\text{synth}}(\mathbf{x}))$ where $\hat{\pi}_{\text{real}}$ and $\hat{\pi}_{\text{synth}}$ are trained on $\mathcal{D}_{\text{real}}$ and $\mathcal{D}_{\text{synth}}$ respectively, and we measure the preservation of $P_{W|\mathbf{X}}$ as in (5).

**To assess the preservation of $P_{W|\mathbf{X}}$**

$$\text{JSD}_\pi(\mathcal{D}_{\text{real}}, \mathcal{D}_{\text{synth}}) = 1 - \mathbb{E}_{P_\mathbf{X}} \left[ \sqrt{\frac{1}{2} D_{KL}(\hat{P}_{W|\mathbf{X}=\mathbf{x}} || M) + \frac{1}{2} D_{KL}(\hat{Q}_{W|\mathbf{X}=\mathbf{x}} || M)} \right] \tag{5}$$

where $M = \frac{1}{2}(\hat{P}_{W|\mathbf{X}=\mathbf{x}} + \hat{Q}_{W|\mathbf{X}=\mathbf{x}})$ and $D_{KL}$ is KL divergence using $\log_2$.

$\text{JSD}_\pi$ can be used to answer **Q2**. We have $0 < \text{JSD}_\pi < 1$, with scores near 1 indicating that $Q_{W|\mathbf{X}}$ matches $P_{W|\mathbf{X}}$ well. The validity of $\text{JSD}_\pi$ will depend on the accuracy of $\hat{\pi}$, so conducting $\hat{\pi}$ model selection is an important pre-evaluation step, although amongst reasonable model choices which exhibit similar performance, the information offered by $\text{JSD}_\pi$ will not significantly differ.

### 5.2.3 THE OUTCOME GENERATION MECHANISM $P_{Y|W,\mathbf{X}}$

To evaluate the preservation of $P_{Y|W,\mathbf{X}}$, we consider a treatment effect analogue of predictive utility. In this, we address the unavailability of ground-truths by aiming for agreement in performance on $\mathcal{D}_{\text{real}}$ and $\mathcal{D}_{\text{synth}}$, rather than attempting to quantify error from an oracle value. Such evaluation is inherently task dependent, yet the specific quantity $\mathcal{D}_{\text{synth}}$ may be used to estimate is unclear. Assessment should therefore centre on a complex task, in which comparable performance will likely imply the same for simpler tasks. In this case, we consider the most difficult treatment effect task likely to arise in the medical field—CATE estimation—as similarity in this between $\mathcal{D}_{\text{synth}}$ and $\mathcal{D}_{\text{real}}$ will tend to imply similarity in simpler tasks, such as ATE estimation. Therefore, we evaluate how well $Q_{Y|W,\mathbf{X}}$ preserves $P_{Y|W,\mathbf{X}}$ by calculating the PEHE between synthetic- and real-trained CATE

---

[3]$\text{JSD}(P \| Q) = \sqrt{\frac{1}{2} D_{\text{KL}}(P \| M) + \frac{1}{2} D_{\text{KL}}(Q \| M)}$, where $M = \frac{1}{2}(P + Q)$ and $D_{\text{KL}}$ is KL divergence.

learners (see Appendix E for alternatives). Given a family $\mathcal{F}$ of CATE learners $\hat{\tau}$, where $\hat{\tau}_{\text{real}}$ and $\hat{\tau}_{\text{synth}}$ are trained on $\mathcal{D}_{\text{real}}$ and $\mathcal{D}_{\text{synth}}$ respectively, we assess the preservation of $P_{Y|W,\mathbf{X}}$ as in (6).

**To assess the preservation of $P_{Y|W,\mathbf{X}}$**

$$U_{\text{PEHE}}(\mathcal{D}_{\text{real}}, \mathcal{D}_{\text{synth}}) = \frac{1}{|\mathcal{F}|} \sum_{\hat{\tau} \in \mathcal{F}} \sqrt{\mathbb{E}_{P_\mathbf{X}}[(\hat{\tau}_{\text{synth}}(\mathbf{X}) - \hat{\tau}_{\text{real}}(\mathbf{X}))^2]} \qquad (6)$$

$U_{\text{PEHE}}$ can answer **Q3**. We average over $\mathcal{F}$ since CATE model validation is difficult (Curth & Van Der Schaar, 2023), so $\hat{\tau}$ cannot be set as the best performing model in a similar fashion as is done for $\text{JSD}_\pi$ (we discuss choices for $\mathcal{F}$ in Appendix E.3.2). As such, $U_{\text{PEHE}}$ rewards generators which permit proximity in CATE estimations across a wide array of potential learners, where a lower $U_{\text{PEHE}}$ indicates better preservation of $P_{Y|W,\mathbf{X}}$.

# 6 GENERATING SYNTHETIC DATA CONTAINING TREATMENTS

To illustrate the standard DGP of data contraining treatments, shown in the middle of Figure 9, consider a simple hospital dataset. Patient covariates $\mathbf{X}$, such as height, weight etc., are drawn from an underlying covariate distribution $P_\mathbf{X}$, which is dictated by the local population. Treatments are then assigned by a domain expert, such as a doctor, conditioned on $\mathbf{X}$, i.e. $W \sim P_{W|\mathbf{X}}$. Finally, patients' outcomes are dictated by the dynamics of their ailments, conditional upon $W$ and $\mathbf{X}$, i.e. $Y \sim P_{Y|\mathbf{X},W}$. We now propose STEAM, a novel method for generating *Synthetic data for Treatment Effect Analysis in Medicine* which mimics the real DGP.

## 6.1 STEAM

Mimicry of the real DGP acts as an inductive bias, pushing $Q$ closer towards the $P$ in structure, and directly targeting each distributions from our desiderata. STEAM, shown on the right of Figure 9, conducts a three-step generation process, involving the following:

1. $Q_\mathbf{X}$. $\mathbf{X}$ is generated from a generative model trained to match the covariate distribution $P_\mathbf{X}$.

2. $Q_{W|\mathbf{X}}$. Treatments are assigned according to a propensity function trained on $\mathcal{D}_{\text{real}}$. If $\mathcal{D}_{\text{real}}$ is experimental data with known $P_{W|\mathbf{X}}$, then $Q_{W|\mathbf{X}}$ can be directly set as the true distribution, negating the need for any optimisation at this step.

3. $Q_{Y|W,\mathbf{X}}$. PO estimators are trained to match $P_{Y|W=0,\mathbf{X}}$ and $P_{Y|W=1,\mathbf{X}}$, and the relevant outcome is generated for each instance based on their assigned treatment.

Each component can be defined with any relevant model. $Q_\mathbf{X}$ can be any generative model, $Q_{W|\mathbf{X}}$ can be any classifier, and $Q_{Y|W,\mathbf{X}}$ can use any regressors.

## 6.2 DIFFERENTIAL PRIVACY WITH STEAM

Theoretical guarantees of the privacy of synthetic data are often required in high-stakes scenarios, such as medicine. STEAM can permit this, satisfying DP when its three component models do, as an application of the post-processing and composition theorems of DP (Dwork & Roth, 2014).

**Proposition 1.** *If $Q_\mathbf{X}$, $Q_{W|\mathbf{X}}$, and $Q_{Y|\mathbf{X}}$ satisfy $(\epsilon_\mathbf{X}, \delta_\mathbf{X})$-, $(\epsilon_W, \delta_W)$-, and $(\epsilon_Y, \delta_Y)$-differential privacy respectively, STEAM satisfies $(\epsilon_{total}, \delta_{total})$-differential privacy, where $\epsilon_{total} = \epsilon_\mathbf{X} + \epsilon_W + \epsilon_Y$, $\delta_{total} = \delta_\mathbf{X} + \delta_W + \delta_Y$.*

*Proof.* See Appendix G. □

There are a number of existing DP generative models, classifiers, and regressors which can be set as $Q_\mathbf{X}$, $Q_{W|\mathbf{X}}$, and $Q_{Y|W,\mathbf{X}}$ respectively to enable this.

# 7 EMPIRICAL ANALYSIS

We now demonstrate the superior performance of STEAM. In Section 7.1, we compare STEAM with generic generation methods in the non-DP setting. In Section 7.2, we examine performance in targeted settings to better understand where STEAM is particularly successful. In Section 7.3 we demonstrate STEAM's capability in satisfying DP generation. To avoid infeasible model selection and unwieldy notation, in STEAM we consistently model $Q_{W|\mathbf{X}}$ using logistic regression, and $Q_{Y|W,\mathbf{X}}$ using T-learner (Künzel et al., 2019) PO estimators. We use the open source `synthcity` (Qian et al., 2023) for all generative models, and we indicate which we set for $Q_\mathbf{X}$ in STEAM with subscript, i.e. STEAM$_\diamond$ uses generative model $\diamond$ for $Q_\mathbf{X}$. We detail experimental set-ups in Appendix H.

Table 3: $P_{\alpha,\mathbf{X}}$, $R_{\beta,\mathbf{X}}$, $\text{JSD}_\pi$, and $U_{\text{PEHE}}$ for the best performing standard and STEAM models on medical data. Full results in Table 11. Averaged over 20 runs, with 95% CIs. Bold indicates significant differences.

| Dataset | Model | $P_{\alpha,\mathbf{X}}$ ($\uparrow$) | $R_{\beta,\mathbf{X}}$ ($\uparrow$) | $\text{JSD}_\pi$ ($\uparrow$) | $U_{\text{PEHE}}$ ($\downarrow$) |
|---|---|---|---|---|---|
| ACTG | TVAE | $0.926 \pm 0.013$ | $0.483 \pm 0.010$ | $0.946 \pm 0.004$ | $0.564 \pm 0.017$ |
|  | STEAM $_{\text{TVAE}}$ | $0.929 \pm 0.008$ | $0.486 \pm 0.009$ | $\mathbf{0.958 \pm 0.004}$ | $\mathbf{0.492 \pm 0.011}$ |
| IHDP | CTGAN | $0.663 \pm 0.018$ | $0.419 \pm 0.013$ | $0.888 \pm 0.010$ | $2.521 \pm 0.161$ |
|  | STEAM $_{\text{CTGAN}}$ | $0.674 \pm 0.014$ | $0.424 \pm 0.011$ | $\mathbf{0.928 \pm 0.009}$ | $\mathbf{1.709 \pm 0.052}$ |
| ACIC | TVAE | $0.763 \pm 0.011$ | $0.515 \pm 0.006$ | $0.926 \pm 0.007$ | $4.202 \pm 0.134$ |
|  | STEAM $_{\text{TVAE}}$ | $0.767 \pm 0.009$ | $0.514 \pm 0.004$ | $\mathbf{0.972 \pm 0.002}$ | $\mathbf{2.013 \pm 0.112}$ |

## 7.1 Generation of medical data containing treatments

**Setup**

We compare STEAM with state-of-the-art generic tabular data generators in the non-DP setting, across three medical datasets:

1. **AIDS Clinical Trial Group (ACTG) study 175.** A clinical trial on subjects with HIV-1 (Hammer et al., 1996).

2. **Infant Health and Development Program (IHDP).** A semi-synthetic medical dataset, with real covariates and simulated outcomes, using data from a randomised experiment designed to evaluate the effect of specialist childcare on the cognitive test scores of premature infants (Brooks-Gunn et al., 1992).

3. **Atlantic Causal Inference Competition 2016 (ACIC).** A semi-synthetic medical dataset, with real covariates and simulated outcomes, containing data from the Collaborative Perinatal Project (Niswander, 1972).

ACTG allows us to assess performance on real-world medical data, and the IHDP and ACIC will be familiar to the causal inference community. We use baselines across the major families of tabular data generators (CTGAN, TVAE (Xu et al., 2019), TabDDPM (Kotelnikov et al., 2022), ARF (Watson et al., 2023), and normalising flow (Rezende & Mohamed, 2016)).

**Takeaway**

We display the performance across our metrics of the best performing standard model, and its STEAM analogue, on each dataset in Table 3 (extended results in Appendix I). STEAM and standard generation demonstrate similar performance in terms of desiderata (i), as $P_{\alpha,\mathbf{X}}$ and $R_{\beta,\mathbf{X}}$ are similar across datasets. This is expected, since both methods approach the modelling of $P_{\mathbf{X}}$ similarly. Larger differences occur in terms of desiderata (ii) and (iii). $\text{JSD}_\pi$ and $U_{\text{PEHE}}$ are improved by STEAM at a statistically significant level across all datasets, indicating that targeted modelling of $P_{W|\mathbf{X}}$ and $P_{Y|W,\mathbf{X}}$ improves their preservation. The most notable improvement is in the $U_{\text{PEHE}}$ metric, which is up to twice as good in STEAM models.

## 7.2 Comparisons on simulated data

To investigate the performance delta between STEAM and standard generation, we design experiments on simulated data using a DGP with tunable *experimental knobs*, similar to that proposed in Crabbé et al. (2022). Our tunable knobs include covariate dimensionality $d$, propensity function $\pi : \mathcal{X}^{(d)} \to [0,1]$, and prognostic and predictive functions $\mu_{\text{prog.}}, \mu_{\text{pred.}} : \mathcal{X}^{(d)} \to \mathbb{R}$.[4] Sample $i$ is generated by drawing $\mathbf{X}^{(i)} \sim \mathcal{N}(0, I_d)$, $W^{(i)} \sim \text{Bern}[\pi(\mathbf{X}^{(i)})]$, and $Y^{(i)} \sim \mathcal{N}(\mu_{\text{prog.}}(\mathbf{X}^{(i)}) + W^{(i)} \cdot \mu_{\text{pred.}}(\mathbf{X}^{(i)}), 1)$. With this DGP, we can assess performance on datasets tailored to specific situations. Across experiments, we consistently compare between TabDDPM and STEAM$_{\text{TabDDPM}}$, and the default settings for each experimental knob are:

$$d = 10, \pi(\mathbf{X}) = (1 + e^{-1/2(X_1{}^2 + X_2{}^2)})^{-1}, \mu_{\text{prog.}}(\mathbf{X}) = X_1{}^2 + X_2{}^2, \mu_{\text{pred.}}(\mathbf{X}) = X_3{}^2 + X_4{}^2$$

---

[4]Prognostic variables affect an outcome regardless of treatment, while predictive variables only affect treated outcomes. Prognostic and predictive functions dictate the effect of each covariate on the outcome.

### 7.2.1 COVARIATE DIMENSIONALITY

**Setup**

To investigate performance as $\mathcal{D}_{\text{real}}$ increases in dimensionality, we vary $d \in \{5, 10, 20, 50\}$, with all other settings at default.

**Takeaway**

The performance delta between STEAM and standard generation grows with the dimensionality of $\mathbf{X}$. This follows the intuition that, as $d$ grows, $P_{\mathbf{X}}$ will dominate the joint distribution, and the comparatively small $P_{W|\mathbf{X}}$ and $P_{Y|W,\mathbf{X}}$ will be overlooked by standard models. The top of Figure 1 shows that, as $d$ increases, both $\text{STEAM}_{\text{TabDDPM}}$ and TabDDPM preserve $P_{Y|W,\mathbf{X}}$ worse, however $\text{STEAM}_{\text{TabDDPM}}$ is less affected by $d$. The bottom of Figure 1 is similar, showing that TabDDPM degrades in performance more than $\text{STEAM}_{\text{TabDDPM}}$ in preserving $P_{W|\mathbf{X}}$ as $d$ grows. Direct modelling with $Q_{W|\mathbf{X}}$ and $Q_{Y|W,\mathbf{X}}$ allows these small, but important, components to be better preserved in high dimensions.

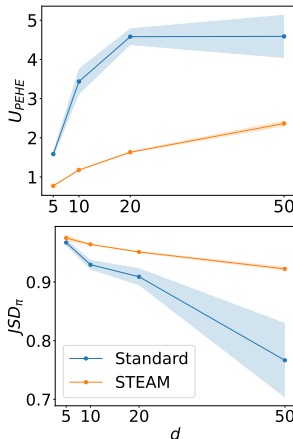

Figure 1: $U_{\text{PEHE}}$ ($\downarrow$) and $\text{JSD}_\pi$ ($\uparrow$) as $d$ increases. Averaged over 10 runs, shaded area represents 95% CIs.

### 7.2.2 TREATMENT ASSIGNMENT COMPLEXITY

**Setup**

To investigate performance as $P_{W|\mathbf{X}}$ increases in complexity, we vary the number of covariates upon which it depends. We set $\pi(\mathbf{X}) = (1 + e^{-1/K \sum_{k=1}^K X_k^2})^{-1}$ for $K \in \{1, 2, 3, 4, 5\}$, with all other settings at default.

**Takeaway**

STEAM increasingly outperforms standard generation in preserving more complex $P_{W|\mathbf{X}}$. Figure 2 shows that, as $K$ increases, $\text{STEAM}_{\text{TabDDPM}}$ maintains a good estimate of $P_{W|\mathbf{X}}$, with $\text{JSD}_\pi$ consistently near 1. On the other hand, the estimate by standard TabDDPM degrades with $K$, widening the performance gap. Direct modelling allows more complex $P_{W|\mathbf{X}}$ to be preserved.

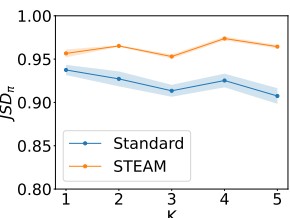

Figure 2: $\text{JSD}_\pi$ ($\uparrow$) as $K$ increases. Averaged over 10 runs, shaded area represents 95% CIs.

### 7.2.3 OUTCOME HETEROGENEITY

**Setup**

To investigate performance as outcomes become increasingly heterogeneous, we vary the number of covariates upon which $P_{Y|W,\mathbf{X}}$ depends. We set $\mu_{\text{pred.}}(\mathbf{X}) = \sum_{k=3}^K X_k^2$, $K \in \{3, 4, 5, 6, 7\}$, with all other settings at default.

**Takeaway**

As $P_{Y|W,\mathbf{X}}$ becomes increasingly heterogeneous, its preservation by $\text{STEAM}_{\text{TabDDPM}}$ degrades slightly, and much more dramatically for TabDDPM, as shown in Figure 3. Again, direct modelling with $Q_{Y|W,\mathbf{X}}$ better preserves complex distributions.

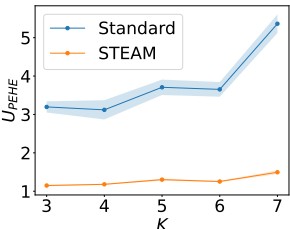

Figure 3: $U_{\text{PEHE}}$ ($\downarrow$) as $K$ increases. Averaged over 10 runs, shaded area represents 95% CIs.

These experiments demonstrate that the performance delta between STEAM and standard generation grows in complex settings. Whether difficulty arises from high-dimensionality, or through complex dependencies in $P_{W|\mathbf{X}}$ or $P_{Y|W,\mathbf{X}}$, STEAM increasingly outperforms in the more difficult scenarios. These situations are likely to emerge in real-world data, which is often highly complex, heightening the relevance of STEAM to the medical setting.

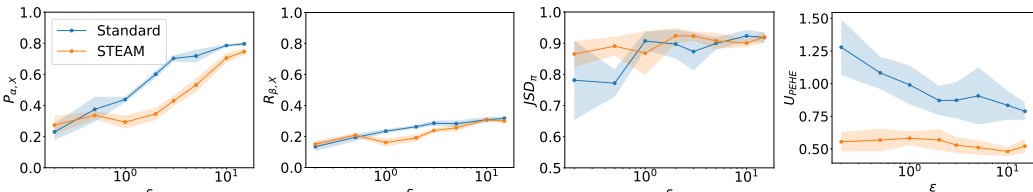

Figure 4: $P_{\alpha,\mathbf{X}}$ ($\uparrow$), $R_{\beta,\mathbf{X}}$ ($\uparrow$), $\mathrm{JSD}_\pi$ ($\uparrow$), and $U_{\mathrm{PEHE}}$ ($\downarrow$) evaluating $\mathrm{STEAM}_{\mathrm{AIM}}$ and standard AIM across privacy budgets. Averaged over 5 runs, shaded area represents 95% CIs.

## 7.3 DIFFERENTIALLY PRIVATE GENERATION WITH STEAM

**Setup**

We examine STEAM's performance in $(\epsilon, \delta)$-DP generation. For comparison we use $(\epsilon, \delta)$-AIM (McKenna et al., 2024), and for STEAM, we set $Q_{\mathbf{X}}$ as $(\epsilon/3, \delta/3)$-AIM, $Q_{W|\mathbf{X}}$ as an $(\epsilon/3, \delta/3)$-DP random forest, and $Q_{Y|W,\mathbf{X}}$ as an $(\epsilon/3, \delta/3)$-T-Learner, such that STEAM is also $(\epsilon, \delta)$-DP. We compare performance on the ACTG dataset across $\epsilon \in \{0.25, 0.5, 1, 2, 3, 5, 10, 15\}$ with $\delta = 10^{-6}$.

**Takeaway**

Figure 4 shows the results, and results with more baselines are in Appendix I.2. $\mathrm{STEAM}_{\mathrm{AIM}}$ models $P_{Y|W,\mathbf{X}}$ better on all tested values of $\epsilon$, as $U_{\mathrm{PEHE}}$ is significantly lower than for standard AIM. $P_{W|\mathbf{X}}$ is better modelled by $\mathrm{STEAM}_{\mathrm{AIM}}$ at small $\epsilon$, with equivalent performance between the methods at less conservative budgets. $P_{\mathbf{X}}$, on the other hand, is better preserved by standard AIM, scoring higher on $P_{\alpha,\mathbf{X}}$ and $R_{\beta,\mathbf{X}}$ at most $\epsilon$. This is likely because assigning $Q_{\mathbf{X}}$ one third of the budget of the standard AIM model and having it model largely the same distribution, save for the removed $W$ and $Y$, is prohibitively restrictive given the high-dimensionality of $\mathbf{X}$. As such, with uniform distribution of $(\epsilon, \delta)$ across each component, there is a trade-off between $\mathrm{STEAM}_{\mathrm{AIM}}$ and standard AIM, where $\mathrm{STEAM}_{\mathrm{AIM}}$ better preserves $P_{W|\mathbf{X}}$ and $P_{Y|W,\mathbf{X}}$, while standard AIM preserves $P_{\mathbf{X}}$ better. Distributing $(\epsilon, \delta)$ differently amongst $Q_{\mathbf{X}}$, $Q_{W|\mathbf{X}}$, and $Q_{Y|W,\mathbf{X}}$ could address this trade-off, as we discuss in Section 8 and Appendix M.

## 8 DISCUSSION

**Impact.** In this paper, we tackle a problem impeding progress in the causal inference for medicine community—unavailability of data. Existing synthetic data solutions are inadequate, producing poor quality data containing treatments, which are evaluated with misaligned metrics. Our evaluation and generation proposals, grounded in our desiderata which stem from the needs of analysts, remedy this. We enable generation of synthetic data of substantially higher quality, which we demonstrate across a range of experiments in Section 7, as well as in an additional ablative study (Appendix J) and hyperparameter stability study (Appendix K). Furthermore, we allow meaningful evaluation with our metrics, proposed in Section 5, that can answer the key questions *Q1-3* of downstream analysts from Section 1. Our paper's impact is heightened by the fact that STEAM increasingly outperforms standard generation in complex situations likely to arise in real-world settings. While we focus on medical data, our methods are also applicable to other fields where data contain treatments, such as education, marketing, and public policy, broadening our impact.

**Limitations.** STEAM has room for refinement. Uniform distribution of the privacy budget across $Q_{\mathbf{X}}$, $Q_{W|\mathbf{X}}$, and $Q_{Y|W,\mathbf{X}}$ during DP generation, as we do in Section 7.3, is sub-optimal, as particular component models may benefit from a larger share of $\epsilon$ depending on their importance and complexity (Appendix M). Also, generative models, used to model $Q_{\mathbf{X}}$, can struggle when covariate shift is high (Appendix N). This does not uniquely affect STEAM, as it occurs under standard generation as well, however it is important to acknowledge that poor performance may occur in this setting.

**Future work.** There are many future research directions in this setting. These include improving the limitations discussed above, and examining further applications of STEAM. For example, in this work we focus on static medical data, and longitudinal data, with continuous measurement of covariates, treatments, and outcomes, may require further novel thought (Appendix P).

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

# A EXAMPLES OF MISUSE OF MEDICAL DATA CONTAINING TREATMENTS

Here, we detail a select few synthetic data papers which use medical data, containing treatments, to demonstrate on, yet they do not consider the downstream task of causal inference and how their methods may need to be altered for this. Note that we do not claim that this list is exhaustive, as this is a pervasive problem in the synthetic data literature, and we mean only to provide a few examples here to demonstrate this problem, and provide motivation for our paper.

## A.1 MEDGAN

In the paper 'Generating Multi-label Discrete Patient Records using Generative Adversarial Networks' (Choi et al., 2018), the authors propose a GAN-based approach to generate 'realistic synthetic patient records'. In doing so, they experiment on multiple datasets containing treatments, including one from Sutter Palo Alto Medical Foundation (PAMF) which consists of longitudinal medical records of 258,000 patients, as well as the MIMIC-III dataset (Johnson et al., 2016), which includes 46,000 intensive care unit patient records. Given the nature of these datasets, they both include treatments administered to patients, and they therefore invite downstream analysts to conduct causal inference tasks, such as treatment effect estimation.

Nevertheless, in this paper, standard generation and evaluation practices are followed, not differentiating between covariates, treatments, and outcomes. In particular, the evaluation protocol relies on marginal comparisons, and predictive utility assessment, which offers limited information as to how well the generation method, `medGAN`, produces useful data for causal inference. Furthermore, `medGAN` itself draws samples directly from the joint distribution, not mimicking the DGP of treatment data, or optimising for the distributions most important for causal inference.

## A.2 TABDDPM

The paper 'TabDDPM: Modelling Tabular Data with Diffusion Models' (Kotelnikov et al., 2022) proposes a diffusion-based tabular data generation method. While not explicitly geared towards medical data, this paper does use medical data with variables that could be seen as treatments in its experiments. It therefore, at least implicitly, positions itself to work on data containing treatments, and invites users to conduct generation with `TabDDPM` on such data. Specifically, the cardiovascular disease dataset from `https://www.kaggle.com/datasets/sulianova/cardiovascular-disease-dataset` is used, and this data could be analysed via causal inference by setting 'physical activity' as a treatment, to estimate its effect on cardiovascular disease.

However, in this paper, only standard evaluation and generation methods are used, and the needs of downstream analysts pursuing causal inference tasks are not acknowledged. Evaluation involves only predictive utility measures, and the `TabDDPM` method generates all variables in a sample simultaneously, not optimising for the distributions most important for causal inference.

## A.3 GREAT

The paper 'Language Models are Realistic Tabular Data Generators' (Borisov et al., 2023) proposes an LLM-based generator. Similar to the `TabDDPM` paper, this paper is not explicitly geared towards medical data, but it demonstrates on medical data containing treatments, thereby implicitly condoning its use on this type of data. Specifically, the dataset `sick` from `https://www.openml.org/search?type=data&sort=runs&id=38&status=active` is demonstrated on, which could be analysed via causal inference to assess the effect of 'thyroxine' or 'antithyroid' treatments. Nevertheless, once again this paper does not consider downstream analysis involving causal inference, only evaluating its `GReaT` method with predictive utility metrics and a discriminator score.

## A.4 BENCHMARKING PROCESS FOR SYNTHETIC ELECTRONIC HEALTH RECORDS

Finally, the paper 'A Multifaceted benchmarking of synthetic electronic health record generation models' proposes a benchmarking framework for use on synthetic electronic health record (EHR) data (Yan et al., 2022). Naturally, EHRs will include treatments administered to patients, and they will likely be analysed with treatment effect estimation in mind. In the proposed benchmarking

framework, the evaluation procedures—including marginal comparison, correlation comparison, and predictive utility—do not differentiate between covariates, treatments, and outcomes, or acknowledge the needs of downstream analysts conducting causal inference.

## B   EXTENDED LITERATURE REVIEW

To provide useful context for readers, we extend on our literature review here.

**Evaluation.**   We extend on the synthetic data evaluation practices summarised in Table 1 here.

*Marginal comparison.* Assessing the distributional distance between synthetic and real marginals is often used to offer a quantitative assessment of how well individual variables are modelled (Yan et al., 2022; Tucker et al., 2020; Goncalves et al., 2020). The distance function $d$ to conduct this can be set from a variety of choices, including including KL divergence (Kullback & Leibler, 1951), Jensen-Shannon distance (Lin, 1991), Wasserstein distance (Kantorovich, 1960), Kolmogorov-Smirnov score (Massey Jr, 1951), MMD (Gretton et al., 2012), and many more.

*Correlation matrix comparison.* Correlation-based assessment can offer a sense of how well inter-dependencies between variables are modelled in synthetic data (Murtaza et al., 2023). This commonly involves calculating synthetic and real 2-way correlation matrices, and assessing their difference, by setting $d$ as a distance such as Frobenius norm (Goncalves et al., 2020) and absolute error (Kaur et al., 2020).

*Joint distribution comparison.* Metrics based on notions of statistical divergence can offer a means of quantifying how different the entire joint distributions of real and synthetic data are (Yoon et al., 2020; Tucker et al., 2020; Torfi et al., 2022). The distance function $d$ can be set to largely to the same family of functions as in the marginal comparison case.

*Precision and recall analysis.* Precision and recall, originally proposed for generative model assessment in Sajjadi et al. (2018), measure if generated samples are covered by real samples, and vice versa. Alpha precision and beta recall (Alaa et al., 2022) are refined versions of the original metrics which account for the densities of the real and generative distributions, rather than just comparing supports.

*Discriminator performance.* Discriminator performance is a slightly unique evaluation practice, involving a 'discriminator', which predicts whether instances are synthetic or real, where poor performance of the discriminator indicates realism in the synthetic data (Kaur et al., 2020; Lee et al., 2020; Emam et al., 2021; Borisov et al., 2023).

*Predictive utility.* Predictive utility metrics offer a practical evaluation of synthetic data by quantifying the performance of a synthetically-trained predictive model. The "train on synthetic, test on real" (TSTR) paradigm (Esteban et al., 2017) is the common approach to such assessment, measuring the accuracy of a synthetically-trained model in predicting a target label on a real test set.

**Generic generative models.**   We expand on the generic generative modelling paradigm outlined in Table 2 by describing specific existing generative models which adhere to it. These models approximate the real joint distribution using a diverse range of techniques.

*GAN-based models.* GANs (Goodfellow et al., 2020) consist of a generator and discriminator network, which are trained adversarially to generate and identify synthetic data, respectively, until the samples are realistic. Originally proposed for image generation, GANs have been adapted to tabular generation, and there are many methods which adopt this popular architecture (e.g. CTGAN (Xu et al., 2019), TableGAN (Park et al., 2018)), including those specifically designed for medical data (e.g. MedGAN (Choi et al., 2018)).

*VAE-based models.* VAEs (Kingma & Welling, 2022) are another common architecture, which learn to encode data into a lower-dimensional latent space and then decode it back to reconstruct the original data. They generate new data samples by sampling from the latent distribution and decoding these samples, and their application to tabular data involve techniques to handle mixed data types (e.g.TVAE (Xu et al., 2019)), and regularisation for improved robustness (e.g. RTVAE Akrami et al. (2020)).

*Diffusion-based models.* Diffusion models (Ho et al., 2020; Song et al., 2020) learn the gradient of the data distribution, and generate data via progressive denoising, beginning with a noisy sample and using a neural network to predict and remove noise over a number of timesteps, including for tabular data (e.g. TabDDPM (Kotelnikov et al., 2022)).

*Forest-based models.* Random forests can estimate the density of a probability distribution, as leaf nodes partition the data space into distinct hyper-rectangles with estimated densities of the proportion of samples which fall into them. Samples can then be drawn from this estimated density. Random forests can easily handle heteorgeneous data types, so their application to tabular data synthesis is natural. They are particularly fast to train and generate from (Shi & Horvath, 2006; Watson et al., 2023)).

*Normalizing flow-based models.* Normalizing flows estimate the target density by transforming a tractable density (e.g. a Gaussian) into the target through a series of invertible transformations, called 'flows'. Probabilities from the target distribution can then be found using the change of variables formula (e.g. Rezende & Mohamed (2016)). Recent work has shown their theoretical similarity to diffusion models (Lipman et al., 2023).

**Causal generative models.** Causal generative models (Komanduri et al., 2024; Blöbaum et al., 2024; Chao et al., 2024) are a class of generative model, distinct from generic tabular data generators, that approximate the underlying structural causal model (SCM) (Pearl, 2009) of a dataset. While such models are related to our work with STEAM, and are likely to better preserve causal relationships than generic generators in settings where they can be used, their assumptions can restrict their practical use cases. In comparison to STEAM, they generally differ in terms of their (1) assumptions, (2) motivation, and (3) flexibility, which we detail here.

(1): Importantly, causal generative models typically assume that the data holder has knowledge of the entire causal graph of the real data, which is a more restrictive assumption than we make in this work. We assume that our specification of the underlying DGP $\mathbf{X} \sim P_{\mathbf{X}}$, $W \sim P_{W|\mathbf{X}}$, $Y \sim P_{Y|W,\mathbf{X}}$ is correct for datasets containing treatments, but we do not require knowledge of the causal graph, as we do not need to know the causal links between individual variables. We do not assume knowledge of the causal relationships amongst the covariates, nor knowledge of which covariates cause treatment assignment or patient outcomes. As such, we make less restrictive assumptions than works which require knowledge of a causal graph, and we suggest that our approach is more realistic in complex, real-world scenarios, such as those that arise in medicine, where the true causal graph is unlikely to be available.

(2): The motivation for causal generative models is typically to allow generation of data to answer graph-specific interventional and counterfactual queries, that require knowledge of the full causal graph. With STEAM, however, we seek to generate useful synthetic data only from the observational distribution, for use by analysts with goals such as treatment effect estimation (e.g., CATE).

(3): STEAM's design can, in principle, incorporate any generative model for $Q_{\mathbf{X}}$, essentially acting as a wrapper around $Q_{\mathbf{X}}$ to improve its generation quality for causal inference tasks. This allows STEAM to very easily empower many existing generative modelling frameworks, without having to incorporate bespoke generators. Also, it allows STEAM models to continuously improve along with the base generative model. Existing causal generative models do not generally allow such flexibility, and therefore cannot as easily benefit from future improvements in underlying generic generative models.

Despite these differences, we conduct empirical comparisons between STEAM and some baseline causal generative models in Appendix O.

**Privacy.** Despite the popularity of some of the above generators, memorisation of training samples is a phenomenon observed in generative models (Ghalebikesabi et al., 2023). Therefore, provably private generation is often desired to limit the amount of information leaked. Differential privacy Dwork et al. (2006) is the most common standard adopted, and there are multiple generators which guarantee this, including GAN-based methods (e.g. PATE-GAN (Jordon et al., 2018), DP-GAN (Xie

et al., 2018)) and query-based methods (e.g. GEM (Liu et al., 2021), MST (McKenna et al., 2021), RAP (Aydore et al., 2021), AIM (McKenna et al., 2022)).

## C  THEOREM 1 PROOF

**Theorem 1.** *Let $P$, $Q^{\theta_1}$, $Q^{\theta_2}$ be of the form described in Section 5.1, and $\mathcal{M}$ be KL divergence. If we assume that $Q^{\theta_1}$ and $Q^{\theta_2}$ have sufficient capacity to have bounded error on each component, i.e. $\forall i$, $0 < \mathcal{M}(P_{\mathbf{X}_i}, Q_{\mathbf{X}_i}^{\theta_{\mathbf{X}}}) < \varepsilon_{\mathbf{X}}$, and $0 < \mathcal{M}(P_{W|\mathbf{X}}, Q_{W|\mathbf{X}}^{\theta_{W,k}}) < \varepsilon_{W,k}$, and $0 < \mathcal{M}(P_{Y|W,\mathbf{X}}, Q_{Y|W,\mathbf{X}}^{\theta_{Y,k}}) < \varepsilon_{Y,k}$, then:*

$$\frac{\mathcal{M}(P_{\mathbf{X},W,Y}, Q_{\mathbf{X},W,Y}^{\theta_1})}{\mathcal{M}(P_{\mathbf{X},W,Y}, Q_{\mathbf{X},W,Y}^{\theta_2})} \to 1, \ as \ d \to \infty \tag{7}$$

*Proof.* From the factorizations of $P$ and $Q$, KL divergence decomposes:

$$D_{\mathrm{KL}}(P\|Q^{\theta_k}) = \sum_{i=1}^{d} D_{\mathrm{KL}}(P_{\mathbf{X}_i}\|Q_{\mathbf{X}_i}^{\theta_{\mathbf{X}}}) + \mathbb{E}_{P_{\mathbf{X}}}\left[D_{\mathrm{KL}}(P_{W|\mathbf{X}}\|Q_{W|\mathbf{X}}^{\theta_{W,k}})\right] + \mathbb{E}_{P_{\mathbf{X},W}}\left[D_{\mathrm{KL}}(P_{Y|W,\mathbf{X}}\|Q_{Y|W,\mathbf{X}}^{\theta_{Y,k}})\right]$$
(8)

As such, the following holds for when KL divergence is set as $\mathcal{M}$.

Define the ratio:

$$R(d) = \frac{\mathcal{M}(P_{\mathbf{X},W,Y}, Q_{\mathbf{X},W,Y}^{\theta_1})}{\mathcal{M}(P_{\mathbf{X},W,Y}, Q_{\mathbf{X},W,Y}^{\theta_2})}.$$

Substituting the decompositions, we have:

$$R(d) = \frac{\sum_{i=1}^{d} \mathcal{M}(P_{\mathbf{X}_i}, Q_{\mathbf{X}_i}^{\theta_{\mathbf{X}}}) + \mathbb{E}_{P_{X}}\left[\mathcal{M}(P_{W|\mathbf{X}}, Q_{W|\mathbf{X}}^{\theta_{W,1}})\right] + \mathbb{E}_{P_{X,W}}\left[\mathcal{M}(P_{Y|W,\mathbf{X}}, Q_{Y|W,\mathbf{X}}^{\theta_{Y,1}})\right]}{\sum_{i=1}^{d} \mathcal{M}(P_{\mathbf{X}_i}, Q_{\mathbf{X}_i}^{\theta_{\mathbf{X}}}) + \mathbb{E}_{P_{X}}\left[\mathcal{M}(P_{W|\mathbf{X}}, Q_{W|\mathbf{X}}^{\theta_{W,2}})\right] + \mathbb{E}_{P_{X,W}}\left[\mathcal{M}(P_{Y|W,\mathbf{X}}, Q_{Y|W,\mathbf{X}}^{\theta_{Y,2}})\right]}.$$

As the dimensionality $d$ increases, the marginal summations $\sum_{i=1}^{d} \mathcal{M}(P_{\mathbf{X}_i}\|Q_{\mathbf{X}_i}^{\theta_{\mathbf{X}}})$ grow linearly with $d$, since each $\mathcal{M}(P_{\mathbf{X}_i}\|Q_{\mathbf{X}_i}^{\theta_{\mathbf{X}}})$ is, by assumption, non-negative, and they therefore dominate the bounded conditional contributions:

$$\mathbb{E}_{P_X}\left[\mathcal{M}(P_{W|\mathbf{X}}, Q_{W|\mathbf{X}}^{\theta_{W,k}})\right] < \varepsilon_{W,k},$$

$$\mathbb{E}_{P_{X,W}}\left[\mathcal{M}(P_{Y|W,\mathbf{X}}, Q_{Y|W,\mathbf{X}}^{\theta_{Y,k}})\right] < \varepsilon_{Y,k}.$$

Thus, $\mathcal{M}(P_{\mathbf{X},W,Y}, Q_{\mathbf{X},W,Y}^{\theta_k}) \sim \sum_{i=1}^{d} \mathcal{M}(P_{\mathbf{X}_i}, Q_{\mathbf{X}_i}^{\theta_{\mathbf{X}}})$ and $R(d) \sim \frac{\sum_{i=1}^{d} \mathcal{M}(P_{\mathbf{X}_i}, Q_{\mathbf{X}_i}^{\theta_{\mathbf{X}}})}{\sum_{i=1}^{d} \mathcal{M}(P_{\mathbf{X}_i}, Q_{\mathbf{X}_i}^{\theta_{\mathbf{X}}})} = 1$.

Therefore:

$$R(d) \to 1, \ as \ d \to \infty$$

$\square$

## D  EMPIRICAL DEMONSTRATIONS OF CURRENT METRIC FAILURE

Table 4: Joint-distribution-level metrics on $\mathcal{D}^i_{\text{synth}}$ which differ in $Q^i_{Y|W,\mathbf{X}}$ architecture only. Averaged over 10 runs, with 95% CIs.

| $Q^i_{Y|W,\mathbf{X}}$ | $P_\alpha$ ($\uparrow$) | $R_\beta$ ($\uparrow$) | Inv. KL ($\uparrow$) | KS ($\uparrow$) | WD ($\downarrow$) | JSD ($\downarrow$) | Oracle ($\downarrow$) |
|---|---|---|---|---|---|---|---|
| T-Learner | $0.927 \pm 0.001$ | $0.584 \pm 0.006$ | $0.947 \pm 0.000$ | $0.979 \pm 0.000$ | $0.002 \pm 0.000$ | $0.002 \pm 0.000$ | $0.525 \pm 0.012$ |
| TARNet | $0.919 \pm 0.002$ | $0.573 \pm 0.005$ | $0.950 \pm 0.006$ | $0.985 \pm 0.001$ | $0.002 \pm 0.000$ | $0.002 \pm 0.000$ | $0.616 \pm 0.015$ |
| DragonNet | $0.921 \pm 0.001$ | $0.574 \pm 0.004$ | $0.947 \pm 0.000$ | $0.984 \pm 0.001$ | $0.002 \pm 0.000$ | $0.002 \pm 0.000$ | $0.618 \pm 0.007$ |
| S-Learner | $0.926 \pm 0.002$ | $0.579 \pm 0.007$ | $0.957 \pm 0.009$ | $0.990 \pm 0.000$ | $0.002 \pm 0.000$ | $0.001 \pm 0.000$ | $1.279 \pm 0.015$ |

## D.1 FAILURE TO IDENTIFY CHANGES TO THE OUTCOME GENERATION MECHANISM

We demonstrate this with a simple experiment investigating how four $\mathcal{D}_{\text{synth}}$ of size $n = 1000$, which only differ in their outcome generation mechanisms, are assessed by an array of current metrics. We simulate $\mathcal{D}_{\text{real}}$ from a simple DGP with 10 covariates with $P_\mathbf{X} = \mathcal{N}(0, I)$, $P_{W|\mathbf{X}} = \text{Bern}(0.5)$, $P_{Y|W,\mathbf{X}} = \mathcal{N}(W \cdot X_1{}^2, 1)$. We generate four $\mathcal{D}^i_{\text{synth}}$ with the same $Q^i_\mathbf{X} \overset{d}{=} P_\mathbf{X}$, $Q^i_{W|\mathbf{X}} \overset{d}{=} P_{W|\mathbf{X}}$, $\forall i \in \{1, 2, 3, 4\}$. We vary each $Q^i_{Y|W,\mathbf{X}} \sim \mathcal{N}(W \cdot \Phi_i(\mathbf{X}, 1) + (1 - W) \cdot \Phi_i(\mathbf{X}, 0), 1)$ where $\Phi_i$ represents a potential outcome (PO) estimator with the architecture from either an S-Learner, T-Learner (Künzel et al., 2019), DragonNet (Shi et al., 2019), or TARNet (Shalit et al., 2017). These four architectures will model $Q^i_{Y|W,X}$ differently, inducing the only point of variation amongst the $\mathcal{D}^i_{\text{synth}}$.

Since we simulate $\mathcal{D}_{\text{real}}$, we know the ground-truth treatment effects, and an oracle metric can be established to determine the true quality of each $\mathcal{D}^i_{\text{synth}}$. We define this as the precision of estimating heterogeneous effects (PEHE) (Hill, 2011) of estimates from a CATE learner trained on $\mathcal{D}^i_{\text{synth}}$ and the ground-truth CATEs. In Table 4 we report the scores of $P_\alpha$, $R_\beta$ (Alaa et al., 2022), inverse KL divergence (Kullback & Leibler, 1951), Kolmogorov-Smirnov (KS) score (Massey Jr, 1951), Wasserstein distance (WD) (Kantorovich, 1960), and Jensen-Shannon distance (JSD) (Lin, 1991) on each $\mathcal{D}^i_{\text{synth}}$. All report very similar scores across the $\mathcal{D}^i_{\text{synth}}$, with most offering no statistically significant best option, suggesting that their quality is the same. The oracle metric, however, determines that $\mathcal{D}^i_{\text{synth}}$ using a T-Learner for $\Phi_i$ is a clear best, and $\mathcal{D}^i_{\text{synth}}$ with an S-Learner as $\Phi_i$ is more than twice as bad at preserving the true treatment effects. Clearly, even in a moderately sized dataset, these metrics cannot reliably identify changes in $Q^i_{Y|W,\mathbf{X}}$, despite the large effect that this distribution has on downstream performance.

**Comparison to $U_{\text{PEHE}}$** Since we only alter $Q^i_{Y|W,\mathbf{X}}$ between each $\mathcal{D}^i_{\text{synth}}$, all have the same $P_{\alpha,\mathbf{X}}$, $R_{\beta,\mathbf{X}}$, and $\text{JSD}_\pi$. In Table 5 we report $U_{\text{PEHE}}$ on each dataset, and we see that it fully reproduces the oracle ranking, and correctly identifies the best dataset to a statistically significant level, which no existing metric could do.

Table 5: $U_{\text{PEHE}}$ on $\mathcal{D}^i_{\text{synth}}$ with varied $Q^i_{Y|W,\mathbf{X}}$. Averaged over 10 runs, with 95% CIs.

| $Q^i_{Y|W,\mathbf{X}}$ | $U_{\text{PEHE}}$ ($\downarrow$) | Oracle ($\downarrow$) |
|---|---|---|
| T-Learner | $0.693 \pm 0.013$ | $0.525 \pm 0.012$ |
| TARNet | $0.731 \pm 0.016$ | $0.616 \pm 0.015$ |
| DragonNet | $0.754 \pm 0.019$ | $0.618 \pm 0.007$ |
| S-Learner | $0.906 \pm 0.019$ | $1.279 \pm 0.015$ |

## D.2 FAILURE TO IDENTIFY CHANGES TO THE TREATMENT ASSIGNMENT MECHANISM

We conduct a similar experiment varying $Q^i_{W|\mathbf{X}}$ across three $\mathcal{D}^i_{\text{synth}}$. We simulate $\mathcal{D}_{\text{real}} \sim P_{\mathbf{X},W,Y}$ from a DGP with 5 covariates, all of which contribute to the propensity score. We set $P_\mathbf{X} = \mathcal{N}(0, I)$, $P_{W|\mathbf{X}} = \text{Bern}(\pi(\mathbf{X}))$, $\pi(\mathbf{X}) = (1 + e^{-1/5 \sum_{i=1}^5 X_i})^{-1}$, $P_{Y|W,\mathbf{X}} = \mathcal{N}(0, 1)$. We generate three $\mathcal{D}^i_{\text{synth}} \sim Q^i_{\mathbf{X},W,Y}$ which vary only in the degree to which they correctly model $\pi(\mathbf{X})$ by setting $Q^i_\mathbf{X} \overset{d}{=} P_\mathbf{X}$, $Q^i_{Y|W,\mathbf{X}} \overset{d}{=} P_{Y|W,\mathbf{X}}$, $\forall i \in \{1, 2, 3\}$ and $Q^i_{W|\mathbf{X}} = \text{Bern}(\pi_i(\mathbf{X}))$ where $\pi_1(\mathbf{X}) = (1 + e^{-X_1})^{-1}$, $\pi_2(\mathbf{X}) = (1 + e^{-1/3 \sum_{i=1}^3 X_i})^{-1}$, and $\pi_3(\mathbf{X}) = (1 + e^{-1/5 \sum_{i=1}^5 X_i})^{-1}$. In this way, we know that, in truth, $Q^3_{\mathbf{X},W,Y}$ is a better model than $Q^2_{\mathbf{X},W,Y}$, which in turn is better than $Q^1_{\mathbf{X},W,Y}$, and we can now assess how well existing metrics, and our $\text{JSD}_\pi$ metric, recover this ranking.

We display the scores of $P_\alpha$, $R_\beta$, inverse KL, Kolmogorov-Smirnov score, Wasserstein distance, Jensen-Shannon distance, and our metric $\text{JSD}_\pi$ on each $\mathcal{D}^i_{\text{synth}}$ in Table 6. We see that the existing metrics report very similar scores across the three datasets, and none offer a statistically significant

Table 6: # correct var.: The number of correctly identified variables in the propensity score. $P_\alpha$: $\alpha$ precision. $R_\beta$: $\beta$ recall. Inv. KL: Inverse KL divergence. KS: Kolmogorov-Smirnov score. WD: Wasserstein distance. JSD: Jensen-Shannon distance. $\text{JSD}_\pi$: Ours. Averaged over 10 runs, with 95% CIs.

| # correct var. | $P_\alpha$ ($\uparrow$) | $R_\beta$ ($\uparrow$) | Inv. KL ($\uparrow$) | KS ($\uparrow$) | WD ($\downarrow$) | JSD ($\downarrow$) | $\text{JSD}_\pi$ ($\uparrow$) |
|---|---|---|---|---|---|---|---|
| 5 | $0.863 \pm 0.024$ | $0.456 \pm 0.017$ | $0.989 \pm 0.002$ | $0.965 \pm 0.003$ | $0.017 \pm 0.001$ | $0.004 \pm 0.000$ | $0.963 \pm 0.007$ |
| 3 | $0.868 \pm 0.022$ | $0.457 \pm 0.012$ | $0.981 \pm 0.006$ | $0.966 \pm 0.003$ | $0.018 \pm 0.001$ | $0.004 \pm 0.000$ | $0.942 \pm 0.006$ |
| 1 | $0.866 \pm 0.021$ | $0.453 \pm 0.013$ | $0.985 \pm 0.003$ | $0.965 \pm 0.003$ | $0.018 \pm 0.001$ | $0.004 \pm 0.000$ | $0.908 \pm 0.013$ |

best option. This contrasts with our $\text{JSD}_\pi$ metric, which correctly orders the three models, and selects $Q^3_{\mathbf{X},W,Y}$ as the best option to a statistically significant level.

To further elucidate the differences in rankings between existing and our metrics, both in this experiment and the outcome generation comparison in Section 5, we list each ranking and their Spearman's rank correlation coefficient with the oracle ranking in Tables 7 and 8. Assessment via our metrics is the only protocol that reproduces the oracle ranking across both experiments.

Table 7: Treatment assignment experiment: rankings by different metrics, sorted by Spearman's rank correlation coefficient ($r_s$) with oracle ranking. Numbering indicates the oracle order of $\pi_i(\mathbf{X})$.

| Metric | Ranking | $r_s$ ($\uparrow$) |
|---|---|---|
| $P_\alpha$ | 2,3,1 | -0.5 |
| KS | 2,1,3 | 0.5 |
| Inv. KL | 1,3,2 | 0.5 |
| $R_\beta$ | 2,1,3 | 0.5 |
| WD | 1,2,3 | 1 |
| $\text{JSD}_\pi$ | 1,2,3 | 1 |

Table 8: Outcome generation experiment: Rankings by different metrics, sorted by Spearman's rank correlation coefficient ($r_s$) with oracle ranking. $Q^i_{Y|W,\mathbf{X}}$ are number by oracle ranking, 1: T-Learner, 2: TARNet, 3: DragonNet, 4: S-Learner.

| Metric | Ranking | $r_s$ ($\uparrow$) |
|---|---|---|
| KS | 4,2,3,1 | -0.80 |
| Inv. KL | 4,2,1,3 | -0.40 |
| $P_\alpha$ | 1,4,3,2 | 0.20 |
| $R_\beta$ | 1,4,3,2 | 0.20 |
| WD | 2,3,1,4 | 0.40 |
| $U_{\text{PEHE}}$ | 1,2,3,4 | 1 |

## D.3 EXISTING METRIC FAILURE: EXTREME EXAMPLE

As a 'proof by contradiction' that current metrics can offer a good level on information on the preservation of (i)-(iii), we present some extreme examples. We show that joint-distribution-level metrics do not have enough resolution to identify how well (i)-(iii) are preserved, even if $Q$ comprehensively fails in modelling any one of the component distributions $P_{\mathbf{X}}$, $P_{W|\mathbf{X}}$, or $P_{Y|W,\mathbf{X}}$.

We perform a series of experiments where we evaluate adversarial synthetic versions of a simulated dataset, with each synthetic version failing in one of the above components, and we show that standard metrics do not identify these failure modes. We simulate real data using the DGP in `CATENets` from Curth & van der Schaar (2021) and we create three $\mathcal{D}_{\text{synth}}$ that perfectly model two component distributions of $\mathcal{D}_{\text{real}}$ but poorly approximate the remaining one. For poorly modelled $P_{\mathbf{X}}$, we set $\mathbf{X} = \mathbf{0}$; for poorly modelled $P_{W|\mathbf{X}}$, we assign all instances with $W = 0$; and for poorly modelled $P_{Y|W,\mathbf{X}}$, we draw $Y$ from a normal distribution with mean 0 regardless of treatment. All such $\mathcal{D}_{\text{synth}}$ are useless for treatment effect estimation.

Table 9: Scores on adversarially created $\mathcal{D}_{\text{synth}}$ which poorly perform on desiderata (i), (ii), or (iii).

|  | Inv. KL ($\uparrow$) | $P_\alpha$ ($\uparrow$) | $R_\beta$ ($\uparrow$) | MMD ($\downarrow$) |
|---|---|---|---|---|
| Poor (i) | 0.681 | 0.902 | 0.368 | 0.085 |
| Poor (ii) | 0.685 | 0.501 | 0.333 | 0.074 |
| Poor (iii) | 0.844 | 0.905 | 0.430 | 0.008 |

We report the inverse of KL divergence, $P_\alpha$, $R_\beta$ and MMD, which all have range $[0, 1]$, for these synthetic datasets in Table 9. We see that these conventional evaluation metrics do not accurately reflect the invalidity of each $\mathcal{D}_{\text{synth}}$ for treatment effect estimation. None report significantly low scores, despite the failure of each $\mathcal{D}_{\text{synth}}$. $R_\beta$ reflects these failures best, although its scores still do not adequately reflect how these datasets render correct treatment effect analysis impossible, and it does not allow a granular enough analysis to disentangle which component distribution is poorly modelled.

In further detail, for the experiments that examine poor modelling of $P_\mathbf{X}$ and $P_{W|\mathbf{X}}$, we simulate $\mathcal{D}_{\text{real}}$ of size $n = 1000$ with $d = 1$ covariate as follows:

$$X \sim \mathcal{N}(0, 1) \tag{9}$$
$$W \sim \text{Bernoulli}(0.5) \tag{10}$$
$$Y(0), Y(1) = 0 \tag{11}$$
$$Y = (1 - W)Y(0) + WY(1) + \epsilon, \ \epsilon \sim \mathcal{N}(0, 1) \tag{12}$$

We manufacture $\mathcal{D}_{\text{synth}}$ that exhibits poor modelling of $P_\mathbf{X}$ by generating $W$ and $Y$ from the true distributions as above, but set $\mathbf{X} = \mathbf{0}$. For $\mathcal{D}_{\text{synth}}$ that exhibits poor modelling of $P_{W|\mathbf{X}}$, we generate $\mathbf{X}$ and $Y$ from their true distributions, but set all $W = 0$.

To demonstrate assessment under poor modelling of $P_{Y|W,\mathbf{X}}$, we set the covariate in $\mathcal{D}_{\text{real}}$ to be predictive, such that it affects the value of the potential outcome $Y(1)$, but not $Y(0)$. The distributions remain the same as the above, although now the potential outcomes are:

$$Y(0) = 0 \tag{13}$$
$$Y(1) = X^2 \tag{14}$$

We manufacture $\mathcal{D}_{\text{synth}}$ that poorly models of $P_{Y|W,\mathbf{X}}$ by generating $\mathbf{X}$ and $W$ from their true distributions, but we set $Y(0), Y(1) = 0$.

# E    DISCUSSION ON ALTERNATIVE METRICS

While we propose a set of metrics $\mathcal{M}$ for evaluation of $\mathcal{D}_{\text{synth}}$, there are many possible alternatives to each choice we make. Our choices enable evaluation of how well $\mathcal{D}_{\text{synth}}$ adheres to our desiderata, but, like any metrics, they may be sub-optimal for certain data holders with specific preferences. Here, we list some alternative definitions, and we detail when they may be preferable. We would like to emphasise that conducting *any* reasonable assessment of the preservation of (i)-(iii) is beneficial compared to standard evaluation practices.

## E.1    ALTERNATIVE COVARIATE DISTRIBUTION ASSESSMENT

As we state in the main paper, comparison of $P_X$ and $Q_X$ is essentially a standard synthetic data evaluation problem, and therefore any standard protocol can be applied.

For example, if the dimensionality of $\mathbf{X}$ is small, manual evaluation via visualisation may be preferable to the precision/recall analysis we suggest, as this can provide a more granular and

interpretable assessment. On the other hand, if a single all-encompassing score is desired, rather than the two-dimensional metric $(P_{\alpha,\mathbf{X}}, R_{\beta,\mathbf{X}})$, then statistical divergence metrics can offer this. These one-dimensional metrics can lead to more straightforward model selection than $(P_{\alpha,\mathbf{X}}, R_{\beta,\mathbf{X}})$, as ordering based on a two-dimensional metric can be ambiguous.

### E.2 ALTERNATIVE TREATMENT ASSIGNMENT MECHANISM ASSESSMENT

Similarly, there is a vast array of metrics which could be substituted into (5) over Jensen-Shannon distance which could measure the difference between $P_{W|\mathbf{X}}$ and $Q_{W|\mathbf{X}}$. These include metrics such as KL divergence and Wasserstein distance, which are also very common in machine learning literature. For example, a data holder may prefer KL divergence if they want to more harshly punish $Q_{W|\mathbf{X}}$ for failing to place density where $P_{W|\mathbf{X}}$ is probable, encouraging *mode-covering* behaviour. On the other hand, if a data holder wants to more harshly punish $Q_{W|\mathbf{X}}$ for spreading mass away from the modes, Wasserstein distance may be preferable, leading to *mode-seeking* behaviour. JSD achieves a balance between these two focuses, but if a data holder has a strong preference for one or the other, these alternate choices would be preferable. Nevertheless, we suggest that, apart from extreme scenarios, most reasonable methods to assess the preservation of $P_{W|\mathbf{X}}$ will lead to similar analysis.

### E.3 ALTERNATIVE OUTCOME GENERATION MECHANISM METRICS

Raw similarity of PEHE in CATE estimation between $\mathcal{D}_{\text{real}}$ and $\mathcal{D}_{\text{synth}}$ may not be the most important quantity of interest for certain data holders. This can be particularly true in medical practice, as raw performance is not the only important aspect of a downstream model. We propose some alternatives which may be more applicable in the following situations:

1. Correct estimation of the *sign* of the CATE may be of heightened importance if the CATE learner is assisting with policy decisions. The wrong CATE sign will lead to incorrect policy administration, whereas the magnitude of the effect may not be as important for decision making.

2. Discovering the correct drivers of effect heterogeneity may be important, as how a learner arrives at its final estimation is particularly important to consider in applications such as in drug discovery or clinical practice (Hermansson & Svensson, 2021; Crabbé et al., 2022).

#### E.3.1 POLICY ASSIGNMENT

If policy guidance is of interest, then quantification of how well the sign of CATE estimates is preserved between $\mathcal{D}_{\text{synth}}$ and $\mathcal{D}_{\text{real}}$ may be desired, which can be done as follows:

$$U_{\text{policy}}(\mathcal{D}_{\text{real}}, \mathcal{D}_{\text{synth}}) = \frac{1}{|\mathcal{F}|} \sum_{\hat{\tau} \in \mathcal{F}} \mathbb{E}_{P_{\mathbf{X}}}[I(\hat{\tau}_{\text{synth}}(\mathbf{X}) \times \hat{\tau}_{\text{real}}(\mathbf{X}) > 0)] \tag{15}$$

where $I$ is the indicator function.

#### E.3.2 FEATURE IMPORTANCE

If assessing how well $\mathcal{D}_{\text{synth}}$ permits the discovery of the correct drivers of effect heterogeneity is important, this can be quantified through the use of feature importance methods. Given a CATE learner $\hat{\tau}$, feature importance methods offer a means to measure the sensitivity of the model to each covariate by assigning an importance score $a_i(\hat{\tau}, \mathbf{x})$ to each feature $x_i$ that reflects its importance in the prediction of the CATE $\hat{\tau}(\mathbf{x})$. There are many different instantiations of feature importance methods with different strengths (Ewald et al., 2024), and the metric we propose here is method-agnostic. We quantify how well $P_{Y|W,\mathbf{X}}$ is modelled according to feature importance similarity between $\mathcal{D}_{\text{synth}}$ and $\mathcal{D}_{\text{real}}$ as follows:

$$U_{\text{int}}(\mathcal{D}_{\text{real}}, \mathcal{D}_{\text{synth}}) = \frac{1}{|\mathcal{F}|} \sum_{\hat{\tau} \in \mathcal{F}} S_C(A_{\text{real},\hat{\tau}}, A_{\text{synth},\hat{\tau}}) \tag{16}$$

where $S_C$ is cosine similarity, and $A_{\text{real},\hat{\tau}}$ and $A_{\text{synth},\hat{\tau}}$ are $d$-dimensional vectors with $i^{th}$ entries

$$A^i_{\diamond,\hat{\tau}} = \mathbb{E}_{P_{\mathbf{X}}}[a_i(\hat{\tau}_\diamond, \mathbf{X})], \diamond \in \{\text{real}, \text{synth}\} \tag{17}$$

$U_{\text{int}} \in [-1, 1]$, where $U_{\text{int}} = 1$ indicates total agreement in the feature importances of $\mathcal{D}_{\text{real}}$ and $\mathcal{D}_{\text{synth}}$, while $U_{\text{int}} = 0$ indicates that the feature importances are uncorrelated, suggesting that $Q_{Y|W,\mathbf{X}}$ does not allow discovery of the correct drivers of heterogeneity. Finally, $U_{\text{int}} = -1$ indicates antithetical feature importances, suggesting a drastic failure of $Q_{Y|W,\mathbf{X}}$ in estimating $P_{Y|W,\mathbf{X}}$.

## F  DEFINING $\mathcal{F}$ FOR $U_{\text{PEHE}}$

In CATE estimation, model validation is a difficult task (Curth & Van Der Schaar, 2023). As such, it is reasonable to expect that a set of downstream analysts conducting CATE estimation on $\mathcal{D}_{\text{synth}}$ will use different learners. Therefore, we want $U_{\text{PEHE}}$ to reflect the expected difference in downstream performance between $\mathcal{D}_{\text{synth}}$ and $\mathcal{D}_{\text{real}}$ across a diverse array of potential learners, such that it is representative for the entire population of analysts, and has limited bias towards any particular learner class. To achieve this, we propose averaging $U_{\text{PEHE}}$ across a family of CATE learners $\mathcal{F}$, and we suggest that larger $|\mathcal{F}|$, and diverse selection of the learners within $\mathcal{F}$, is preferable.

Of course, there is a trade off between the size of $\mathcal{F}$, and therefore the stability of $U_{\text{PEHE}}$, and the computational cost of repeated CATE estimation. With this in mind, to limit the computation involved in calculating $U_{\text{PEHE}}$, we suggest that users should be selective of the learners included in $\mathcal{F}$ to maximize learner diversity, and minimise $|\mathcal{F}|$. For example, in our experiments we set $|\mathcal{F}| = 4$, and we chose learners from both of the high-level CATE learning strategies described in Curth & van der Schaar (2021) (i.e. one-step plug-in learners, and two-step learners). Specifically, for the one-step learners we use S- and T- learners (Künzel et al., 2019), and for the two-step learners we use RA and DR learners (Kennedy et al., 2020). All four of these learners conduct CATE estimation differently, and encode different inductive biases in their approaches, and thus they form a good diverse base for $\mathcal{F}$.

For our experiments, on each of the real datasets from Section 7.1, the runtime for calculating $U_{\text{PEHE}}$ for one run are shown in Table 10. Note that these are much less than the typical generation times for each dataset, so this step is unlikely to be a large time burden for the data holder. Also note that these calculations can be parallelized across the learner classes, which we did not do, and this can improve the computational feasibility of using a larger $|\mathcal{F}|$.

Table 10: Runtime to calculate $U_{\text{PEHE}}$

| Dataset | $U_{\text{PEHE}}$ runtime (s) |
|---------|------------------------------|
| ACTG    | 26                           |
| IHDP    | 60                           |
| ACIC    | 191                          |

## G  STEAM DIFFERENTIAL PRIVACY PROOF

The theoretical guarantee of STEAM's differential privacy (DP) when using individual DP components is grounded in the post-processing and composition theorems of DP (Dwork & Roth, 2014), as we state in the main body of the paper. We make this derivation clear here, by first outlining the post-processing and composition theorems in full.

**Theorem** (Post-Processing Theorem). *Let $M : \mathbb{N}^{|\mathcal{X}|} \to \mathcal{R}$ be a randomized algorithm that is $(\epsilon, \delta)$-differentially private. Let $f : \mathcal{R} \to \mathcal{R}'$ be an arbitrary randomized mapping. Then the composition $f \circ M : \mathbb{N}^{|\mathcal{X}|} \to \mathcal{R}'$ is $(\epsilon, \delta)$-differentially private.*

**Theorem** (Composition Theorem). *Let $M_i : \mathbb{N}^{|\mathcal{X}|} \to \mathcal{R}_i$ be an $(\epsilon_i, \delta_i)$-differentially private algorithm for $i \in [k]$. Define $M_{[k]} : \mathbb{N}^{|\mathcal{X}|} \to \prod_{i=1}^{k} \mathcal{R}_i$ as:*

$$M_{[k]}(x) = (M_1(x), M_2(x), \dots, M_k(x)),$$

then $M_{[k]}$ is $\left( \sum_{i=1}^{k} \epsilon_i, \sum_{i=1}^{k} \delta_i \right)$-differentially private.

Given these theorems, we have our guarantee of DP generation with STEAM. Specifically:

**Proposition 1.** *If $Q_{\mathbf{X}}$ satisfies $(\epsilon_{\mathbf{X}}, \delta_{\mathbf{X}})$-differential privacy, $Q_{W|\mathbf{X}}$ satisfies $(\epsilon_W, \delta_W)$-differential privacy, and $Q_{Y|W,\mathbf{X}}$ satisfies $(\epsilon_Y, \delta_Y)$-differential privacy, STEAM satisfies $(\epsilon_{total}, \delta_{total})$-differential privacy, where $\epsilon_{total} = \epsilon_{\mathbf{X}} + \epsilon_W + \epsilon_Y$, $\delta_{total} = \delta_{\mathbf{X}} + \delta_W + \delta_Y$.*

*Proof.* $Q_{\mathbf{X}}$ generates $\mathbf{X}$, and satisfies $(\epsilon_{\mathbf{X}}, \delta_{\mathbf{X}})$-differential privacy by assumption.

By the post-processing theorem, inputting $\mathbf{X}$ as the condition to $Q_{W|\mathbf{X}}$ does not affect its privacy. $Q_{W|\mathbf{X}}$ generates $W$, and satisfies $(\epsilon_W, \delta_W)$-differential privacy by assumption.

By the post-processing theorem, inputting $W$ and $\mathbf{X}$ as the conditions to $Q_{Y|W,\mathbf{X}}$ does not affect their privacy. $Q_{Y|W,\mathbf{X}}$ generates $Y$, and satisfies $(\epsilon_Y, \delta_Y)$-differential privacy by assumption.

STEAM generates $(\mathbf{X}, W, Y)$, and is the composition of $Q_{\mathbf{X}}$, $Q_{W|\mathbf{X}}$, and $Q_{Y|W,\mathbf{X}}$, i.e. STEAM = $(Q_{\mathbf{X}}, Q_{W|\mathbf{X}}, Q_{Y|W,\mathbf{X}})$

Therefore, by the composition theorem STEAM satisfies $(\epsilon_{\text{total}}, \delta_{\text{total}})$-differential privacy, where $\epsilon_{\text{total}} = \epsilon_{\mathbf{X}} + \epsilon_W + \epsilon_Y$, $\delta_{\text{total}} = \delta_{\mathbf{X}} + \delta_W + \delta_Y$. $\qquad\square$

# H MAIN EXPERIMENTAL DETAILS

Here we add any additional details to the experiment set-ups from Section 7. All experiments were run on an Azure VM with a 48-Core AMD EPYC Milan CPU, an A100 GPU with 80GB of VRAM, and 880GB of RAM. We report typical runtimes where relevant. An estimated total compute time for all experimental runs is ~72 hours. This does not include the compute required for preliminary experimentation.

For all generative models, we use the open source library `synthcity` Qian et al. (2023) (Apache-2.0 License), and we do not change the default hyperparameters. We set the treatment and outcome generators of STEAM as a logistic regression function from `scikit-learn` Pedregosa et al. (2011) and T-Learner from `CATENets` Curth & van der Schaar (2021), respectively.

## H.1 GENERATION OF MEDICAL DATA CONTAINING TREATMENTS

To assess sequential generation in a number of real-world scenarios, we evaluate performance on ACTG (Hammer et al., 1996) and on the popular treatment effect estimation datasets IHDP (Hill, 2011) and ACIC (Dorie et al., 2018). We also report further results in Table 11 on a non-medical dataset, Jobs (LaLonde, 1986), which is also popular amongst the treatment effect estimation community, to show that STEAM can be applied beyond the medical context, to any dataset containing treatments. More in depth descriptions of the datasets used are here:

1. **AIDS Clinical Trial Group (ACTG) study 175.** A clinical trial on subjects with HIV-1 (Hammer et al., 1996). Preprocessed as in Hatt et al. (2022) to compare CD4 counts at the beginning of the study and after $20 \pm 5$ weeks across treatment arms using zidovudine (ZDV) and zalcitabine (ZAL) vs. ZDV only. The ACTG dataset contains $n = 1056$ instances with $d = 12$ covariates and a continuous outcome, and we use the publicly available version from `https://github.com/tobhatt/CorNet`.

2. **Infant Health and Development Program (IHDP).** A semi-synthetic medical dataset, with real covariates and simulated outcomes, using data from a randomised experiment designed to evaluate the effect of specialist childcare on the cognitive test scores of premature infants (Brooks-Gunn et al., 1992). Confounding and treatment imbalance were introduced in Hill (2011) to mimic an observational dataset. The IHDP dataset consists of $n = 747$ instances with $d = 25$ covariates and a continuous outcome. We use the publicly available version from `https://github.com/AMLab-Amsterdam/CEVAE` (Louizos et al., 2017), with the first batch of simulated outcomes.

3. **Atlantic Causal Inference Competition 2016 (ACIC).** A semi-synthetic medical dataset, with real covariates and simulated outcomes, containing data from the Collaborative Perinatal

Project (Niswander, 1972). The data was modified in Dorie et al. (2018) to simulate an observational study examining the impact of birth weight in twins on IQ. The ACIC dataset consists of $n = 4802$ instances with $d = 58$ covariates and a continuous outcome. We use the publicly available version from the `causallib` package (Shimoni et al., 2019) (Apache-2.0 License) available here `https://github.com/BiomedSciAI/causallib`, using the first simulated set of treatments and potential outcomes.

4. **Jobs.** Jobs contains experimental data from a male sub-sample from the National Supported Work Demonstration from LaLonde (1986) to evaluate the effect of job training on income. The Jobs dataset consists of $n = 722$ instances with $d = 7$ covariates and a continuous outcome. We use the publicly available version used in Dehejia & Wahba (1998; 2002), from `https://users.nber.org/~rdehejia/data/.nswdata2.html`.

We report extended results for all models tested, and the further results on the Jobs dataset, in Table 11. For each model on each dataset we conduct 20 runs. A typical run for a given real dataset and generative model took 15 minutes.

### H.2 SIMULATED EXPERIMENTS

For our simulated insight experiments, we compare performance of a standard TabDDPM with STEAM$_{\text{TabDDPM}}$, and we report average results over 10 runs. A typical run took 5 minutes. For simulation of $\mathcal{D}_{\text{real}}$, we use the DGP from `CATENets` (Curth & van der Schaar, 2021).

### H.3 DIFFERENTIALLY PRIVATE GENERATION

For our experiment which showcases the performance of STEAM when satisfying DP, we compare the generative performance of baseline methods AIM (McKenna et al., 2022), GEM (Liu et al., 2021), MST (McKenna et al., 2021), RAP (Aydore et al., 2021) with their STEAM counterparts. We use the code provided by McKenna et al. (2022) in their GitHub `https://github.com/ryan112358/private-pgm` for the AIM and MST implementations, and we use the code provided in the GitHub `https://github.com/terranceliu/dp-query-release` for the GEM and RAP implementations. We use the default hyperparameter settings of these implementations, with the workload set as 3-way marginals. For the STEAM models, we use the relevant base model for $Q_{\mathbf{X}}$, DP random forest from the `diffprivlib` library (Holohan et al., 2019) (MIT License) for $Q_{W|\mathbf{X}}$, and a custom implementation of a T-Learner Künzel et al. (2019) based on Curth & van der Schaar (2021) which guarantees DP by training with DP stochastic gradient descent, implemented with the `Opacus` library (Yousefpour et al., 2021) (Apache-2.0 License). We report comparative results on varying $\epsilon$, averaged over 5 runs.

## I EXTENDED RESULTS

### I.1 SECTION 7.1 EXTENDED RESULTS

We report the full set of results for each model and dataset from Section 7.1 in Table 11. We pair each standard model with its STEAM analogue, and report the relative difference between them for each metric, where (green) indicates better performance by STEAM, and (red) indicates better performance by standard modelling. We see that STEAM clearly outperforms. Almost all STEAM models perform better in each metric than all standard models.

### I.2 SECTION 7.3 EXTENDED RESULTS

We report the full $(\epsilon, \delta)$-DP generation results on the ACTG across a set of baseline models with the same set-ups as in Section 7.3. In Figure 5 we compare GEM (Liu et al., 2021) with STEAM$_{\text{GEM}}$, in Figure 6 we compare MST (McKenna et al., 2021) with STEAM$_{\text{MST}}$, and in Figure 7 we compare RAP (Aydore et al., 2021) with STEAM$_{\text{RAP}}$. While there are some nuances to each baseline comparison,

Table 11: $P_{\alpha,\mathbf{x}}$, $R_{\beta,\mathbf{x}}$, $\text{JSD}_\pi$, and $U_{\text{PEHE}}$ values for STEAM and standard models. Averaged over 20 runs, with 95% confidence intervals. Each STEAM model is placed after its corresponding standard model. Coloured numbers in brackets indicate relative difference between standard and STEAM model, where (green) indicates better performance by STEAM, and (red) indicates better performance by standard modelling.

| Dataset | Model | $P_{\alpha,\mathbf{x}}$ ($\uparrow$) | $R_{\beta,\mathbf{x}}$ ($\uparrow$) | $\text{JSD}_\pi$ ($\uparrow$) | $U_{\text{PEHE}}$ ($\downarrow$) |
|---|---|---|---|---|---|
| ACTG | TVAE | $0.926 \pm 0.013$ | $0.483 \pm 0.010$ | $0.946 \pm 0.004$ | $0.564 \pm 0.017$ |
| | STEAM $_{\text{TVAE}}$ | $0.929 \pm 0.008$ (+0.003) | $0.486 \pm 0.009$ (+0.003) | $0.958 \pm 0.004$ (+0.012) | $0.492 \pm 0.011$ (-0.072) |
| | ARF | $0.818 \pm 0.012$ | $0.453 \pm 0.007$ | $0.960 \pm 0.004$ | $0.577 \pm 0.015$ |
| | STEAM $_{\text{ARF}}$ | $0.836 \pm 0.008$ (+0.018) | $0.464 \pm 0.007$ (+0.011) | $0.962 \pm 0.004$ (+0.002) | $0.423 \pm 0.016$ (-0.154) |
| | CTGAN | $0.889 \pm 0.020$ | $0.446 \pm 0.014$ | $0.934 \pm 0.008$ | $0.586 \pm 0.017$ |
| | STEAM $_{\text{CTGAN}}$ | $0.892 \pm 0.017$ (+0.003) | $0.435 \pm 0.012$ (-0.011) | $0.959 \pm 0.005$ (+0.025) | $0.436 \pm 0.012$ (-0.150) |
| | NFlow | $0.817 \pm 0.032$ | $0.418 \pm 0.008$ | $0.913 \pm 0.016$ | $0.643 \pm 0.026$ |
| | STEAM $_{\text{NFlow}}$ | $0.837 \pm 0.040$ (+0.020) | $0.417 \pm 0.015$ (-0.001) | $0.962 \pm 0.005$ (+0.049) | $0.445 \pm 0.020$ (-0.198) |
| | TabDDPM | $0.067 \pm 0.060$ | $0.036 \pm 0.035$ | $0.812 \pm 0.029$ | $1.761 \pm 0.230$ |
| | STEAM $_{\text{TabDDPM}}$ | $0.612 \pm 0.106$ (+0.545) | $0.310 \pm 0.055$ (+0.274) | $0.952 \pm 0.009$ (+0.140) | $0.468 \pm 0.013$ (-1.293) |
| IHDP | CTGAN | $0.663 \pm 0.018$ | $0.419 \pm 0.013$ | $0.888 \pm 0.010$ | $2.521 \pm 0.161$ |
| | STEAM $_{\text{CTGAN}}$ | $0.674 \pm 0.014$ (+0.011) | $0.424 \pm 0.011$ (+0.005) | $0.928 \pm 0.009$ (+0.040) | $1.709 \pm 0.052$ (-0.812) |
| | TabDDPM | $0.477 \pm 0.036$ | $0.340 \pm 0.022$ | $0.862 \pm 0.011$ | $2.706 \pm 0.138$ |
| | STEAM $_{\text{TabDDPM}}$ | $0.553 \pm 0.029$ (+0.076) | $0.396 \pm 0.015$ (+0.056) | $0.918 \pm 0.011$ (+0.056) | $2.346 \pm 0.088$ (-0.360) |
| | ARF | $0.528 \pm 0.009$ | $0.381 \pm 0.010$ | $0.921 \pm 0.009$ | $3.019 \pm 0.117$ |
| | STEAM $_{\text{ARF}}$ | $0.565 \pm 0.014$ (+0.037) | $0.394 \pm 0.010$ (+0.013) | $0.921 \pm 0.009$ (+0.000) | $1.629 \pm 0.056$ (-1.390) |
| | TVAE | $0.622 \pm 0.014$ | $0.410 \pm 0.010$ | $0.880 \pm 0.014$ | $3.198 \pm 0.172$ |
| | STEAM $_{\text{TVAE}}$ | $0.629 \pm 0.015$ (+0.007) | $0.412 \pm 0.011$ (+0.002) | $0.927 \pm 0.007$ (+0.047) | $2.100 \pm 0.075$ (-1.098) |
| | NFlow | $0.406 \pm 0.028$ | $0.309 \pm 0.012$ | $0.882 \pm 0.012$ | $3.835 \pm 0.345$ |
| | STEAM $_{\text{NFlow}}$ | $0.435 \pm 0.034$ (+0.029) | $0.333 \pm 0.020$ (+0.024) | $0.921 \pm 0.007$ (+0.039) | $2.177 \pm 0.118$ (-1.658) |
| ACIC | TVAE | $0.763 \pm 0.011$ | $0.515 \pm 0.006$ | $0.926 \pm 0.007$ | $4.202 \pm 0.134$ |
| | STEAM $_{\text{TVAE}}$ | $0.767 \pm 0.009$ (+0.004) | $0.514 \pm 0.004$ (-0.001) | $0.972 \pm 0.002$ (+0.046) | $2.013 \pm 0.112$ (-2.189) |
| | ARF | $0.936 \pm 0.003$ | $0.396 \pm 0.003$ | $0.948 \pm 0.002$ | $4.742 \pm 0.165$ |
| | STEAM $_{\text{ARF}}$ | $0.939 \pm 0.004$ (+0.003) | $0.393 \pm 0.004$ (-0.003) | $0.977 \pm 0.002$ (+0.029) | $2.176 \pm 0.141$ (-2.566) |
| | CTGAN | $0.880 \pm 0.016$ | $0.421 \pm 0.013$ | $0.942 \pm 0.005$ | $4.518 \pm 0.186$ |
| | STEAM $_{\text{CTGAN}}$ | $0.873 \pm 0.014$ (-0.007) | $0.424 \pm 0.014$ (+0.003) | $0.972 \pm 0.002$ (+0.030) | $2.268 \pm 0.154$ (-2.250) |
| | NFlow | $0.691 \pm 0.052$ | $0.298 \pm 0.014$ | $0.872 \pm 0.024$ | $5.222 \pm 0.332$ |
| | STEAM $_{\text{NFlow}}$ | $0.673 \pm 0.044$ (-0.018) | $0.285 \pm 0.019$ (-0.013) | $0.973 \pm 0.002$ (+0.101) | $2.790 \pm 0.337$ (-2.432) |
| | TabDDPM | $0.260 \pm 0.043$ | $0.001 \pm 0.000$ | $0.787 \pm 0.032$ | $10.104 \pm 1.205$ |
| | STEAM $_{\text{TabDDPM}}$ | $0.273 \pm 0.035$ (+0.013) | $0.001 \pm 0.000$ (+0.000) | $0.941 \pm 0.020$ (+0.154) | $6.178 \pm 0.619$ (-3.926) |
| Jobs | TabDDPM | $0.890 \pm 0.014$ | $0.477 \pm 0.011$ | $0.949 \pm 0.004$ | $3.335 \pm 0.516$ |
| | STEAM $_{\text{TabDDPM}}$ | $0.929 \pm 0.009$ (+0.039) | $0.493 \pm 0.008$ (+0.016) | $0.954 \pm 0.003$ (+0.005) | $1.446 \pm 0.052$ (-1.889) |
| | ARF | $0.832 \pm 0.010$ | $0.431 \pm 0.019$ | $0.964 \pm 0.004$ | $3.173 \pm 0.691$ |
| | STEAM $_{\text{ARF}}$ | $0.863 \pm 0.011$ (+0.031) | $0.481 \pm 0.016$ (+0.050) | $0.953 \pm 0.004$ (-0.011) | $2.280 \pm 0.381$ (-0.893) |
| | TVAE | $0.886 \pm 0.017$ | $0.288 \pm 0.009$ | $0.944 \pm 0.006$ | $4.471 \pm 0.336$ |
| | STEAM $_{\text{TVAE}}$ | $0.887 \pm 0.014$ (+0.001) | $0.300 \pm 0.012$ (+0.012) | $0.949 \pm 0.004$ (+0.005) | $1.540 \pm 0.167$ (-2.931) |
| | CTGAN | $0.830 \pm 0.049$ | $0.339 \pm 0.023$ | $0.925 \pm 0.033$ | $4.608 \pm 0.792$ |
| | STEAM $_{\text{CTGAN}}$ | $0.778 \pm 0.076$ (-0.052) | $0.298 \pm 0.030$ (-0.041) | $0.939 \pm 0.007$ (+0.014) | $1.846 \pm 0.270$ (-2.762) |
| | NFlow | $0.716 \pm 0.058$ | $0.374 \pm 0.017$ | $0.920 \pm 0.018$ | $5.445 \pm 0.883$ |
| | STEAM $_{\text{NFlow}}$ | $0.800 \pm 0.041$ (+0.084) | $0.375 \pm 0.017$ (+0.001) | $0.952 \pm 0.006$ (+0.032) | $2.666 \pm 0.200$ (-2.779) |

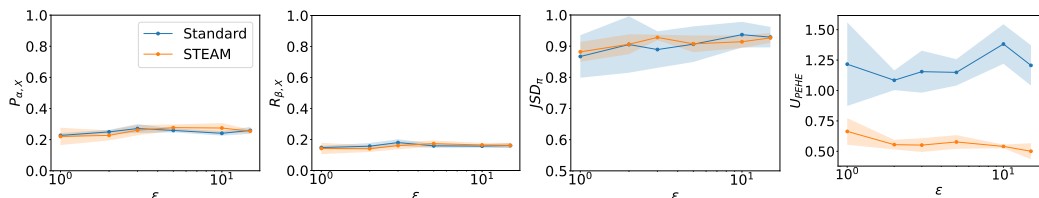

Figure 5: $P_{\alpha,\mathbf{X}}$ ($\uparrow$), $R_{\beta,\mathbf{X}}$ ($\uparrow$), $\mathrm{JSD}_\pi$ ($\uparrow$), and $U_{\mathrm{PEHE}}$ ($\downarrow$) evaluating STEAM$_{\mathrm{GEM}}$ and standard GEM across privacy budgets. Averaged over 5 runs, shaded area represents 95% CIs.

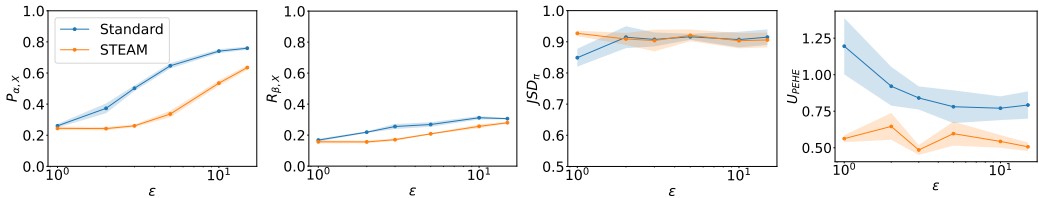

Figure 6: $P_{\alpha,\mathbf{X}}$ ($\uparrow$), $R_{\beta,\mathbf{X}}$ ($\uparrow$), $\mathrm{JSD}_\pi$ ($\uparrow$), and $U_{\mathrm{PEHE}}$ ($\downarrow$) evaluating STEAM$_{\mathrm{MST}}$ and standard MST across privacy budgets. Averaged over 5 runs, shaded area represents 95% CIs.

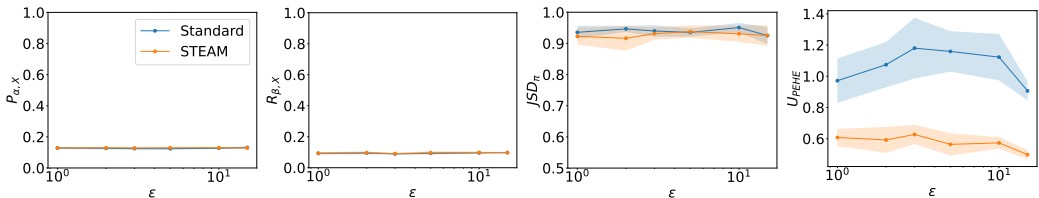

Figure 7: $P_{\alpha,\mathbf{X}}$ ($\uparrow$), $R_{\beta,\mathbf{X}}$ ($\uparrow$), $\mathrm{JSD}_\pi$ ($\uparrow$), and $U_{\mathrm{PEHE}}$ ($\downarrow$) evaluating STEAM$_{\mathrm{RAP}}$ and standard RAP across privacy budgets. Averaged over 5 runs, shaded area represents 95% CIs.

the general takeaway remains similar to those reported in Section 7.3 - STEAM models preserve $P_{W|\mathbf{X}}$ and $P_{Y|W,\mathbf{X}}$ better, while standard models preserve $P_{\mathbf{X}}$ better.

It is worth noting, however, that when baseline models perform poorly in modelling $P_{\mathbf{X}}$, as is the case for GEM and RAP, then the relevant STEAM model exhibits similar performance in this regard.

## J    ABLATIVE STUDY

To add to the evidence of STEAM's efficacy, we conduct an ablative study by assessing how jointly modelling $P_{\mathbf{X},W}$ affects performance. On the medical datasets used in the main body of the paper, we compare performance of the best standard models with their relevant ablation STEAM $_{\diamond,\,\text{joint X,W}}$, which models $P_{\mathbf{X},W}$ with the generative model and $P_{Y|W,\mathbf{X}}$ with a PO estimator, and regular STEAM. We report the results in Table 12.

We see that the ablative model, while often improving upon standard generation, is not as effective as STEAM. Directly modelling $P_{W|\mathbf{X}}$, as STEAM does, better preserves the treatment assignment and outcome generation mechanisms, and both JSD$_\pi$ and $U_{\text{PEHE}}$ are significantly improved by STEAM in most cases. Using the full inductive bias of directly modelling each distribution of our desiderata, and following the true DGP of data containing treatments is the best approach to generation.

Table 12: $P_{\alpha,\mathbf{X}}, R_{\beta,\mathbf{X}}, \text{JSD}_\pi$, and $U_{\text{PEHE}}$ values on standard, ablation, and STEAM models. Averaged over 10 runs, with 95% CIs.

| Dataset | Model | $P_{\alpha,\mathbf{X}}$ (↑) | $R_{\beta,\mathbf{X}}$ (↑) | JSD$_\pi$ (↑) | $U_{\text{PEHE}}$ (↓) |
|---|---|---|---|---|---|
| ACTG | TVAE | $0.926 \pm 0.013$ | $0.483 \pm 0.010$ | $0.946 \pm 0.004$ | $0.564 \pm 0.017$ |
| | STEAM $_{\text{TVAE, joint X,W}}$ (*ablation*) | $0.918 \pm 0.021$ | $0.473 \pm 0.012$ | $0.939 \pm 0.010$ | $0.475 \pm 0.012$ |
| | STEAM $_{\text{TVAE}}$ | $0.929 \pm 0.008$ | $0.486 \pm 0.009$ | $0.958 \pm 0.004$ | $0.492 \pm 0.011$ |
| IHDP | CTGAN | $0.663 \pm 0.018$ | $0.419 \pm 0.013$ | $0.888 \pm 0.010$ | $2.521 \pm 0.161$ |
| | STEAM $_{\text{CTGAN, joint X,W}}$ (*ablation*) | $0.639 \pm 0.021$ | $0.428 \pm 0.009$ | $0.908 \pm 0.019$ | $2.140 \pm 0.134$ |
| | STEAM $_{\text{CTGAN}}$ | $0.674 \pm 0.014$ | $0.424 \pm 0.011$ | $0.928 \pm 0.009$ | $1.709 \pm 0.052$ |
| ACIC | TVAE | $0.763 \pm 0.011$ | $0.515 \pm 0.006$ | $0.926 \pm 0.007$ | $4.202 \pm 0.134$ |
| | STEAM $_{\text{TVAE, joint X,W}}$ (*ablation*) | $0.747 \pm 0.023$ | $0.506 \pm 0.005$ | $0.920 \pm 0.009$s | $2.530 \pm 0.187$ |
| | STEAM $_{\text{TVAE}}$ | $0.767 \pm 0.009$ | $0.514 \pm 0.004$ | $0.972 \pm 0.002$ | $2.013 \pm 0.112$ |

## K    HYPERPARAMETER STABILITY

Generative modelling performnce is typically sensitive to hyperparameters. To assess the stability of STEAM's performance across hyperparameters, on IHDP we compare the performance of CTGAN with STEAM$_{\text{CTGAN}}$ with multiple hyperparameter configurations. We report results by changing three hyperparameters: number of hidden units within the generator layers (`generator_n_hidden_units`) (Table 13), number of hidden layers within the generator (`generator_n_hidden_layers`) (Table 14), and activation functions used in the generator (`generator_nonlin`) (Table 15), keeping all other hyperparameters default.

The performance gap between STEAM$_{\text{CTGAN}}$ and CTGAN is relatively stable across these configurations. STEAM$_{\text{CTGAN}}$ outperforms CTGAN in each metric at almost all hyperparameter levels. The most statistically significant differences are consistently noted in the JSD$_\pi$ and $U_{\text{PEHE}}$ metrics, which is compatible with the results displayed in the main paper.

Table 13: Comparison of STEAM with standard generation on IHDP at different `generator_n_hidden_units` levels. Averaged over 5 runs, with 95% CIs.

| `generator_n_hidden_units` | Model | $P_{\alpha,\mathbf{X}}$ (↑) | $R_{\beta,\mathbf{X}}$ (↑) | JSD$_\pi$ (↑) | $U_{\text{PEHE}}$ (↓) |
|---|---|---|---|---|---|
| 5 | CTGAN | $0.517 \pm 0.026$ | $0.396 \pm 0.015$ | $0.863 \pm 0.033$ | $2.914 \pm 0.047$ |
| | STEAM $_{\text{CTGAN}}$ | $0.565 \pm 0.011$ | $0.405 \pm 0.011$ | $0.941 \pm 0.000$ | $2.194 \pm 0.265$ |
| 50 | CTGAN | $0.622 \pm 0.028$ | $0.411 \pm 0.043$ | $0.916 \pm 0.15$ | $2.282 \pm 0.141$ |
| | STEAM $_{\text{CTGAN}}$ | $0.664 \pm 0.020$ | $0.444 \pm 0.017$ | $0.905 \pm 0.041$ | $1.960 \pm 0.174$ |
| 100 | CTGAN | $0.607 \pm 0.038$ | $0.418 \pm 0.032$ | $0.894 \pm 0.010$ | $2.560 \pm 0.289$ |
| | STEAM $_{\text{CTGAN}}$ | $0.682 \pm 0.016$ | $0.439 \pm 0.018$ | $0.912 \pm 0.004$ | $2.097 \pm 0.095$ |
| 300 | CTGAN | $0.619 \pm 0.030$ | $0.434 \pm 0.030$ | $0.908 \pm 0.023$ | $2.426 \pm 0.289$ |
| | STEAM $_{\text{CTGAN}}$ | $0.699 \pm 0.018$ | $0.458 \pm 0.015$ | $0.928 \pm 0.016$ | $2.028 \pm 0.163$ |
| 500 | CTGAN | $0.663 \pm 0.018$ | $0.419 \pm 0.013$ | $0.888 \pm 0.010$ | $2.521 \pm 0.161$ |
| | STEAM $_{\text{CTGAN}}$ | $0.674 \pm 0.014$ | $0.424 \pm 0.011$ | $0.928 \pm 0.009$ | $1.709 \pm 0.052$ |

Table 14: Comparison of STEAM with standard generation on IHDP at different `generator_n_hidden_layers` levels. Averaged over 5 runs, with 95% CIs.

| generator_n_hidden_layers | Model | $P_{\alpha,X}$ ($\uparrow$) | $R_{\beta,X}$ ($\uparrow$) | $JSD_\pi$ ($\uparrow$) | $U_{PEHE}$ ($\downarrow$) |
|---|---|---|---|---|---|
| 2 | CTGAN | $0.663 \pm 0.018$ | $0.419 \pm 0.013$ | $0.888 \pm 0.010$ | $2.521 \pm 0.161$ |
|  | STEAM $_{CTGAN}$ | $0.674 \pm 0.014$ | $0.424 \pm 0.011$ | $0.928 \pm 0.009$ | $1.709 \pm 0.052$ |
| 3 | CTGAN | $0.595 \pm 0.067$ | $0.395 \pm 0.066$ | $0.868 \pm 0.064$ | $2.982 \pm 0.647$ |
|  | STEAM $_{CTGAN}$ | $0.693 \pm 0.075$ | $0.441 \pm 0.043$ | $0.924 \pm 0.018$ | $2.028 \pm 0.143$ |
| 4 | CTGAN | $0.583 \pm 0.049$ | $0.259 \pm 0.074$ | $0.807 \pm 0.054$ | $3.278 \pm 0.191$ |
|  | STEAM $_{CTGAN}$ | $0.596 \pm 0.220$ | $0.301 \pm 0.084$ | $0.886 \pm 0.014$ | $2.690 \pm 0.836$ |
| 5 | CTGAN | $0.490 \pm 0.092$ | $0.313 \pm 0.069$ | $0.770 \pm 0.127$ | $2.871 \pm 0.599$ |
|  | STEAM $_{CTGAN}$ | $0.691 \pm 0.071$ | $0.386 \pm 0.041$ | $0.915 \pm 0.010$ | $2.498 \pm 0.536$ |

Table 15: Comparison of STEAM with standard generation on IHDP at different `generator_nonlin` settings. Averaged over 5 runs, with 95% CIs.

| generator_nonlin | Model | $P_{\alpha,X}$ ($\uparrow$) | $R_{\beta,X}$ ($\uparrow$) | $JSD_\pi$ ($\uparrow$) | $U_{PEHE}$ ($\downarrow$) |
|---|---|---|---|---|---|
| ReLU | CTGAN | $0.663 \pm 0.018$ | $0.419 \pm 0.013$ | $0.888 \pm 0.010$ | $2.521 \pm 0.161$ |
|  | STEAM $_{CTGAN}$ | $0.674 \pm 0.014$ | $0.424 \pm 0.011$ | $0.928 \pm 0.009$ | $1.709 \pm 0.052$ |
| SELU | CTGAN | $0.604 \pm 0.020$ | $0.419 \pm 0.015$ | $0.855 \pm 0.023$ | $2.509 \pm 0.160$ |
|  | STEAM $_{CTGAN}$ | $0.699 \pm 0.017$ | $0.445 \pm 0.025$ | $0.929 \pm 0.014$ | $2.043 \pm 0.130$ |
| Leaky ReLU | CTGAN | $0.648 \pm 0.045$ | $0.415 \pm 0.015$ | $0.889 \pm 0.016$ | $2.482 \pm 0.210$ |
|  | STEAM $_{CTGAN}$ | $0.699 \pm 0.028$ | $0.457 \pm 0.019$ | $0.916 \pm 0.011$ | $2.036 \pm 0.135$ |

## L CONGENIALITY BIAS

Congeniality bias (Curth & Van Der Schaar, 2023) is a phenomenon which may arise from generation with STEAM. In this scenario it refers to the fact that downstream models which are structurally similar to the outcome generator, $Q_{Y|W,\mathbf{X}}$, may be advantaged in their performance on $\mathcal{D}_{synth}$. For example, if the POs from an S-learner are used for $Q_{Y|W,\mathbf{X}}$, the outcome generation mechanism in $\mathcal{D}_{synth}$ may be modelled in such a way that it allows downstream S-learners to better estimate CATEs than other learners. While we acknowledge this phenomenon may disadvantage certain downstream models, we note that our outcome error metric, $U_{PEHE}$, averages across a number of downstream learner types, such that conducting generative model selection with $U_{PEHE}$ should lead to good performance across a wide variety of downstream learners, not just those similar to $Q_{Y|W,\mathbf{X}}$, helping to reduce this congeniality bias.

## M ALLOCATION OF THE PRIVACY BUDGET IN STEAM

In STEAM, uniform distribution of the privacy budget $\epsilon$ amongst the three component models ensures $(\epsilon, \delta)$-DP. However, such allocation is uninformed on the difficulty of modelling of $P_{\mathbf{X}}$, $P_{W|\mathbf{X}}$, and $P_{Y|W,\mathbf{X}}$, and their relative importance to downstream analysts.

In relation to the importance of each distribution, one immediate improvement can be to distribute $\epsilon$ according to some preference function $f : (0, \infty) \times \triangle^2 \to \epsilon \cdot \triangle^2$ (where $\triangle^2$ is the 2-simplex) which takes input of the budget $\epsilon$ and weights $\mathbf{w}$ for the relative importance of good modelling in $Q_{\mathbf{X}}$, $Q_{W|\mathbf{X}}$, and $Q_{Y|W,\mathbf{X}}$, and outputs a corresponding $\epsilon$ distribution. For example, a simple preference function definition would be $f(\epsilon, \mathbf{w}) = \epsilon \cdot \mathbf{w}$ where $\mathbf{w}$ could be defined by a data holder with some prior knowledge of the importance level of each component distribution to downstream analysts. Another approach, if it is not necessary to specify the desired $\epsilon$ distribution *a priori*, is to treat it as a hyperparameter, to be tuned over a series of runs to optimize some metric, such as a combination of $P_{\alpha,X}$, $R_{\beta,X}$, $JSD_\pi$, and $U_{PEHE}$.

Incorporating knowledge of the complexity of modelling $P_{\mathbf{X}}$, $P_{W|\mathbf{X}}$, and $P_{Y|W,\mathbf{X}}$ is more difficult. While some proxy measures could be established, such as the number of covariates in $\mathbf{X}$ indicating the

complexity of $P_{W|\mathbf{X}}$, establishing a robust understanding of how the complexity of these distributions relate and compare, is highly non-trivial, and as such we leave this for future work.

## N    GENERATION UNDER COVARIATE SHIFT

Generative model performance degrades under *covariate shift*. Covariate shift is a phenomenon which arises in data containing treatments, as the covariates of treated and untreated patients tend to differ, i.e. treated and untreated covariates tend to be drawn from different distributions. As the difference between these distributions grows, so too does covariate shift. We demonstrate degradation under high covariate shift through a simple experiment, where we create $\mathcal{D}_{\text{real}}$ with $d = 50$ covariates that are drawn from the following mixture model:

$$\mathbf{X} \sim \frac{9}{10}\mathcal{N}(\mu, I) + \frac{1}{10}\mathcal{N}(-\mu, I) \tag{18}$$

We set the mixture to have uneven weights between the two distributions to more closely simulate a real-world scenario, as treated instances are typically outnumbered in datasets, such as in IHDP and Jobs, which have under 20% treated. We model $\mathcal{D}_{\text{synth}}$ on $\mathcal{D}_{\text{real}}$ with varied $\mu$ using a TabDDPM model, and we see in Figure 8 that the metrics $P_{\alpha}$ and $R_{\beta}$ decrease as $\mu$, and therefore the covariate shift, increases.

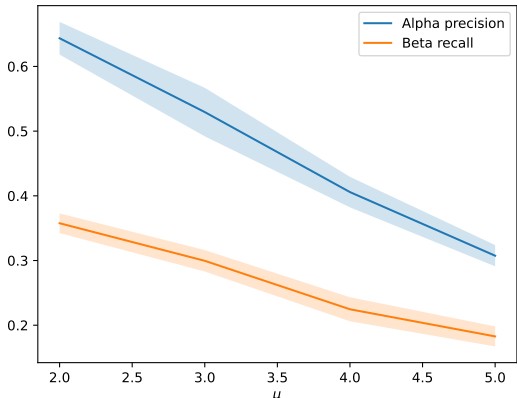

Figure 8: $P_{\alpha}$ and $R_{\beta}$ for $\mathcal{D}_{\text{synth}}$ as covariate shift within $\mathcal{D}_{\text{real}}$ increases.

Since this effect is observed in standard generative models, it will also be observed in STEAM, which uses a generative model for $Q_{\mathbf{X}}$. Performance degradation under this setting is unideal, since some degree of covariate shift is likely to arise in observational data, and therefore future works to remedy this are important.

A simple idea we explored was to use separate models to generate the covariates of each treatment group. We found, however, that the reduction in sample size that each generator received, as a result of the data splitting, negated any gains made by removing covariate shift, except in extreme shift scenarios. This was especially true for treated instances, due to their typical under-representation in data. There are many potential avenues that stem from this approach. Generation with fine-tuning (Hinton et al., 2012) on each treatment group is one possible direction. Another possibility is designing custom generative models that approach generation similarly to plug-in CATE learners, by scaling the degree to which a representation space is shared between treated and untreated instances. Given the non-trivial nature of this issue, we leave such explorations to future work.

## O    CAUSAL GENERATIVE MODEL COMPARISON

Causal generative models, which estimate the underlying structural causal model of a dataset, are a related family of generative models. We discuss the differences in positioning and assumptions

of our work compared to causal generative models in our extended related works, and here we produce empirical results for these model on the ACTG, IHDP, and ACIC datasets from Section 7.1. For baseline models, we consider two causal generative models: the additive noise model (ANM) (Hoyer et al., 2008) implementation in the `DoWhy-GCM` python package (Blöbaum et al., 2024), and a diffusion-based causal model (DCM) from Chao et al. (2024). We use the code provided by Chao et al. (2024) in their GitHub `https://github.com/patrickrchao/DiffusionBasedCausalModels` for the baseline implementations, and we use the same hyperparameter settings for both ANM and DCM as in that work.

However, in order to fairly compare these models with STEAM, we must first reconcile the differences in assumptions made, as discussed in Appendix B. For ACTG, IHDP, and ACIC, we do not have knowledge of the true causal graph, we simply know which features are the treatment and outcome. As such, we must first construct some reasonable causal graph using this knowledge to supply to these methods. We do so with three methods:

1. Construction of a naive graph $\mathcal{G}_{\text{naive}}$, in which each covariate causes $W$ and $Y$, $W$ causes $Y$, and every pair of covariates has a causal relationship between them;

2. Using the constraint-based PC causal discovery algorithm (Spirtes et al., 2001) to discover an estimated graph $\mathcal{G}_{\text{discovered}}$ from the Markov Equivalence Class for the true causal graph and;

3. Pruning the discovered causal graph $\mathcal{G}_{\text{discovered}}$ by removing any edges which contradict the DGP we assume. As such, any edges from $Y$ to $W$ or $\mathbf{X}$, or from $W$ to $\mathbf{X}$ are removed to form $\mathcal{G}_{\text{pruned}}$.

In Table 16 we report the results for ANM and DCM with each of these graph discovery methods, and we compare to the best performing STEAM models from Appendix I. For each dataset, we see that the relevant STEAM model outperforms all instantiations of the causal generative models in almost every metric, only being outperformed to a statistically significant level in $R_{\beta,X}$ on the ACIC dataset. These results validate that, when the true causal graph is not known, our less restrictive assumptions enable more useful generation of synthetic data containing treatments. We also see that the differences between the graph discovery methods are relatively small.

## P  FUTURE WORK: EXTENSIONS TO OTHER DGPS

STEAM places minimal assumptions upon the causal graph of $\mathcal{D}_{\text{real}}$. We see that our assumed DGP involving $P_{\mathbf{X}}$, $P_{W|\mathbf{X}}$, and $P_{Y|W,\mathbf{X}}$ is applicable across a very wide range of scenarios for datasets containing treatments, particularly in medical settings, as we describe in Section 6. If this DGP is known to not hold for $\mathcal{D}_{\text{real}}$, then it would be unadvisable to apply STEAM in its current form, as its inductive biases may not be helpful. However, if a similar setting arises where the overarching DGP is known, but the specific causal graph is not, then altering the generation order between $\mathbf{X}$, $W$, and $Y$ to mimic this alternative DGP, with a similar underlying motivation as in STEAM - that generation mimicking the real DGP is preferable to joint-level generation - can be done.

An example of a more complicated setting, where STEAM is not immediately applicable is when dealing with longitudinal data, which has time-varying covariates, treatments, and outcomes. In this setting, our assumed DGP would not hold in general, as features at a specific time point will likely have temporal causal relationships with earlier features, e.g. at time $t$, $\mathbf{X}_t$ could be affected by $W_{t-1}$, which is not modelled in our assumed DGP. Potential alterations to STEAM could adjust for this time-varying DGP, where one could consider a 'base' STEAM model, as described in this work, operating at each time step, with additional inputs from the immediately preceding time step to model temporal causal relationships.

Table 16: $P_{\alpha,\mathbf{X}}$, $R_{\beta,\mathbf{X}}$, $\text{JSD}_\pi$, and $U_{\text{PEHE}}$ values for the best performing STEAM models from Table I in comparison to causal generative models. Averaged over 20 runs, with 95% confidence intervals. Bold indicates the statistically significant best performing model.

| Dataset | Model | $P_{\alpha,\mathbf{X}}$ ($\uparrow$) | $R_{\beta,\mathbf{X}}$ ($\uparrow$) | $\text{JSD}_\pi$ ($\uparrow$) | $U_{\text{PEHE}}$ ($\downarrow$) |
|---|---|---|---|---|---|
| ACTG | STEAM $_{\text{TVAE}}$ | $\mathbf{0.929 \pm 0.008}$ | $\mathbf{0.486 \pm 0.009}$ | $0.958 \pm 0.004$ | $\mathbf{0.492 \pm 0.011}$ |
|  | DCM $\mathcal{G}_{\text{naive}}$ | $0.773 \pm 0.013$ | $0.369 \pm 0.006$ | $0.937 \pm 0.006$ | $0.665 \pm 0.034$ |
|  | DCM $\mathcal{G}_{\text{discovered}}$ | $0.756 \pm 0.011$ | $0.350 \pm 0.007$ | $0.956 \pm 0.005$ | $0.605 \pm 0.023$ |
|  | DCM $\mathcal{G}_{\text{pruned}}$ | $0.758 \pm 0.013$ | $0.358 \pm 0.007$ | $0.957 \pm 0.003$ | $0.596 \pm 0.017$ |
|  | ANM $\mathcal{G}_{\text{naive}}$ | $0.787 \pm 0.007$ | $0.389 \pm 0.008$ | $0.954 \pm 0.005$ | $0.580 \pm 0.017$ |
|  | ANM $\mathcal{G}_{\text{discovered}}$ | $0.836 \pm 0.007$ | $0.419 \pm 0.007$ | $0.952 \pm 0.004$ | $0.578 \pm 0.019$ |
|  | ANM $\mathcal{G}_{\text{pruned}}$ | $0.839 \pm 0.008$ | $0.412 \pm 0.005$ | $0.952 \pm 0.005$ | $0.582 \pm 0.014$ |
| IHDP | STEAM $_{\text{CTGAN}}$ | $0.674 \pm 0.014$ | $\mathbf{0.424 \pm 0.011}$ | $\mathbf{0.928 \pm 0.009}$ | $\mathbf{1.709 \pm 0.052}$ |
|  | DCM $\mathcal{G}_{\text{naive}}$ | $0.557 \pm 0.010$ | $0.340 \pm 0.009$ | $0.883 \pm 0.016$ | $4.878 \pm 0.395$ |
|  | DCM $\mathcal{G}_{\text{discovered}}$ | $0.658 \pm 0.011$ | $0.360 \pm 0.007$ | $0.893 \pm 0.008$ | $2.059 \pm 0.140$ |
|  | DCM $\mathcal{G}_{\text{pruned}}{}^{*}$ | $0.658 \pm 0.011$ | $0.360 \pm 0.007$ | $0.893 \pm 0.008$ | $2.059 \pm 0.140$ |
|  | ANM $\mathcal{G}_{\text{naive}}$ | $0.597 \pm 0.029$ | $0.379 \pm 0.011$ | $0.900 \pm 0.005$ | $1.868 \pm 0.147$ |
|  | ANM $\mathcal{G}_{\text{discovered}}$ | $0.589 \pm 0.012$ | $0.359 \pm 0.009$ | $0.892 \pm 0.008$ | $1.865 \pm 0.059$ |
|  | ANM $\mathcal{G}_{\text{pruned}}{}^{*}$ | $0.589 \pm 0.012$ | $0.359 \pm 0.009$ | $0.892 \pm 0.008$ | $1.865 \pm 0.059$ |
| ACIC$^{\dagger}$ | STEAM $_{\text{ARF}}$ | $0.939 \pm 0.004$ | $0.393 \pm 0.004$ | $\mathbf{0.977 \pm 0.002}$ | $\mathbf{2.176 \pm 0.141}$ |
|  | DCM $\mathcal{G}_{\text{discovered}}$ | $0.942 \pm 0.004$ | $0.422 \pm 0.003$ | $0.957 \pm 0.003$ | $4.249 \pm 0.132$ |
|  | DCM $\mathcal{G}_{\text{pruned}}$ | $0.939 \pm 0.004$ | $0.420 \pm 0.004$ | $0.959 \pm 0.002$ | $4.340 \pm 0.159$ |
|  | ANM $\mathcal{G}_{\text{discovered}}$ | $0.929 \pm 0.003$ | $0.404 \pm 0.003$ | $0.872 \pm 0.002$ | $4.193 \pm 0.127$ |
|  | ANM $\mathcal{G}_{\text{pruned}}$ | $0.930 \pm 0.004$ | $0.404 \pm 0.003$ | $0.880 \pm 0.002$ | $4.481 \pm 0.174$ |

$^{*}$ $\mathcal{G}_{\text{pruned}}$ is the same as $\mathcal{G}_{\text{discovered}}$ for IHDP

$^{\dagger}$ Excessive runtime caused the exclusion of $\mathcal{G}_{\text{naive}}$ ACIC results

## Q STEAM DIAGRAM

See the below for a pictoral representation of the DGPs generic synthetic data generation methods, real datasets containing treatments, and STEAM. STEAM is designed to closely mimic the real DGP.

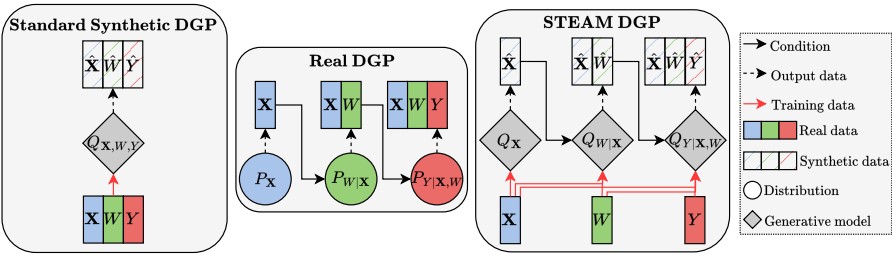

Figure 9: DGPs for generic generative models (left), real datasets (middle), and STEAM (right).

