# OpenReview forum: "Generation and Evaluation of Synthetic Data Containing Treatments"
_ICLR.cc/2025/Conference — Submitted to ICLR 2025_

### Official Review · Reviewer_uQ2Q · 2024-11-03

**Soundness:** 1
**Presentation:** 2
**Contribution:** 1
**Rating:** 3
**Confidence:** 5

**Summary:**

This paper introduces some metrics for synthetic data containing treatments (with the motivating downstream use case of causal inference on medical data). The authors then introduce STEAM, a synthetic data generation framework somewhat designed to optimize for these desiderata. They note that this framework admits DP components. Using the metrics they propose, they evaluate STEAM’s performance on some causal medical data scenarios.

**Strengths:**

S1: The authors have identified an interesting problem, in that it does seem like there could be improvements to how we think about evaluating synthetic data for medical causal inference. The metrics they present are interesting and appear to be novel.

S2: I appreciate the authors attempts to format the paper in a way that is readable and to highlight results requiring attention in different color blocks.

**Weaknesses:**

W1: This paper only considers DP-GAN as far as I can tell, for evaluations of DP synthetic data generation. This ignores a rich vein of work over the past decade, listing extensively here for the authors convenience as they improve upon their paper: PrivBayes [Zhang et al., SIGMOD 2017], HDMM+PGM, MWEM+PGM [McKenna et al., ICML 2019], PATECTGAN [Rosenblatt et al. 2020], FEM/GEM [Vietri et al., ICML 2020, Liu et al., NIPS 2021], RAP [Aydore et al., PMLR 2021, NIPS 2022], MST [McKenna et al., JPC 2021], AIM [McKenna et al., PVLDB 2022], Private Genetic Synthetic Data [Liu et al., ICML 2023]. Much of this work has been shown to outperform DP-GAN, and likely better encodes causal relationships in the data.

W2: Noting that DP composition is possible in the STEAM framework, and performing a naive budget allocation, does not really require a proof. Overall, the DP treatment of the framework is severely limited, as to not really be a contribution at all beyond observing that one can use existing DP methods blackbox, in conjunction. Formal reasoning about budget allocation, potential privacy gains, etc. is lacking. Under this view, it's hard to see STEAM as a sufficiently interesting or extensive contribution as an algorithmic framework.

W3: There is only an evaluation of $U_{PEHE}$ vs. the oracle metric in Table 4. What about $JSD_{\pi}$? It’s also not clear how this argument is structured - is it that the metrics in Table 3 can’t identify T-Learner as producing the best dataset? It’s true that in Table 3 this holds out (at least to some extent, although I’d say if I had to guess based on the results in Table 3 I’d select T-learner), but it’s not obvious to me why I would assume these results would generalize past this specific DGP. I think there should be some formal distinguishable results to clearly separate the quantity these metrics provide from prior quantities. Just setting up the DGP formerly and reasoning in closed form about what would distinguish things would be a step in the right direction.

**Questions:**

Q1: The authors use words like “standard” often, but its not clear how they’ve established that these methods are standard, particularly w.r.t. model selection for their empirical results. How have you determined what is “standard” throughout this paper? When there is not a clear standard, you should evaluate multiple methods that have been shown to perform well, which they fail to do (see W1).

Q2: Some of the methods for generating synthetic DP data listed in W1 admit general query workloads. W.r.t. Table 1 in your paper, these queries are often marginal and thus don’t necessarily differentiate between $\mathbf{X}$, $W$ and $Y$ in the way you desire. However, its not clear to me why you couldn’t use the different distributional measurements you frame in Eq. 2 3 and 4 with these methods. Can you explore this? Particularly with methods like GEM and AIM, which are arguably state-of-the-art for DP data generation.

Q3: The DGP given in 5.1 is simple enough as to admit formal, closed-form results distinguishing the different metric quantities that the authors are interested in. Can this be worked through? For example, under this DGP, we can bound the $JSD_{\pi}$ quantity as stated, and make a direct comparison with some of the other metrics in Table 3

---

> ### Author Response · Authors · 2024-11-22
> **Author response to uQ2Q (Part 1/2)**
>
> Thank you for your thoughtful comments and suggestions. We give answers to each of the following in turn, as well as pointing out corresponding updates to the revised manuscript:
>
> - (A) How we determined ‘standard’ baselines
> - (B) Comparison with more competitive differentially private generative models
> - (C) Formal motivation for new metrics
> - (D) Minor changes
>
> ---
>
> As a preface, kindly allow us to emphasise that the main contribution of our work **is not a new method for incorporating differential privacy (DP)** into synthetic data generation. Rather, we **focus on improving the potential of synthetic data for downstream causal inference tasks**, by proposing both useful metrics for this use case, and a framework for useful generation that does not make the restrictive assumption that we have full knowledge of the causal graph. In doing so, **we mainly focus on the non-DP setting** (hence the majority of our comparisons are against non-DP tabular data generators as in Section 7.1 and 7.2), in which there is extensive literature. Nevertheless, **we do believe that private generation is an important direction** to follow to improve the real-world applications of synthetic data, and so we highlight that STEAM is immediately compatible with DP in Section 6.2, and demonstrate good performance in DP generation in Section 7.3. We agree, however, that with respect to DP generation, our original paper was lacking robust comparisons, and we thank you for particularly helping us improve this.
>
> ---
>
> **(A) How we determined ‘standard’ baselines**
>
> In our empirical analysis discussed in Section 7.1, we compare STEAM models with a broad array of tabular generative models. While we only include the best performing models in the main body of the paper, Table 11 in Appendix I displays the full set of results for each generator we compare to (CTGAN,  TVAE , TabDDPM, ARF, NFlow). Our choice of these ‘standard’ models as baselines is guided by examining the tabular synthetic data generation literature, where we see that a similar selection of baseline models is made in a number of papers (e.g [1] include CTGAN, TVAE, NFlow, TabDDPM, [2] include ARF, TVAE, TabDDPM, [3] include CTGAN, TVAE, NFlow, [4] include TVAE, CTGAN), and by wanting to compare to models that span across different families of generative models (VAE-based, GAN-based, flow-based, diffusion-based, forest-based). Furthermore, as we state in Section 7, we primarily use the $\texttt{synthcity}$ package, an open source package which implements an array of generative models, to facilitate reproducibility and fair comparison of these different methods. For DP generation, see (B).
>
> ---
>
> **(B) Comparison with more competitive differentially private generative models**
>
> We thank you for the added literature on DP generative models, we have now **added these to our extended related works** discussion in Appendix B. For our empirical study in the DP setting, you are correct in saying that our comparison is less robust, and our choice of baseline is lacking. As such, we have we have **run a further set of DP generation experiments, comparing STEAM with four baselines - AIM [5], GEM [6], MST [7], and RAP [8]** - on the ACTG dataset used in Section 7.1. We now base our updated Section 7.3 on this experiment, rather than comparing with DP-GAN. Of these four baselines, we found that AIM produced the best data across different $\epsilon$, and its STEAM analogue was the best performing STEAM model. As such, we include these results for AIM vs. $STEAM_{AIM}$ in the main body of the updated manuscript, and include the full results using the other baseline models in Appendix I. For convenience, we will include the updated takeaway from this experiment here.
>
> Figure 5, in Section 7.3 shows the results. **$STEAM_{AIM}$ models $P_{Y|W,\mathbf{X}}$ better on all tested values of $\epsilon$**, as $U_{PEHE}$ is significantly lower than for standard AIM. **$P_{W|\mathbf{X}}$ is better modelled by $STEAM_{AIM}$ at small $\epsilon$**, with equivalent performance between the methods at less conservative budgets. **$P_{\mathbf{X}}$, on the other hand, is better preserved by standard AIM**, scoring higher on $P_{\alpha, \mathbf{X}}$ and $R_{\beta, \mathbf{X}}$ at most $\epsilon$. This is likely because assigning $Q_\mathbf{X}$ one third of the budget of the standard AIM model and having it model largely the same distribution, save for the removed $W$ and $Y$, is prohibitively restrictive given the high-dimensionality of $\mathbf{X}$. As such, with uniform distribution of ($\epsilon, \delta$) across each component, we see a trade-off between $STEAM_{AIM}$ and standard AIM, where **$STEAM_{AIM}$ better preserves $P_{W|\mathbf{X}}$ and $P_{Y|W,\mathbf{X}}$, while standard AIM preserves $P_{\mathbf{X}}$ better**. Distributing ($\epsilon, \delta$) differently amongst  $Q_\mathbf{X}$, $Q_{W|\mathbf{X}}$, and $Q_{Y|W,\mathbf{X}}$ could address this trade-off, as we discuss in Appendix M.

---

> ### Author Response · Authors · 2024-11-22
> **Author response to uQ2Q (Part 2/2)**
>
> With the current uniform allocation of ($\delta$, $\epsilon$) amongst the STEAM components, these results can be seen as a **lower bound on the potential performance of STEAM** in DP generation. As such, we are encouraged that STEAM remains competitive against this updated set of baseline models, and we see attempts at sophisticated optimisation of the privacy budget distribution as a fruitful area of future work.
>
> **_Update:_**  We have **extended our related works on DP generation** in Appendix B to better represent the recent literature. We have changed Section 7.3 and **added new results to compare DP generation performance** between AIM and $STEAM_{AIM}$, on the ACTG dataset. We have **added more results for other DP baselines tested (GEM, MST, RAP)** in Appendix I.
>
> ---
>
> **(C) Formal motivation for new metrics**
>
> You are correct that in Section 5, in our illustrative experiment on the failure of current metrics to identify differences in $Q_{Y|W,\mathbf{X}}$, we only highlight the improvement of our $U_{PEHE}$ metric. However, as we state in our original manuscript, a similar experiment, which instead shows how current metrics also struggle to identify differences in $Q_{W|\mathbf{X}}$, in comparison to our $JSD_\pi$ metric, is also included in Appendix C.
>
> Nevertheless, we agree that a more **formal approach can be made** to show that existing, joint-distribution-level metrics, such as Kullback-Leibler, will tend to **lose sight of changes in $Q_{Y|W,\mathbf{X}}$ and $Q_{W|\mathbf{X}}$**, under certain assumptions. Such an argument will generalize better than the empirical results currently shown, and in conjunction they will both provide **strong motivation for establishing our own set of metrics** for synthetic data containing treatments. **We make this formal argument in our updated Section 5.1**, please see Theorem 1 and the related assumptions in the updated manuscript for context.
>
> **Theorem 1 shows that KL divergence loses sensitivity to $W|\mathbf{X}$ and $Y|W,\mathbf{X}$ as $d$ grows**, suggesting that it will struggle in selecting between any proposal distributions which take the assumed form, since their scores will converge to similar values despite any difference in their modelling of $P_{W|\mathbf{X}}$ or $P_{Y|W,\mathbf{X}}$. Of course, the assumptions we make here will not necessarily hold for either the true distribution, or the proposal distribution, but we see this result as a nice motivational point. Furthermore, our **empirical verification of this phenomena, and its apparent extension to other joint-distribution-level metrics** in Appendix D, adds to our motivation for establishing a new set of metrics. Also, we see **the importance of our desiderata**, outlined in Section 4, as **strong motivation in itself for establishing metrics to disentangle between each of these three distributions** (in order to answer **_Q1-3_** as posed in Section 1), which metrics that do not directly target these distributions (such as all-encompassing, joint-distribution-level metrics) inherently cannot do.
>
> Since $JSD_\pi$ and $U_{PEHE}$ directly measure the preservation of $P_{W|\mathbf{X}}$ and $P_{Y|W,\mathbf{X}}$ without also trying to incorporate some notion of divergence between the covariate distributions $P_{\mathbf{X}}$ and $Q_{\mathbf{X}}$ , these do not exhibit this 'curse of dimensionality'-type behaviour, assuming the machine-learning models used in their calculation are well-specified.
>
> **_Update:_** We have **added this new formal argument** to Section 5.1 on the **inadequacy of existing joint-distribution-level metrics**. We have moved the illustrative experiments into the Appendix as further evidence that **such phenomena are observable empirically**.
>
> ---
>
> **(D) Minor changes**
>
> We agree that noting the DP composition theorem applies to STEAM is a trivial result, and so we have now labeled it as a proposition, rather than a theorem. We include the proof in Appendix G for the benefit of the causal inference community, who may not be so familiar with the relevant DP theorems.
>
> ---
>
> Thank you once again. We hope that we have addressed all your comments, and we greatly appreciate your feedback.

---

> ### Author Response · Authors · 2024-11-22
> **References**
>
> [1] Seedat, Nabeel, et al. "Curated llm: Synergy of llms and data curation for tabular augmentation in ultra low-data regimes." arXiv preprint arXiv:2312.12112 (2023).
>
> [2] Li, Jiayu, et al. "TAEGAN: Generating Synthetic Tabular Data For Data Augmentation." arXiv preprint arXiv:2410.01933 (2024).
>
> [3] Liu, Tennison, et al. "GOGGLE: Generative modelling for tabular data by learning relational structure." The Eleventh International Conference on Learning Representations. 2023.
>
> [4] Rashidi, Hooman H., et al. "A novel and fully automated platform for synthetic tabular data generation and validation." Scientific Reports 14.1 (2024): 23312.
>
> [5] Ryan McKenna, Brett Mullins, Daniel Sheldon, and Gerome Miklau. Aim: An adaptive and iterative mechanism for differentially private synthetic data, 2024. URL https://arxiv.org/abs/2201.12677.
>
> [6] Terrance Liu, Giuseppe Vietri, and Steven Z Wu. Iterative methods for private synthetic data: Unifying framework and new methods. Advances in Neural Information Processing Systems, 34: 690–702, 2021.
>
> [7] McKenna, Ryan & Miklau, Gerome & Sheldon, Daniel. (2021). Winning the NIST Contest: A scalable and general approach to differentially private synthetic data. Journal of Privacy and Confidentiality. 11. 10.29012/jpc.778.
>
> [8] Sergul Aydore, William Brown, Michael Kearns, Krishnaram Kenthapadi, Luca Melis, Aaron Roth, and Ankit A Siva. Differentially private query release through adaptive projection. In International Conference on Machine Learning, pp. 457–467. PMLR, 2021.

---

> > ### Comment · Reviewer_uQ2Q · 2024-11-24
> >
> > Apologies for this brief comment, I am still going through your responses to my review and to the other reviewers along with the revised PDF.
> >
> > I only want to say that, in my opinion, the rebuttal and the revised PDF borderline does not "clearly communicate what has been changed," as is required according to the ICLR guidelines.
> >
> > General practice is that minor changes can made without highlighting, but major additions, rewrites and new figures/tables should be **highlighted by a different color text.**

---

> > > ### Author Response · Authors · 2024-11-24
> > >
> > > Dear reviewer uQ2Q,
> > >
> > > We give our sincere apologies for this oversight, and we are very thankful that you have pointed this out to us. Please see the new revised manuscript, which has the additions, rewrites, and moved sections highlighted in teal. Please let us know if anything else is unclear, and we thank you again for your continued commitment to reviewing our work!

---

> > > > ### Comment · Reviewer_uQ2Q · 2024-11-25
> > > >
> > > > In light of the substantial efforts made during the rebuttal, and reasonable attempts made at the reviewer's concerns/suggestions, I have decided to raise my score from a Strong Reject to a Reject. However, **I caution the AC in accepting this paper, for the following reasons:**
> > > > 1. I believe this version of the paper has now diverged **substantially** from the original submission, to the extent that it should likely undergo a fresh review process with new reviewers.
> > > > 2. I expect that "Theorem 1," on KL divergence, has been observed before in the stats/ML literature, and though the theory helps us understand why KL divergence might not make the most sense in causal settings, its not particularly motivating for the author's proposed metrics. Essentially, the authors use that you can decomposed $D_{KL}(P | Q) = \sum^d_{i=1} D_{KL}(P_{X_i} | Q_{X_i}) +$ $bounded terms.$ since each $D_{KL}(P_{X_i} | Q_{X_i})$ is non-negative and bounded away from zero, their sum grows linearly with $d$. The bounded terms then don't depend on $d$ and thus become negligible in comparison as $d \to \infty$. This is straightforward to see given properties of KL, i.e. that the marginal divergences over $X_i$ will dominate the total divergence in high dimensions, as its a direct consequence of summing a large number of terms where each gives a fixed amount to the total divergence. So my question to the authors: how does this observation lead naturally/formally to your proposed metrics?
> > > > 3. I am still unconvinced by the privacy story, and by the "novelty" of the STEAM framework in the privacy setting. Sequential modeling synthesizers of this form are not a new idea (see Bowen et. al, https://arxiv.org/pdf/1602.01063 , Section 3.2.3 , or QUAIL from Rosenblatt et. al, https://arxiv.org/pdf/2011.05537 ). Thus, the findings (even with AIM/GEM) are not particularly surprising. A more interesting, truly algorithmic question, would be some sort of utility bound given that shows how the budget allocation of the STEAM framework to the conditional distribution leads to direct improvement on the metrics that the authors seem to care about. Otherwise, this paper seems like more of a story about "what you could do" than "what you should do."
> > > >
> > > > On a hopeful note: the authors have clearly demonstrated good faith efforts to improve their work, and in that respect, I commend them. Perhaps, if its true that "the main contribution of [their] work is not a new method for incorporating differential privacy (DP) into synthetic data generation", and that STEAM is not particularly novel in that respect, then the paper could clarify its contribution by removing the DP angle.

---

> > > > > ### Author Response · Authors · 2024-11-30
> > > > >
> > > > > Dear Reviewer uQ2Q,
> > > > >
> > > > > Thank you once again for your thoughtful feedback and involvement during this discussion period, which has significantly improved our paper. We hope that our follow-up response has addressed your concerns about the number of changes made to our paper, and has clarified our positioning on the main contributions of our work.
> > > > >
> > > > > As the discussion period is nearing its end, we wanted to check if you have any remaining concerns or questions we can address. If our revisions meet your expectations, we hope you might consider increasing your score from Reject.
> > > > >
> > > > > We truly appreciate your time and invaluable insights!

---

> > > > > > ### Comment · Reviewer_uQ2Q · 2024-12-01
> > > > > >
> > > > > > Thank you again for all your efforts in the rebuttal, and for improving the paper. However, I have decided to maintain my score, and to allow the AC to review all of the reviewers comments and make their final decision based on their own perspective of the paper.

---

> ### Author Response · Authors · 2024-11-25
>
> Thank you for your continued engagement with our work, we appreciate the effort you have made during this discussion period. We will respond to each of your points in turn.
>
> ---
>
> **1. Changes made to the paper**
>
> During this discussion period, we have made an effort to engage with **all** reviewers and their respective comments, leading to additions and revisions for our submission. We believe that this has been **well within the spirit of ICLR's discussion period**, and we politely disagree that our changes necessitate new reviews. **There has not been substantial change to the setting of our paper, nor our proposed metrics/generation method, nor the conclusions drawn from our results.** We believe that all additions and revisions we have made fall in line with what is acceptable, and standard, for a conference rebuttal period, in that they have not substantially changed the main contributions of our work. While **these changes have improved our work**, they are **not so fundamental as to oblige new reviews**. We give our reasoning now.
>
> As we state in Section 1, the contributions of our work are the following:
>
> 1. Establishing a set of desiderata for what a 'good' synthetic dataset containing treatments involves.
>
> 2. Defining a set of metrics to measure adherence to these desiderata.
>
> 3. Proposing a generation framework that optimises for these desiderata.
>
> 4. Empirically demonstrating state-of-the-art performance in generating synthetic data containing treatments.
>
> These were our main contributions before the discussion period began, and they still remain our main contributions. In total, the key changes made to the main body of the paper during the discussion period amass to 74 lines (with figures and equations). These include:
>
> 1. Addition of a small **formal result that complements our existing empirical demonstration** of a phenomenon which forms part of our motivation to propose new metrics (Section 5.1 lines 216-246). While adding credibility and generality to our previous motivational piece, this formal result does not fundamentally change our motivation for proposing our metrics. This motivation comes directly from our desiderata in Section 4, and from the desire to answer the key questions **_Q1-3_** from Section 1, which remain unchanged.
>
> 2. Addition of an **extended definition and description** of an existing metric which we deploy, **to improve accessibility for unfamiliar readers** (Section 5.2.1 lines 258-272). Importantly, while this has ostensibly changed the definitions for $P_{\alpha, \mathbf{X}}$ and $R_{\beta, \mathbf{X}}$, in reality they remain **exactly the same**. We simply bring forward the full, mathematical definition from [1], which we previously withheld and referred to only by way of reference.
>
> 3. **Re-running a previous experiment on a new dataset, with new baseline methods** for more robust conclusions (Section 7.3 lines 486-516). Adding baseline methods and investigating performance on different datasets is an update often made during rebuttal periods, and the broad takeaway from this experiment - that STEAM is competitive in differentially-private generation - remains the same.
>
> Beyond these, additions were made to the appendix, which chiefly serve to better contextualise our work amongst related fields, list results for extended sets of baselines, give advice for practical applications of our metrics, and suggest directions for future work. Again, none of these additions change our paper at a fundamental level, they simply better position our work and strengthen the evidence for the efficacy of our method.
>
> ---
>
> [1] Alaa, Ahmed, et al. "How faithful is your synthetic data? sample-level metrics for evaluating and auditing generative models." International Conference on Machine Learning. PMLR, 2022.

---

> ### Author Response · Authors · 2024-11-25
>
> ---
> **2. Theorem 1**
>
> Decomposition of KL divergence via the chain rule is a well-known property, and we have no doubt that it has been used in a variety of applications in the statistics/ML literature. We have **not** seen a theorem, however, that elaborates on this decomposition and how it leads to KL divergence losing sensitivity to particular one-dimensional conditional distributions, such as $W|\mathbf{X}$ and $Y|W,\mathbf{X}$. If you have such an example in mind, **we would happily cite it**, as we do not suggest that this theorem is a core contribution of our work, nor that it is strictly necessary for motivating our proposed metrics. We will clarify this now.
>
> We have **multiple motivational points for metrics of synthetic data containing treatments**, at different parts of our paper. Firstly, we intuitively suggest that a set of **metrics in this setting should answer _Q1-3_** (Section 1), because these are important questions for downstream analysts. Next, as a more formal requirement, we suggest that **a metric set should measure adherence to our desiderata**, which are very related to **_Q1-3_**. Finally, in application, **a metric set should be able to conduct correct model selection** and rank models based on a particular desideratum.
>
> Existing metrics do **not** satisfy any of these three points. Inherently, they cannot answer **_Q1-3_** because they do not differentiate between $\mathbf{X},\ W$, and $Y$ (Table 1). Similarly, while they may loosely correlate with our desiderata, they **do not directly measure the distributions of interest**. As a result, in some circumstances **their model selection will be poor**, particularly if desideratum (ii) or (iii) is the focus. Formally, we show that this is true for KL divergence in theorem 1, and empirically we show this occurs for a wide array of metrics in Appendix D.
>
> Naturally, the inadequacy of existing metrics acts as motivation to propose new metrics. These metrics should **directly target our desiderata**, **be able to answer _Q1-3_**, and **conduct generally correct model selection**. These are the three underlying motivations for our metric set. As we describe in Section 5.2, we propose metrics that do indeed directly target an individual desideratum, and that can answer our key questions. Furthermore, because the metrics targeting the $W|\mathbf{X}$ and $Y|W,\mathbf{X}$ distributions do not attempt to also incorporate divergence between covariate distributions, they will not lose sensitivity to these distributions as $d \rightarrow \infty$. We show that this results in better model selection than existing metrics through the empirical investigation in Appendix D.
>
> ---
> **3. Privacy contribution**
>
> Thank you for the additional references, they are useful for clarifying the context of our work, and we will cite them accordingly. We do believe it is important to note, however, that STEAM conducts a particular kind of sequential generation, which is **tailored to the setting of synthetic data containing treatments**, mimicking the typical DGP for this data. Sequential generation in **this setting**, without assuming full knowledge of the causal graph, is an important contribution of our work. Indeed, as we reference in Section 1, many related works which focus on synthetic data in medicine do **not** consider sequential generation, despite its benefits in preserving important relationships for tasks such as treatment effect estimation, as demonstrated in our empirical analyses. We believe that **such downstream tasks are largely overlooked in the broad synthetic data literature**. Bringing the data needs of the treatment effect estimation community to the attention of the synthetic data community is an important step to make.
>
> We agree with your comment that formal reasoning on utility bounds for STEAM is an interesting direction, and would be a great contribution to make. We see this as an extension beyond the scope of this work, however. Indeed, in our contributions in Section 1, **the majority are not related to DP**. We do mention, in contribution 3, that STEAM satisfies DP **if desired** indicating that this setting is **not the only focus of this work**. Indeed, subsequent **sections on differential privacy (Section 6.2 and Section 7.3) are relatively small** (less than a page). In contrast, **our other contributions, and experiments in the non-DP setting, take up substantially more space** in the paper. The desiderata discussion spans 1.5 pages, the metrics span 2 pages, and our method and its performance in the non-DP setting spans 2.5 pages. In comparison, the DP aspect is small. Since privacy is inherently relevant to synthetic data, especially in medicine, we see the compatibility of STEAM with DP as useful, and its competitive performance is promising.

---

### Official Review · Reviewer_9kc4 · 2024-11-04

**Soundness:** 2
**Presentation:** 3
**Contribution:** 3
**Rating:** 8
**Confidence:** 3

**Summary:**

The paper presents **STEAM**, a method for generating synthetic medical data to support causal inference while addressing privacy limitations. STEAM is designed to replicate critical aspects of real data, including covariates, treatment assignments, and outcome distributions, to enable accurate analysis of treatment effects.

To evaluate synthetic data quality, the authors introduce new metrics that assess how well the generated data supports causal inference, addressing gaps in traditional evaluation methods. STEAM also incorporates differential privacy to enhance security. Empirical results suggest that STEAM performs well in complex, high-dimensional scenarios, making it suitable for applications in healthcare and other fields requiring secure, synthetic data for causal analysis.

**Strengths:**

1.**Novelty**

The paper presents a novel approach to synthetic data generation that prioritizes causal relationships, often overlooked in traditional methods. **STEAM** addresses this by modeling covariates, treatment assignments, and outcomes, while ensuring usual Differential Privacy concerns are easily transferable.

2.**Quality**

The methodology is thorough, with clear desiderata and structured evaluation metrics. Empirical results show STEAM’s effectiveness, especially in complex, high-dimensional scenarios.

3.**Clarity**

The paper is well-organized, clearly guiding the reader through the problem, methodology, and results. Each component and metric is explained with clarity.

4.**Significance**

By enabling privacy-preserving, causally accurate synthetic data, this work has broad applications, particularly in healthcare and other sensitive fields. The proposed metrics and STEAM framework make synthetic data more viable for impactful, real-world research.

**Weaknesses:**

1. **Limited Accessibility of Metrics**
   The paper would benefit from including reminders of key equations, particularly for Jensen-Shannon Divergence (JSD), $P_\alpha$, and $R_\beta$. This addition would improve accessibility for non-expert readers, helping them better understand and apply the proposed evaluation metrics.

2. **Lack of Comparison with Relevant Causal Generative Models**
   The evaluation does not include comparisons with established causal generative models that handle interventional and counterfactual data, such as DCM [1], VACA [2], and DoWhy-GCM [3]. These models account for treatment, outcome, and counterfactuals and are capable of generating data with similar causal structures. Benchmarking STEAM against these methods could provide a more comprehensive view of its relative performance and potential advantages in synthetic data generation for causal inference.

3. **Omission of Closely Related Work**
   The paper does not sufficiently reference the reasearch area of causal generative models. Inclusing mentions of these works would strengthen the contextual background, positioning STEAM within the landscape of existing work and clarifying its contributions to the field.

References:
1. Blöbaum, P., Götz, P., Budhathoki, K., Mastakouri, A. A., & Janzing, D. (2024). *DoWhy-GCM: An extension of DoWhy for causal inference in graphical causal models*. [arXiv:2206.06821](https://arxiv.org/abs/2206.06821).
2. Sanchez-Martin, P., Rateike, M., & Valera, I. (2021). *VACA: Design of Variational Graph Autoencoders for Interventional and Counterfactual Queries*. [arXiv:2110.14690](https://arxiv.org/abs/2110.14690).
3. Chao, P., Blöbaum, P., Patel, S., & Kasiviswanathan, S. P. (2024). *Modeling Causal Mechanisms with Diffusion Models for Interventional and Counterfactual Queries*. [arXiv:2302.00860](https://arxiv.org/abs/2302.00860).

**Questions:**

The $U_{PEHE}$ metric is based on evaluating a family of CATE estimators. Could the authors clarify what size this family should ideally be for reliable estimation, and outline the computational cost involved? Additional context on this aspect would be helpful to assess the feasibility of $U_{PEHE}$ across different applications.

How might the STEAM approach be extended or adapted to support arbitrary causal graphs? Any insights on this would provide useful context for understanding the broader applicability of the method to more complex causal structures.

---

> ### Author Response · Authors · 2024-11-22
> **Author response to 9kc4 (Part 1/4)**
>
> Thank you for your thoughtful comments and suggestions. We give answers to each of the following in turn, as well as pointing out corresponding updates to the revised manuscript:
>
> - (A) Accessibility of metrics
> - (B) Comparison with causal generative models
> - (C) Guidelines for $\mathcal{F}$ when calculating $U_{PEHE}$
> - (D) Extension to arbitrary causal graphs
> ---
>
> **(A) Accessibility of metrics**
>
> We hope that our metrics are used beyond our work, as we think they add important information beyond joint-distribution-level metrics when assessing synthetic data for causal inference, so we are thankful for your help in improving their accessibility. We have added reminders of the definitions of Jensen-Shannon Distance, $P_\alpha$, and $R_\beta$ prior to their use in defining our metrics.
>
> **_Update:_** We have **added definitions for Jensen-Shannon Distance, $P_\alpha$, and $R_\beta$** to Section 5.2.
>
> ---
>
> **(B) Comparison with causal generative models**
>
> We agree that adding a discussion and experimental comparison with causal generative models is an important addition to our paper. We will discuss the papers you have cited, and give a **broader overview of causal generative models, in our related works in Appendix B** to better contextualize our work. In doing so, we will highlight some **important points of difference** between STEAM and existing methods.
>
> Most notably,  [1], [2], and [3] **assume that the entire causal graph is known** to the data holder, which is a **more restrictive assumption than we make with STEAM**. We assume that our specification of the underlying DGP ($X \sim P_\mathbf{X},\ W \sim P_{W|\mathbf{X}},\ Y \sim P_{Y|W,\mathbf{X}}$) is correct for datasets containing treatments, but we do **not** require knowledge of the causal graph, as we do not need to know the causal links between individual variables. We do not assume knowledge of the causal relationships amongst the covariates, nor knowledge of which covariates cause treatment assignment or patient outcomes. As such, we make **less restrictive assumptions** than these works, and we suggest that our approach is more realistic in complex, real-world scenarios, such as those that arise in medicine, where the true causal graph is unlikely to be available. Note that our setting follows established literature in causal inference that does not impose any assumption on the full underlying causal graph.
>
> Furthermore, the motivation for these works differs from ours. We seek to generate useful synthetic data only from the observational distribution, for use by analysts with goals such as **treatment effect estimation (e.g., CATE)**, while [1], [2], and [3] primarily focus on generating data to answer **graph-specific interventional and counterfactual queries that require knowledge of the full causal graph**. Nevertheless, you are correct in point out that such methods can be used to generate data from the observational distribution, and we agree that experimental comparisons will strengthen the contribution of our paper.
>
> **Experiments with new baselines:** In order to compare to some baseline causal generative models (specifically the diffusion-based causal model (DCM) from [3] and the additive noise model (ANM) [5] implementation in the $\texttt{DoWhy-GCM}$ python [1]) we must first reconcile their differences in assumptions. For the real-world datasets we use in Section 7.1, we do not have access to the true causal graph, we simply know which features are the treatment and outcome. As such, we must first construct a causal graph to supply to these methods. We do so with three methods:
>
> 1. Construction of a naive graph $\mathcal{G}_{naive}$, in which each covariate causes $W$ and $Y$, $W$ causes $Y$, and every pair of covariates has a causal relationship between them;
> 2. Using the constraint-based PC causal discovery algorithm [4] to discover an estimated graph $\mathcal{G}_{discovered}$ from the Markov Equivalence Class for the true causal graph and;
> 3. Pruning the discovered causal graph by removing any edges which contradict the DGP we assume. As such, any edges from $Y$ to $W$ or $\mathbf{X}$, or from $W$ to $\mathbf{X}$ are removed to form $\mathcal{G}_{pruned}$.
>
> We compare the DCM and ANM methods using each of these causal graph discovery methods with the best performing STEAM models on each of the datasets used in Section 7.1. **We include the new results in the Appendix O**, and we add the table in the next comment for convenience.
>
> For each dataset, we see that the relevant **STEAM model outperforms all of the causal generative models in almost every metric**, only being outperformed to a statistically significant level in $R_{\beta, X}$ on the ACIC dataset. These results validate that, when the true causal graph is not known, our less restrictive assumptions enable more useful generation of synthetic data containing treatments. We also see that the differences between the graph discovery methods are relatively small.

---

> ### Author Response · Authors · 2024-11-22
> **Author response to 9kc4 (Part 2/4)**
>
> Table 16: $P_{\alpha, \mathbf{X}}$, $R_{\beta, \mathbf{X}}$, $JSD_\pi$, and $U_\text{PEHE}$ values for the best performing STEAM models from Appendix I in comparison to causal generative models. Averaged over 20 runs, with 95\% confidence intervals. Bold indicates the statistically significant best performing model on that metric and dataset.
> | Dataset  | Model                   | $\boldsymbol{P_{\alpha, \mathbf{X}}}$ ($\uparrow$) | $\boldsymbol{R_{\beta, \mathbf{X}}}$ ($\uparrow$) | $\boldsymbol{\textbf{JSD}_\pi}$ ($\uparrow$) | $\boldsymbol{U_\textbf{PEHE}}$ ($\downarrow$) |
> |-------------------------------------------|-------------------------------------|----------------------------------------------------|---------------------------------------------------|----------------------------------------------|-----------------------------------------------|
> | ACTG                  | STEAM$_\text{TVAE}$  |**0.929 $\pm$ 0.008**| **0.486 $\pm$ 0.009**| 0.958 $\pm$ 0.004   | **0.492 $\pm$ 0.011**   |
> |  | DCM $\mathcal{G}_\text{naive}$      | 0.773 $\pm$ 0.013           | 0.369 $\pm$ 0.006     | 0.937 $\pm$ 0.006   | 0.665 $\pm$ 0.034   |
> | | DCM $\mathcal{G}_\text{discovered}$ | 0.756 $\pm$ 0.011     | 0.350 $\pm$ 0.007       | 0.956 $\pm$ 0.005                            | 0.605 $\pm$ 0.023 |
> |   | DCM $\mathcal{G}_\text{pruned}$     | 0.758 $\pm$ 0.013   | 0.358 $\pm$ 0.007  | 0.957 $\pm$ 0.003                            | 0.596 $\pm$ 0.017                             |
> |   | ANM $\mathcal{G}_\text{naive}$      | 0.787 $\pm$ 0.007                                  | 0.389 $\pm$ 0.008                                 | 0.954 $\pm$ 0.005                            | 0.580 $\pm$ 0.017                             |
> |                                           | ANM $\mathcal{G}_\text{discovered}$ | 0.836 $\pm$ 0.007                                  | 0.419 $\pm$ 0.007                                 | 0.952 $\pm$ 0.004                            | 0.578 $\pm$ 0.019                             |
> |                                           | ANM $\mathcal{G}_\text{pruned}$     | 0.839 $\pm$ 0.008                                  | 0.412 $\pm$ 0.005                                 | 0.952 $\pm$ 0.005                            | 0.582 $\pm$ 0.014                             |
> | |  | | | |  |
> | IHDP                  | STEAM$_\text{CTGAN}$              | 0.674 $\pm$ 0.014  | **0.424 $\pm$ 0.011**| **0.928 $\pm$ 0.009**| **1.709 $\pm$ 0.052** |
> |        | DCM $\mathcal{G}_\text{naive}$      | 0.557 $\pm$ 0.010  | 0.340 $\pm$ 0.009    | 0.883 $\pm$ 0.016                            | 4.878 $\pm$ 0.395                             |
> |                                           | DCM $\mathcal{G}_\text{discovered}$ | 0.658 $\pm$ 0.011| 0.360 $\pm$ 0.007                                 | 0.893 $\pm$ 0.008                            | 2.059 $\pm$ 0.140                             |
> |                                           | DCM ${\mathcal{G}_\text{pruned}}^*$ | 0.658 $\pm$ 0.011                                  | 0.360 $\pm$ 0.007                                 | 0.893 $\pm$ 0.008                            | 2.059 $\pm$ 0.140                             |
> |                                           | ANM $\mathcal{G}_\text{naive}$      | 0.597 $\pm$ 0.029                                  | 0.379 $\pm$ 0.011                                 | 0.900 $\pm$ 0.005                            | 1.868 $\pm$ 0.147                             |
> |                                           | ANM $\mathcal{G}_\text{discovered}$ | 0.589 $\pm$ 0.012                                  | 0.359 $\pm$ 0.009                                 | 0.892 $\pm$ 0.008                            | 1.865 $\pm$ 0.059                             |
> |                                           | ANM ${\mathcal{G}_\text{pruned}}^*$ | 0.589 $\pm$ 0.012 | 0.359 $\pm$ 0.009                                 | 0.892 $\pm$ 0.008                            | 1.865 $\pm$ 0.059   |
> | |  | | | |  |
> | $\text{ACIC}^\dagger$ | STEAM$_\text{ARF}$  | 0.939 $\pm$ 0.004| 0.393 $\pm$ 0.004  | **0.977 $\pm$ 0.002** | **2.176 $\pm$ 0.141**   |
> | | DCM $\mathcal{G}_\text{discovered}$ | 0.942 $\pm$ 0.004   | 0.422 $\pm$ 0.003  | 0.957 $\pm$ 0.003                            | 4.249 $\pm$ 0.132 |
> |                                           | DCM $\mathcal{G}_\text{pruned}$     | 0.939 $\pm$ 0.004 | 0.420 $\pm$ 0.004| 0.959 $\pm$ 0.002                            | 4.340 $\pm$ 0.159                             |
> |   | ANM $\mathcal{G}_\text{discovered}$ | 0.929 $\pm$ 0.003 | 0.404 $\pm$ 0.003   | 0.872 $\pm$ 0.002                            | 4.193 $\pm$ 0.127   |
> |          | ANM $\mathcal{G}_\text{pruned}$     | 0.930 $\pm$ 0.004 | 0.404 $\pm$ 0.003| 0.880 $\pm$ 0.002                            | 4.481 $\pm$ 0.174|
>
> \* $\mathcal{G}$ pruned is the same as $\mathcal{G}$ discovered for IHDP
>
> $\dagger$ Excessive runtime caused the exclusion of $\mathcal{G}_\text{naive}$ ACIC results

---

> ### Author Response · Authors · 2024-11-22
> **Author response to 9kc4 (Part 3/4)**
>
> Furthermore, STEAM’s design can, in principle, incorporate **any** generative model for $Q_\mathbf{X}$, essentially acting as a wrapper around $Q_\mathbf{X}$ to improve its generation quality for causal inference tasks. This allows STEAM to very easily empower existing generative modelling frameworks, without having to incorporate bespoke generators. Also, it allows STEAM models to continuously improve along with the base generative model. Existing causal generative models do not allow such flexibility, and therefore cannot as easily benefit from future improvements.
>
> Finally, the causal generative models we compared against here cannot immediately satisfy differential privacy, and any causal generative model which requires knowledge of the causal graph must ensure that the causal discovery methods used are also privacy-enabled.
>
> **_Update:_** We have added a **discussion on causal generative models** to our related works section. We have added **new experimental results using the causal generative models DCM and ANM as baselines** to Appendix O.
>
> ---
>
> **(C) Guideline for $\mathcal{F}$ when calculating $U_{PEHE}$**
>
> In CATE estimation, model validation is a difficult task [6]. As such, it is reasonable to expect that a set of downstream analysts conducting CATE estimation on $D_{synth}$ will use different learners. Therefore, we want $U_{PEHE}$ to reflect the expected difference in downstream performance between $D_{synth}$ and $D_{real}$ across a diverse array of potential learners, such that it is representative for the entire population of analysts, and has limited bias towards any particular learner class. To achieve this, we propose averaging $U_{PEHE}$ across a family of CATE learners $\mathcal{F}$, and we suggest that larger $|\mathcal{F}|$, and diverse selection of the learners within $\mathcal{F}$, is preferable.
>
> Of course, there is a trade off between the size of $\mathcal{F}$, and therefore the stability of $U_\text{PEHE}$, and the computational cost of repeated CATE estimation. With this in mind, to limit the computation involved in calculating $U_\text{PEHE}$, we suggest that users should be selective of the learners included in $\mathcal{F}$ to maximize learner diversity, and minimise $|\mathcal{F}|$. For example, in our experiments we set $|\mathcal{F}|=4$, and we chose learners from both of the high-level CATE learning strategies described in [7] (i.e. one-step plug-in learners, and two-step learners). Specifically, for the one-step learners we use S- and T- learners [8], and for the two-step learners we use RA and DR learners [9]. All four of these learners conduct CATE estimation differently, and encode different inductive biases in their approaches, and thus they form a good diverse base for $\mathcal{F}$. We see across the runs in Section 7.1 that the 95% confidence intervals we report for $U_\text{PEHE}$ using this $\mathcal{F}$ seem reasonable compared to the magnitude of the metric itself, and we suggest that such $\mathcal{F}$ is a good starting point for a user. If this user has requirements on their computational costs, or desired stability of $U_\text{PEHE}$, then $\mathcal{F}$ can be adjusted accordingly.
>
>
> For our experiments, on each of the real datasets from Section 7.1, the runtime for calculating $U_{PEHE}$ for one run was 26, 60, and 191 seconds on the ACTG, IHDP, ACIC respectively. Note that these are much less than the typical generation times for each dataset, so this step is unlikely to be a large time burden for the data holder. Also note that these calculations can be parallelized across the learner classes, which we did not do, and this can improve the computational feasibility of using a larger $|\mathcal{F}|$.
>
> **_Update:_** We have **added a discussion on the considerations that should be made when choosing $\mathcal{F}$**, and we have **specified our experimental settings and the typical run times** for calculating $U_{PEHE}$  on each dataset in Appendix F.

---

> ### Author Response · Authors · 2024-11-22
> **Author response to 9kc4 (Part 4/4)**
>
> **(D) Extension to arbitrary causal graphs**
>
> As we note in (B), STEAM places **minimal assumptions upon the causal graph of $\mathcal{D}_{real}$**, and it does not require full knowledge of variable-level causality. We see that our assumed DGP involving $P_{\mathbf{X}}$, $P_{W|\mathbf{X}}$, and $P_{Y|W,\mathbf{X}}$ is applicable across a very wide range of scenarios for datasets containing treatments, particularly in medical settings, as we describe in Section 6. If this DGP is known to not hold for $\mathcal{D}_{real}$, then it would be unadvisable to apply STEAM in its current form, as its inductive biases may not be helpful. However, if a similar setting arises where the overarching DGP is known, but the specific causal graph is not, then altering the generation order between $\mathbf{X}$, $W$, and $Y$ to mimic this alternative DGP, with a similar underlying motivation as in STEAM - that generation mimicking the real DGP is preferable to joint-level generation - can be done.
>
> An example of a more complicated setting, where STEAM is not immediately applicable is when dealing with longitudinal data, which has time-varying covariates, treatments, and outcomes. In this setting, our assumed DGP would not hold in general, as features at a specific time point will likely have temporal causal relationships with earlier features, e.g. at time t, $X_{t}$ could be affected by $W_{t-1}$, which is not modelled in our assumed DGP. Potential alterations to STEAM could adjust for this time-varying DGP, where one could consider a ‘base’ STEAM model, as described in this work, operating at each time step, with additional inputs from the immediately preceding time step to model temporal causal relationships.
>
> **_Update:_** We have **included this discussion on potential extensions of STEAM to alternative DGPs** in our future works Appendix P.
>
> ---
>
> Thank you once again. We hope that we have addressed all your comments, and we greatly appreciate your feedback.

---

> ### Author Response · Authors · 2024-11-22
> **References**
>
> [1] Blöbaum, P., Götz, P., Budhathoki, K., Mastakouri, A. A., & Janzing, D. (2024). DoWhy-GCM: An extension of DoWhy for causal inference in graphical causal models. arXiv:2206.06821.
>
> [2] Sanchez-Martin, P., Rateike, M., & Valera, I. (2021). VACA: Design of Variational Graph Autoencoders for Interventional and Counterfactual Queries. arXiv:2110.14690.
>
> [3] Chao, P., Blöbaum, P., Patel, S., & Kasiviswanathan, S. P. (2024). Modeling Causal Mechanisms with Diffusion Models for Interventional and Counterfactual Queries. arXiv:2302.00860.
>
> [4] Peter Spirtes, Clark Glymour, and Richard Scheines. Causation, prediction, and search. MIT press, 2001.
>
> [5]  Patrik Hoyer, Dominik Janzing, Joris M Mooij, Jonas Peters, and Bernhard Schölkopf. Nonlinear causal discovery with additive noise models. Advances in neural information processing systems, 21, 2008.
>
> [6] Alicia Curth and Mihaela Van Der Schaar. In search of insights, not magic bullets: Towards
> demystification of the model selection dilemma in heterogeneous treatment effect estimation. In
> International Conference on Machine Learning, pp. 6623–6642. PMLR, 2023.
>
> [7] Alicia Curth and Mihaela van der Schaar. Nonparametric estimation of heterogeneous treatment
> effects: From theory to learning algorithms. In Arindam Banerjee and Kenji Fukumizu (eds.),
> Proceedings of The 24th International Conference on Artificial Intelligence and Statistics, volume
> 130 of Proceedings of Machine Learning Research, pp. 1810–1818. PMLR, 13–15 Apr 2021. URL
> https://proceedings.mlr.press/v130/curth21a.html.
>
> [8] Sören R Künzel, Jasjeet S Sekhon, Peter J Bickel, and Bin Yu. Metalearners for estimating heteroge-
> neous treatment effects using machine learning. Proceedings of the national academy of sciences,
> 116(10):4156–4165, 2019.
>
> [9] Edward H Kennedy et al. Optimal doubly robust estimation of heterogeneous causal effects. arXiv
> preprint arXiv:2004.14497, 5, 2020.

---

> ### Comment · Reviewer_9kc4 · 2024-11-25
>
> Thank you for your very thorough comment and revision to the paper. I feel like the issues I pointed at have been largely addressed, and I don't find any major weakness left. I have changed my rating accordingly.

---

> ### Author Response · Authors · 2024-11-25
>
> We are extremely thankful for your helpful feedback and commitment to reviewing our work. We see that, in addressing your points, the manuscript has substantially improved, and we are very glad that you agree and have increased your score from a 5 to an 8!

---

### Official Review · Reviewer_ABMs · 2024-11-04

**Soundness:** 2
**Presentation:** 4
**Contribution:** 2
**Rating:** 5
**Confidence:** 4

**Summary:**

The paper considers the problem of generating synthetic data in the medical domain, where an organization may wish to release a privacy-preserving synthetic dataset generated from a model of a "real" dataset. The paper starts by defining the problem, considering what aspects of the causal problem should be preserved and how this should be measured, and then presents STEAM, a method for generating synthetic data.

**Strengths:**

The authors have identified what appears to be a really interesting question, and have been absolutely clear in communicating the setting of the paper. In many parts, the clarity of what is presented is good, and the authors have made a strong effort to highlight the relevant parts of the paper, and to help the reader navigate.

Many parts of the paper are very accessible to a broad ML audience: I appreciated that this complex topic was presented without making it unnecessarily "mathy".

I enjoyed the exposition in Section 7.2 of the paper, where the authors illustrated what I think is the main take-way form the work: if you generate synthetic data based on the joint, without considering causal structure, you may lose the ability to model the causal structure of interest.

The authors have provided code (which I have not checked beyond a cursory glance at the repo).

**Weaknesses:**

**Related work**
I'm not an expert in synthetic data generation, but I'm really surprised that no one has considered using the causal graph in synthetic data generation. Is this right? If so, why? If not, why not compare your STEAM method with them?

**Section 4** rather lacks a punchline: it would be great to see an example of when these desiderata would be breached, and what the consequences would be. The paper does move on to an illustrated example in section 5, but I felt rather disappointed at the end of section 4 that the authors did not land their argument with a conclusive demonstration of their point.

Section 5.2.1. I found the discussion of precision and recall as a metric for assessing a data distribution rathe confusing and non-intuitive, and equation (2) was completely unhelpful. I understand what precision and recall are, but I failed to grok from the work how they are used in the "widespread" application in evaluating a data density: I am not familiar with this work.

5.2.2 Acronym JSD not defined.

The main weakness of the paper for me was in section 6, where the thrust of the paper is presented:
"Mimicry of the real DGP acts as an inductive bias, pushing Q closer towards the P in structure, and directly targeting each distributions from our desiderata"

**Section 6**
The main thrust of the paper is summarised in section 6:

"Mimicry of the real DGP acts as an inductive bias, pushing Q closer towards the P in structure, and directly targeting each distributions from our desiderata"

I felt that the lede was rather buried here. This sentence would surely have fitted better a the end of section 1?

I was disappointed that the differential-privacy aspect of the problem was not treated with the same thoroughness as the causal/metric aspect. Yes, we can apply DP to any aspect of our generative model, but I feel the authors have much more they could add here. What is the interplay between DP and the ability of the dataset to represent a realistic problem? What if only the covariates need be DP?   What if there is no need for DP on the treatment assignment (is this realistic?)?

 Right now Section 6 adds very little value to the paper, beyond demonstrating the authors' understanding of causal statistics and DP.

**Section 7**
> X using T-learner PO estimators.

reference needed please.

**7.1** This section could do with a more detailed explanation of the setup - it probably seems obvious to the authors, but I'm not able to follow how this experiment was conducted.  I think what's happening here is that there is no DP happening (epsilon=0). Please clarify.

Section 7.3 is where the paper should really pull together to illustrate the utility of what's being proposed, and I'm afraid it falls short. I cannot figure out what dataset is being synthesised in Figure 5. Is it the toy illustration from section 7.2 above? If so, why not run on the real datasets from 7.1?

**Questions:**

Is it possible to make a realistically differentially private dataset where causal effects are still discoverable?

To answer this, I'd recommend setting up a DP attack where you attempt to de-anonymise a row of the data (i.e. estimate one of the covariates based on some others). Think carefully about the construction: perhaps there's a realistic case in one of the datasets where you can estimate the age and location of a participant, and whether they have a particular disease. Then, can you use STEAM to construct a dataset which would both be useful to scientists _and_ protect he identity of the individual?

---

> ### Author Response · Authors · 2024-11-22
> **Author response to ABMs (Part 1/5)**
>
> Thank you for your thoughtful comments and suggestions. We give answers to each of the following in turn, as well as pointing out corresponding updates to the revised manuscript:
>
> - (A) Comparison with causal generative models
> - (B) Discussion on precision and recall
> - (C) Insights on the interaction between differential privacy and treatment effect estimation
> - (D) Added differential privacy experiment on real data
> - (E) Minor updates
>
> ---
> **(A) Comparison with causal generative models**
>
> You are correct that existing works consider the generation of synthetic data from a known causal graph. Causal generative models (e.g. [1], [2]) typically try to approximate the underlying structural causal model (SCM) of a dataset, to answer observational, interventional, and counterfactual queries. We agree that adding a discussion and experimental comparison with causal generative models is an important addition to our paper, and there are important differences to clarify in the assumptions and motivation between such methods and ours. Most notably,  causal generative models typically **assume that the entire causal graph is known** to the data holder, which is a **more restrictive assumption than we make with STEAM**. We assume that our specification of the underlying DGP ($X \sim P_\mathbf{X},\ W \sim P_{W|\mathbf{X}},\ Y \sim P_{Y|W,\mathbf{X}}$) is correct for datasets containing treatments, but we do **not** require knowledge of the causal graph, as we do not need to know the causal links between individual variables. We do not assume knowledge of the causal relationships amongst the covariates, nor knowledge of which covariates cause treatment assignment or patient outcomes. As such, we make **less restrictive assumptions** than works which require knowledge of a causal graph, and we suggest that our approach is more realistic in complex, real-world scenarios, such as those that arise in medicine, where the true causal graph is unlikely to be available. Note that our setting follows established literature in causal inference that does not impose any assumption on the full underlying causal graph.
>
> Furthermore, the motivation for these works differs from ours. We seek to generate useful synthetic data only from the observational distribution, for use by analysts with goals such as **treatment effect estimation (e.g., CATE)**, while [1], [2], and other causal generative models primarily focus on generating data to answer **graph-specific interventional and counterfactual queries that require knowledge of the full causal graph**. Nevertheless, these methods can be used to generate data from the observational distribution, and we agree that experimental comparisons will strengthen the contribution of our paper.
>
> **Experiments with new baselines:** In order to compare to some baseline causal generative models (specifically the diffusion-based causal model (DCM) from [1] and the additive noise model (ANM) [3] implementation in the $\texttt{DoWhy-GCM}$ python [2]) we must first reconcile their differences in assumptions. For the real-world datasets we use in Section 7.1, we do not have access to the true causal graph, we simply know which features are the treatment and outcome. As such, we must first construct a causal graph to supply to these methods. We do so with three methods:
>
> 1. Construction of a naive graph $\mathcal{G}_{naive}$, in which each covariate causes $W$ and $Y$, $W$ causes $Y$, and every pair of covariates has a causal relationship between them;
> 2. Using the constraint-based PC causal discovery algorithm [4] to discover an estimated graph $\mathcal{G}_{discovered}$ from the Markov Equivalence Class for the true causal graph and;
> 3. Pruning the discovered causal graph by removing any edges which contradict the DGP we assume. As such, any edges from $Y$ to $W$ or $\mathbf{X}$, or from $W$ to $\mathbf{X}$ are removed to form $\mathcal{G}_{pruned}$.
>
> We compare the DCM and ANM methods using each of these causal graph discovery methods with the best performing STEAM models on each of the datasets used in Section 7.1. **We include the new results in the Appendix O**, and we add the table in the next comment for convenience.
>
> For each dataset, we see that the relevant **STEAM model outperforms all instantiations of the causal generative models in almost every metric**, only being outperformed to a statistically significant level in $R_{\beta, X}$ on the ACIC dataset. These results validate that, when the true causal graph is not known, our less restrictive assumptions enable more useful generation of synthetic data containing treatments. We also see that the differences between the graph discovery methods are relatively small.

---

> > ### Comment · Reviewer_ABMs · 2024-11-26
> >
> > Thanks for the edits to the paper, especially Figure 5.
> >
> > *(A) Comparison with causal generative models*
> >
> > thanks for improving this. I still think the positioning of the paper is unclear in the current draft. With the discussion in here I can now tease out your contribution, but I'm left unconvinced that a fresh reader will easily grok the importance of your contirubution, of how it's distinguished form other related works on synthetic generation.
> >
> > *(B) Discussion on precision and recall*
> >
> > Much better, thanks.
> >
> > *(C) Insights on the interaction between differential privacy and treatment effect estimation*
> >
> > Good, thanks.
> >
> > *(D) Added differential privacy experiment on real data*
> >
> > Thanks for the hard work on this.
> >
> > Can you help me intuit some of the results in Fig 5 a little? I see that there's a big effect on U_pehe, which is a direct result of your data generation. But why is there an effect on P_ax or R_bx? I the case where X covers the majority of dimensions, we'd expect STEAM and "joint" generation methods to match, right? Why is the behaviour as it is?
> >
> > *(E) Minor updates*
> >
> > Great, glad to have been able to help.

---

> > > ### Author Response · Authors · 2024-11-26
> > >
> > > Thank you for your continued engagement with our work. We appreciate the effort you have made during this discussion period, and we are glad that our changes are well received. We hope to address your remaining points here.
> > >
> > > ---
> > >
> > > **(A) Positioning of the paper**
> > >
> > > We are very thankful for your efforts in helping us convey the correct message to new readers, and we agree that we can better contextualise our work. We see that the comparison with causal generative models is an important point to raise early, and as such we have now extended our Table 2 in the related works (Section 3) to encompass this approach. Furthermore, we bring forward some of the relevant points that have arisen from this discussion (which we initially placed in Appendix B) into the main body of the paper in Sections 1 and 3. We believe that these changes will enable readers to adequately place our work amongst the related literature, and clearly see how we differentiate from existing works. Please see the new revised manuscript for these further updates.
> > >
> > > **_Update_:** We have **extended Table 2** to encompass causal generative models, and **moved some comments from Appendix B into Sections 1 and 3** to clearly communicate our points of difference from existing works.
> > >
> > > ---
> > >
> > > **(D) DP results**
> > >
> > > We can help with the interpretation of the results from Section 7.3. Indeed, the improvement we see in the $U_{PEHE}$ metric by STEAM compared to 'joint' DP models is a direct result of our targeted modelling of this distribution, allowing it to be better preserved.
> > >
> > > For $P_{\alpha,\mathbf{X}}$ and $R_{\beta,\mathbf{X}}$, your intuition, that STEAM and 'joint' models would be expected to score similarly (especially when the dimensionality of $\mathbf{X}$ is high), is **correct in the non-DP setting**. We see this in the results from Section 7.1, where, across each dataset, **the STEAM and 'joint' models score very similarly on $P_{\alpha,\mathbf{X}}$ and $R_{\beta,\mathbf{X}}$**. This is because the $Q_\mathbf{X}$ component model in STEAM has the same **form and capacity** as its 'joint' counterpart, and they approach the modelling of $P_\mathbf{X}$ **in almost the same way**, save for the one-dimensional $W$ and $Y$ variables which $Q_\mathbf{X}$ ignores.
> > >
> > > This result **does not hold in the DP setting**, however, because $Q_\mathbf{X}$ **no longer has the same effective capacity as its 'joint' counterpart**, because of the difference in how $\epsilon$ is distributed by STEAM vs. 'joint' models. With our uniform distribution of the privacy budget amongst $Q_\mathbf{X}$, $Q_{W|\mathbf{X}}$, and $Q_{Y|W,\mathbf{X}}$ in STEAM, $Q_\mathbf{X}$ receives one third of the total budget, as we seek to ensure that the conditional models $Q_{W|\mathbf{X}}$, and $Q_{Y|W,\mathbf{X}}$ receive adequate budget to perform well, because of their importance to downstream analysts.
> > >
> > > The 'joint' model, however, is supplied with the entire privacy budget, and internally distributes it across variables according to its particular DP algorithm. Because of the dimensionality of $\mathbf{X}$, its reasonable to assume that most algorithms will distribute more of the privacy budget to modelling $\mathbf{X}$ than $W$ and $Y$. Therefore, we expect 'joint' models will tend to preserve $P_\mathbf{X}$ better than STEAM models in this setting, because they supply more of the privacy budget to this distribution, and therefore have more effective capacity in modelling it well. Indeed, this is the phenomenon we observe in Section 7.3, where STEAM models $P_\mathbf{X}$ worse because of the conservative budget supplied to $Q_\mathbf{X}$. By enforcing that the conditional models $Q_{W|\mathbf{X}}$, and $Q_{Y|W,\mathbf{X}}$ in STEAM are respected and receive adequate budget due to their importance, we naturally sacrifice the ability to model $P_\mathbf{X}$ as well as DP 'joint' methods at the same $\epsilon$ level.
> > >
> > > ---
> > >
> > > Thank you once again for your continued interaction. We hope that we have addressed your concerns, and, if so, that you will consider moving your recommendation to acceptance.

---

> > > ### Author Response · Authors · 2024-11-30
> > >
> > > Dear Reviewer ABMs,
> > >
> > > Thank you once again for your thoughtful feedback, which has significantly improved our paper. We are glad that our initial response addressed many of your concerns, and we have since carefully answered your follow-up comments.
> > >
> > > As the discussion period is nearing its end, we wanted to check if you have any remaining concerns or questions we can address. If our revisions meet your expectations, we hope you might consider increasing your score to recommend acceptance.
> > >
> > > We truly appreciate your time and invaluable insights!

---

> > > ### Author Response · Authors · 2024-12-02
> > >
> > > Dear Reviewer ABMs,
> > >
> > > We apologise for the repeated messages, however with only one day left in the discussion period, we are hoping to get your feedback on our response to your additional queries. We are glad that we have received positive feedback during the discussion period, with two reviewers now at an 8, and we hope that, if we have satisfactorily addressed your comments, you too might move your recommendation towards acceptance.
> > >
> > > Many thanks for your time and effort!

---

> ### Author Response · Authors · 2024-11-22
> **Author response to ABMs (Part 2/5)**
>
> Table 16: $P_{\alpha, \mathbf{X}}$, $R_{\beta, \mathbf{X}}$, $JSD_\pi$, and $U_\text{PEHE}$ values for the best performing STEAM models from Appendix I in comparison to causal generative models. Averaged over 20 runs, with 95\% confidence intervals. Bold indicates the statistically significant best performing model on that metric and dataset.
> | Dataset  | Model                   | $\boldsymbol{P_{\alpha, \mathbf{X}}}$ ($\uparrow$) | $\boldsymbol{R_{\beta, \mathbf{X}}}$ ($\uparrow$) | $\boldsymbol{\textbf{JSD}_\pi}$ ($\uparrow$) | $\boldsymbol{U_\textbf{PEHE}}$ ($\downarrow$) |
> |-------------------------------------------|-------------------------------------|----------------------------------------------------|---------------------------------------------------|----------------------------------------------|-----------------------------------------------|
> | ACTG                  | STEAM$_\text{TVAE}$  |**0.929 $\pm$ 0.008**| **0.486 $\pm$ 0.009**| 0.958 $\pm$ 0.004   | **0.492 $\pm$ 0.011**   |
> |  | DCM $\mathcal{G}_\text{naive}$      | 0.773 $\pm$ 0.013           | 0.369 $\pm$ 0.006     | 0.937 $\pm$ 0.006   | 0.665 $\pm$ 0.034   |
> | | DCM $\mathcal{G}_\text{discovered}$ | 0.756 $\pm$ 0.011     | 0.350 $\pm$ 0.007       | 0.956 $\pm$ 0.005                            | 0.605 $\pm$ 0.023 |
> |   | DCM $\mathcal{G}_\text{pruned}$     | 0.758 $\pm$ 0.013   | 0.358 $\pm$ 0.007  | 0.957 $\pm$ 0.003                            | 0.596 $\pm$ 0.017                             |
> |   | ANM $\mathcal{G}_\text{naive}$      | 0.787 $\pm$ 0.007                                  | 0.389 $\pm$ 0.008                                 | 0.954 $\pm$ 0.005                            | 0.580 $\pm$ 0.017                             |
> |                                           | ANM $\mathcal{G}_\text{discovered}$ | 0.836 $\pm$ 0.007                                  | 0.419 $\pm$ 0.007                                 | 0.952 $\pm$ 0.004                            | 0.578 $\pm$ 0.019                             |
> |                                           | ANM $\mathcal{G}_\text{pruned}$     | 0.839 $\pm$ 0.008                                  | 0.412 $\pm$ 0.005                                 | 0.952 $\pm$ 0.005                            | 0.582 $\pm$ 0.014                             |
> | |  | | | |  |
> | IHDP                  | STEAM$_\text{CTGAN}$              | 0.674 $\pm$ 0.014  | **0.424 $\pm$ 0.011**| **0.928 $\pm$ 0.009**| **1.709 $\pm$ 0.052** |
> |        | DCM $\mathcal{G}_\text{naive}$      | 0.557 $\pm$ 0.010  | 0.340 $\pm$ 0.009    | 0.883 $\pm$ 0.016                            | 4.878 $\pm$ 0.395                             |
> |                                           | DCM $\mathcal{G}_\text{discovered}$ | 0.658 $\pm$ 0.011| 0.360 $\pm$ 0.007                                 | 0.893 $\pm$ 0.008                            | 2.059 $\pm$ 0.140                             |
> |                                           | DCM ${\mathcal{G}_\text{pruned}}^*$ | 0.658 $\pm$ 0.011                                  | 0.360 $\pm$ 0.007                                 | 0.893 $\pm$ 0.008                            | 2.059 $\pm$ 0.140                             |
> |                                           | ANM $\mathcal{G}_\text{naive}$      | 0.597 $\pm$ 0.029                                  | 0.379 $\pm$ 0.011                                 | 0.900 $\pm$ 0.005                            | 1.868 $\pm$ 0.147                             |
> |                                           | ANM $\mathcal{G}_\text{discovered}$ | 0.589 $\pm$ 0.012                                  | 0.359 $\pm$ 0.009                                 | 0.892 $\pm$ 0.008                            | 1.865 $\pm$ 0.059                             |
> |                                           | ANM ${\mathcal{G}_\text{pruned}}^*$ | 0.589 $\pm$ 0.012 | 0.359 $\pm$ 0.009                                 | 0.892 $\pm$ 0.008                            | 1.865 $\pm$ 0.059   |
> | |  | | | |  |
> | $\text{ACIC}^\dagger$ | STEAM$_\text{ARF}$  | 0.939 $\pm$ 0.004| 0.393 $\pm$ 0.004  | **0.977 $\pm$ 0.002** | **2.176 $\pm$ 0.141**   |
> | | DCM $\mathcal{G}_\text{discovered}$ | 0.942 $\pm$ 0.004   | 0.422 $\pm$ 0.003  | 0.957 $\pm$ 0.003                            | 4.249 $\pm$ 0.132 |
> |                                           | DCM $\mathcal{G}_\text{pruned}$     | 0.939 $\pm$ 0.004 | 0.420 $\pm$ 0.004| 0.959 $\pm$ 0.002                            | 4.340 $\pm$ 0.159                             |
> |   | ANM $\mathcal{G}_\text{discovered}$ | 0.929 $\pm$ 0.003 | 0.404 $\pm$ 0.003   | 0.872 $\pm$ 0.002                            | 4.193 $\pm$ 0.127   |
> |          | ANM $\mathcal{G}_\text{pruned}$     | 0.930 $\pm$ 0.004 | 0.404 $\pm$ 0.003| 0.880 $\pm$ 0.002                            | 4.481 $\pm$ 0.174|
>
> \* $\mathcal{G}$ pruned is the same as $\mathcal{G}$ discovered for IHDP
>
> $\dagger$ Excessive runtime caused the exclusion of $\mathcal{G}_\text{naive}$ ACIC results

---

> ### Author Response · Authors · 2024-11-22
> **Author response to ABMs (Part 3/5)**
>
> Furthermore, STEAM’s design can, in principle, incorporate **any** generative model for $Q_\mathbf{X}$, essentially acting as a wrapper around $Q_\mathbf{X}$ to improve its generation quality for causal inference tasks. This allows STEAM to very easily empower existing generative modelling frameworks, without having to incorporate bespoke generators. Also, it allows STEAM models to continuously improve along with the base generative model. Existing causal generative models do not allow such flexibility, and therefore cannot as easily benefit from future improvements.
>
> Finally, the causal generative models we compared against here cannot immediately satisfy differential privacy, and any causal generative model which requires knowledge of the causal graph must ensure that the causal discovery methods used are also privacy-enabled.
>
> **_Update:_** We have added a **discussion on causal generative models** to our related works section. We have added **new experimental results using the causal generative models DCM and ANM as baselines** to Appendix O.
>
> ---
>
> **(B) Discussion on precision and recall**
>
> The alpha-precision and beta-recall metrics we use in Section 5.2.1 were introduced in [5] as metrics for evaluating the quality of synthetic data. They are extensions of the conventional precision/recall metrics in the supervised learning setting to the synthetic data generation setting. Intuitively, alpha-precision captures how much of the synthetic data falls within the support of real data, whereas beta-recall reflects how much of the real data is covered by the support of synthetic data. Together, they capture the faithfulness and diversity of the synthetic data. Since their introduction in 2022, they have been used in variety of publications (197 citations on Google Scholar), hence our description of their use as 'widespread'.
>
> **_Update:_** We have **updated our descriptions of alpha-precision and beta-recall in Section 5**, and **changed equation (3) and (4) to incorporate the full definitions from [5].**
>
> ---
>
> **(C) Insights on the interaction between differential privacy and treatment effect estimation**
>
> Our consideration of differential privacy (DP) stems from our desire to **facilitate data release**. In this work, we consider the application of releasing synthetic data for causal inference tasks, such as treatment effect estimation, which means that **all of the covariates, the treatment assignment and the response must be released**, and are **subject to potential privacy attacks**. Only ensuring DP for a subset of variables (such as only the covariates) violates the DP of the entire dataset, as **DP for a subset of variables does not lead to DP for all variables**. Therefore, **pursuing this would remove any guarantees on privacy for individuals** whose data are being released. This could be ok on a case-by-case basis, but we hope for our method to be as broadly applicable as possible, and so seek to adhere to DP across the entire dataset.
>
> Furthermore, to your question on whether a data holder can release _‘a realistically differentially private dataset where causal effects are still discoverable’_, DP is a formal and provable way to establish data privacy - however **it is not directly associated with one particular style of privacy attack** (such as membership inference or attribute inference). In fact, a criticism of DP is that its practical implication is very often unclear [6]. For instance, DP provides little insight into how many individuals in the data are vulnerable to particular re-identification attacks. While **increasing the degree of DP will generally reduce the success of all privacy attacks**, it does not establish provable guarantees for any particular methodology. This has triggered new lines of research into privacy enhancement for specific attack modality, such as in [6], which is fundamentally different from DP privacy. Since we use DP in particular in the work, we do not focus on any particular attack method.
>
> Nevertheless, if we assume that a data holder wants to release a **DP synthetic dataset for CATE estimation**, but they have some **threshold on the degree of added error** that analysts will encounter using this dataset for this task, we can show that STEAM facilitates the release of more private datasets than standard DP generation. We can see this is true simply by examining the DP generation results in Section 7.3, and the full results on the extended set of baselines in Appendix I. Comparing each STEAM model with its standard analogue, for any threshold a data holder may set on an acceptable $U_{PEHE}$ score (which measures the difference in the CATEs found from the synthetic vs. real data) we see that the STEAM model always allows a synthetic dataset with smaller $\epsilon$ (and therefore greater degree of privacy) before hitting this set threshold.

---

> ### Author Response · Authors · 2024-11-22
> **Author response to ABMs (Part 4/5)**
>
> **(D) Added differential privacy experiment on real data**
>
> We have **run a further set of DP generation experiments**, comparing STEAM with four extra baselines - AIM [7], GEM [8], MST [9], and RAP [10] - **on the ACTG dataset used in Section 7.1**, for a more **robust DP comparison on more realistic data**. We now base our updated Section 7.3 using this experiment. Of these four baselines, we found that AIM produced the best data across different epsilon, and its STEAM analogue was the best performing STEAM model. As such, we include these results for AIM vs. $STEAM_{AIM}$ in the main body of the updated manuscript, and include the full results with the other baseline models in the Appendix I
>
> We will now discuss our updated takeaways from this DP generation experiment. Figure 5, in Section 7.3 shows the results. **$STEAM_{AIM}$ models $P_{Y|W,\mathbf{X}}$ better on all tested values of $\epsilon$**, as $U_{PEHE}$ is significantly lower than for standard AIM. **$P_{W|\mathbf{X}}$ is better modelled by $STEAM_{AIM}$ at small $\epsilon$**, with equivalent performance between the methods at less conservative budgets. **$P_{\mathbf{X}}$, on the other hand, is better preserved by standard AIM**, scoring higher on $P_{\alpha, \mathbf{X}}$ and $R_{\beta, \mathbf{X}}$ at most $\epsilon$. This is likely because assigning $Q_\mathbf{X}$ one third of the budget of the standard AIM model and having it model largely the same distribution, save for the removed $W$ and $Y$, is prohibitively restrictive given the high-dimensionality of $\mathbf{X}$. As such, with uniform distribution of ($\epsilon, \delta$) across each component, we see a trade-off between $STEAM_{AIM}$ and standard AIM, where **$STEAM_{AIM}$ better preserves $P_{W|\mathbf{X}}$ and $P_{Y|W,\mathbf{X}}$, while standard AIM preserves $P_{\mathbf{X}}$ better**. Distributing ($\epsilon, \delta$) differently amongst  $Q_\mathbf{X}$, $Q_{W|\mathbf{X}}$, and $Q_{Y|W,\mathbf{X}}$ could address this trade-off, as we discuss in Appendix M.
>
> **_Update:_** We have **changed Section 7.3 so that we compare DP generation on the real world ACTG dataset**, between AIM and $STEAM_{AIM}$. We have **added more results for other DP baselines tested (GEM, MST, RAP)** in Appendix I.
>
> ---
> **(E) Minor updates**
>
> Thank you for helping us improve the writing and flow of our paper. We summarise some such changes we have made that stem from your review here:
> - Described our motivation for STEAM earlier, moving a sentiment similar to the sentence ‘Mimicry of the real DGP acts as an inductive bias, pushing Q closer towards the P in structure, and directly targeting each distributions from our desiderata’ in Section 6, into our contributions in Section 1
> - Added citation for T-learner in Section 7
> - Defined acronym JSD in Section 5
> - Clearly explained that Section 7.1 is indeed focussing on the non-DP setting
> - Made Section 4 more conclusive by adding pointers to later parts in the paper, where we show the poor performance of both existing metrics and generative models in respecting our desiderata.
>
> ---
>
> Thank you once again. We hope that we have addressed all your comments, and we greatly appreciate your feedback.

---

> ### Author Response · Authors · 2024-11-22
> **Author response to ABMs (Part 5/5)**
>
> **References**
>
> [1] Patrick Chao, Patrick Blöbaum, Sapan Patel, and Shiva Prasad Kasiviswanathan. Modeling causal
> mechanisms with diffusion models for interventional and counterfactual queries, 2024. URL
> https://arxiv.org/abs/2302.00860.
>
> [2] Patrick Blöbaum, Peter Götz, Kailash Budhathoki, Atalanti A Mastakouri, and Dominik Janzing.
> Dowhy-gcm: An extension of dowhy for causal inference in graphical causal models. Journal of
> Machine Learning Research, 25(147):1–7, 2024.
>
> [3] Patrik Hoyer, Dominik Janzing, Joris M Mooij, Jonas Peters, and Bernhard Schölkopf. Nonlinear
> causal discovery with additive noise models. Advances in neural information processing systems,
> 21, 2008.
>
> [4] Peter Spirtes, Clark Glymour, and Richard Scheines. Causation, prediction, and search. MIT press,
> 2001.
>
> [5] Ahmed Alaa, Boris Van Breugel, Evgeny S. Saveliev, and Mihaela van der Schaar. How faithful
> is your synthetic data? Sample-level metrics for evaluating and auditing generative models. In
> Kamalika Chaudhuri, Stefanie Jegelka, Le Song, Csaba Szepesvari, Gang Niu, and Sivan Sabato
> (eds.), Proceedings of the 39th International Conference on Machine Learning, volume 162
> of Proceedings of Machine Learning Research, pp. 290–306. PMLR, 17–23 Jul 2022. URL
> https://proceedings.mlr.press/v162/alaa22a.html.
>
> [6] Van Breugel, Boris, et al. "Membership inference attacks against synthetic data through overfitting detection." arXiv preprint arXiv:2302.12580 (2023).
>
> [7] Ryan McKenna, Brett Mullins, Daniel Sheldon, and Gerome Miklau. Aim: An adaptive and iterative mechanism for differentially private synthetic data, 2024. URL https://arxiv.org/abs/2201.12677.
>
> [8] Terrance Liu, Giuseppe Vietri, and Steven Z Wu. Iterative methods for private synthetic data: Unifying framework and new methods. Advances in Neural Information Processing Systems, 34: 690–702, 2021.
>
> [9] Ryan McKenna, Gerome Miklau, and Daniel Sheldon. Winning the nist contest: A scalable and general approach to differentially private synthetic data. arXiv preprint arXiv:2108.04978, 2021.
>
> [10] Sergul Aydore, William Brown, Michael Kearns, Krishnaram Kenthapadi, Luca Melis, Aaron Roth, and Ankit A Siva. Differentially private query release through adaptive projection. In International Conference on Machine Learning, pp. 457–467. PMLR, 2021.

---

### Official Review · Reviewer_UJZ3 · 2024-11-05

**Soundness:** 3
**Presentation:** 3
**Contribution:** 3
**Rating:** 8
**Confidence:** 4

**Summary:**

This paper addresses the problem of generating synthetic data that includes treatment and outcome variables. The goal is to enable the release of this synthetic data, potentially with differential privacy, to support various downstream tasks. In particular, causal inference tasks such as treatment effect analysis. The paper formulates this problem into three questions, then claims that existing methods based on fidelity and utility of synthetic data are not adequate to answer these questions. Additionally, when the covariate distribution has a relatively high dimensionality, these methods fail to accurately estimate the treatment assignment and outcome generation mechanisms. To support this claim, the paper includes an experiment with a DGP to show existing metrics do not change significantly by changing the outcome generation mechanism. In contrast, this change is clearly detected by a CATE-based metric. If the joint distribution of the covariates, treatments, and outcomes is factorized using the DGP, then the required inductive bias can be established which existing methods do not capitalize on. Answers for the proposed questions are provided, based on the DGP, by first establishing a set of desiderata that the synthetic data should satisfy, and then deriving a set of metrics that relate to the performance of downstream learners. The paper introduces STEAM, a data generation method which mimics the real DGP, then demonstrates its performance by comparing it to the other data generation methods using the proposed metrics.

**Strengths:**

* The paper addresses an important problem.
* The problem statement and contributions are clear.
* The approach is novel, intuitive, and general.
* The claims are supported by experiments.
* STEAM outperforms existing methods on the proposed metrics in both the non-DP setting and the DP setting when $\epsilon > 1$.

**Weaknesses:**

* The uniform distribution of the privacy budget for STEAM based on Theorem 1 is not optimal. The paper already acknowledges this in the discussion.
* In Section 7.3, the $\delta$ in the experiments is set to $10^{-3}$ which might not be ideal and DP-GAN performs better than STEAM around $\epsilon = 1$ on the first three metrics. There are related questions below.

**Questions:**

* In page 5, in the first paragraph at the beginning: "Particularly as X grows in size", should "size" be replaced with "dimensionality"?
* Related to section 7.3, typical values for $\delta$ are $10^{-5}$, $\frac{1}{n}$, and $\frac{1}{n^2}$. Was the specific choice of $\delta$ in the experiments based on $n = 1000$ by setting $\delta = \frac{1}{n}$? It would be safer to choose $10^{-5}$ or $10^{-6}$.
* Also related to Section 7.3, STEAM performs worse than DP-GAN around $\epsilon = 1$ on the first three metrics. Is that caused by the choice of some hyperparameters (e.g. DP-SGD hyperparameters)? I would also suggest to see whether this holds when $\epsilon < 1$.

---

> ### Author Response · Authors · 2024-11-22
> **Author response to UJZ3 (Part 1/2)**
>
> Thank you for your thoughtful comments and suggestions. We give answers to each of the following in turn, as well as pointing out corresponding updates to the revised manuscript:
>
> * (A) Performance at smaller $\delta$
> * (B) Distribution of $\epsilon$ in STEAM
> * (C) Performance for $\epsilon < 1$
> * (D) Minor changes
> ---
> **(A) Performance at smaller $\delta$**
>
> You are correct that, for the differential privacy (DP) experiment in Section 7.3, we set $\delta = 1/n$, with $n = 1000$. In settings where more privacy is necessary, then $\delta$ may need to be reduced. We have run **new DP experiments with $\mathbf{\delta = 10^{-6}}$** and we include these results in Section 7.3. Please note as well that, in this section, we now use the real world ACTG dataset from Section 7.1, to inspect DP performance in a more realistic setting, and we compare against an updated set of more competitive baseline DP generative models. We discuss these results further in addressing point (C).
>
> **_Update:_** We have included **new results** in Section 7.3 which **compare STEAM with standard DP generative models when $\mathbf{\delta = 10^{-6}}$**.
>
> ---
> **(B) Distribution of $\mathbf{\epsilon}$ in STEAM**
>
> In our original manuscript (Section 7.3) we explore the performance of STEAM in DP generation with uniform distribution of $\epsilon$ across $Q_\mathbf{X}$, $Q_{W|\mathbf{X}}$, and $Q_{Y|W,\mathbf{X}}$. While this is one potential mechanism to ensure DP, a more general approach is to distribute $\epsilon$ according to some preference function $f:(0,\infty) \times \triangle^2  \rightarrow \epsilon \cdot \triangle^2$ (where $\triangle^2$ is the 2-simplex) which takes input of the budget $\epsilon$ and weights $\mathbf{w}$ for the relative importance of good modelling in $Q_\mathbf{X}$, $Q_{W|\mathbf{X}}$, and $Q_{Y|W,\mathbf{X}}$, and outputs a corresponding $\epsilon$ distribution. For example, a simple preference function definition would be $f(\epsilon, \mathbf{w}) = \epsilon \cdot \mathbf{w}$ where $\mathbf{w}$ could be defined by a data holder with some prior knowledge of the importance level of each component distribution to downstream analysts. Another approach, if it is not necessary to specify the desired $\epsilon$ distribution _a priori_, is to treat it as a hyperparameter, to be tuned over a series of runs to optimize some metric, such as a combination of $U_{PEHE}$, $P_{\alpha, X}$, $R_{\beta, X}$, and $\text{JSD}_{\pi}$.
>
> **_Update:_** We have **extended our discussion on the above alternatives to uniform distribution of the privacy budget** amongst the STEAM component models in Appendix M.
>
> ---
> **(C) Performance for $\epsilon < 1$**
>
> In Section 7.3, in the new experiment on the ACTG dataset, we **added new results for $\mathbf{\epsilon \in [0.25, 0.5, 1, 2, 3, 5, 10, 15]}$ against an updated set of DP baselines**. We display results for the AIM [1] DP generative model and $\text{STEAM}_{AIM}$ in the main body of the paper, with extended results using GEM [2], MST [3], and RAP [4] DP generative models and their STEAM analogues in Appendix I.
>
> We generally see that $STEAM_{AIM}$ performs comparatively better in preserving $P_{W|\mathbf{X}}$, and $P_{Y|W,\mathbf{X}}$, and comparatively worse for $P_{\mathbf{X}}$ at the same privacy budget as AIM. We do not suggest that the comparatively worse modelling of $P_{\mathbf{X}}$ is because of any particular hyperparameter settings in the DP methods used within STEAM. Rather, it is a natural consequence of the inverse relationship between $\epsilon$ and the noise required to satisfy DP. Since we allocate $\epsilon$ uniformly across $Q_{\mathbf{X}}$, $Q_{W|\mathbf{X}}$ and $Q_{Y|W,\mathbf{X}}$ in this experiment, each STEAM component model only receives one third of the budget compared to the baseline AIM model, and this seems to become prohibitively restrictive for modelling $P_{\mathbf{X}}$. Intuitively, this makes sense, since $P_{\mathbf{X}}$ is likely the hardest distribution to model given its high dimensionality, and STEAM’s $Q_{\mathbf{X}}$ and the standard AIM method are modelling it almost the same way, save for the removed $W$ and $Y$ variables from STEAM, yet $Q_{\mathbf{X}}$ receives one third of the budget. To mitigate this effect, since STEAM outperforms in the  $U_{PEHE}$ and $JSD_{\pi}$ metric, and the performance of STEAM’s $Q_{Y|W,\mathbf{X}}$ and $Q_{W|\mathbf{X}}$ do not appear to be largely affected by reductions in $\epsilon$, one could redistribute some of the budget assigned to $Q_{Y|W,\mathbf{X}}$ and $Q_{W|\mathbf{X}}$ to $Q_{\mathbf{X}}$, for a more even performance across all three distributions.
>
> **_Update:_** We have **included new results in Section 7.3** which showcase DP generation for $\epsilon < 1$, and we have **updated the takeaways** to include discussion on why $STEAM_{AIM}$ preserves $Q_{\mathbf{X}}$ worse than AIM.

---

> > ### Comment · Reviewer_UJZ3 · 2024-11-25
> >
> > Thank you for addressing the issues. I believe that your explanation of why STEAM is working worse for modelling $P_X$ is correct and that it is related to the uniform privacy budget which you have already acknowledged as a limitation of the current work. I have no further comments.

---

> > > ### Author Response · Authors · 2024-11-30
> > >
> > > Dear Reviewer UJZ3,
> > >
> > > We are deeply grateful for your thoughtful consideration of our responses, and for your recognition of the contribution of our work. Your comments have been invaluable in refining our paper.
> > >
> > > Thank you once again for your feedback and support.

---

> ### Author Response · Authors · 2024-11-22
> **Author response to UJZ3 (Part 2/2)**
>
> ---
> **(D) Minor changes**
>
> Thank you for pointing out the mistake on line 225, your interpretation is indeed correct. We have now updated this to say ‘as $\mathbf{X}$ grows in dimensionality’.
>
> ---
>
> Thank you once again. We hope that we have addressed all your comments, and we greatly appreciate your feedback.
>
> ---
> **References**
>
> [1] Ryan McKenna, Brett Mullins, Daniel Sheldon, and Gerome Miklau. Aim: An adaptive and iterative
> mechanism for differentially private synthetic data, 2024. URL https://arxiv.org/abs/2201.12677.
>
> [2] Terrance Liu, Giuseppe Vietri, and Steven Z Wu. Iterative methods for private synthetic data:
> Unifying framework and new methods. Advances in Neural Information Processing Systems, 34:
> 690–702, 2021.
>
> [3] Ryan McKenna, Gerome Miklau, and Daniel Sheldon. Winning the nist contest: A scalable and
> general approach to differentially private synthetic data. arXiv preprint arXiv:2108.04978, 2021.
>
> [4] Sergul Aydore, William Brown, Michael Kearns, Krishnaram Kenthapadi, Luca Melis, Aaron Roth,
> and Ankit A Siva. Differentially private query release through adaptive projection. In International
> Conference on Machine Learning, pp. 457–467. PMLR, 2021.

---

### Author Response · Authors · 2024-11-22
**Global response**

We are very grateful for the time the reviewers have dedicated to helping us improve our work. We are glad that reviewers appreciate the importance of our work, along with the novelty and clarity of our method. We have taken into account the thoughtful feedback we have received, and we have made the following key changes to improve the paper:

1. **Extended empirical investigation on differential privacy.** We have added more experiments to further assess differentially private generation against an extended set of baseline models, on real-world data, and across a broader range of privacy budgets, in Section 7.3 and Appendix I.
2. **Comparison with causal generative models.** We have added a discussion on the related field of causal generative models to our related works in Section 3. We also extend on this, clarifying differences in motivation, assumptions, and flexibility between our work and existing methods in Appendix B. We have also added empirical comparisons of STEAM with baseline causal generative models in Appendix O.
3. **Formal motivation for establishing metrics.** We have derived formal arguments to show the inadequacy of joint-distribution-level KL divergence in evaluating synthetic data containing treatments in Section 5.1, adding to the motivation from our illustrative experiments now in Appendix D to propose new metrics.
4. **Improved accessibility of metrics.** We have made our metrics more easily accessible by improving their definitions, and giving practical guidelines for their calculation in Section 5 and Appendix F.

We have uploaded the revised manuscript with changes coloured in teal, and we are very pleased with its improvement.

---

With thanks,

The Authors of #3845

---

### Meta-Review · Area_Chair_6bdF · 2024-12-20

**Metareview:**

The paper highlights how synthetic data generated naively using a global generative model may be suboptimal for causal inference tasks, and presents metrics for evaluating generation as well as simple case studies highlighting the advantages of the proposed approach.

**Strengths:**
* Reviewers all agree the paper is well presented
* Reviewers all agree the paper raises an important issue on causal inference using synthetic data

**Weaknesses:**
* The reviewers considered the technical contribution beyond raising the key point to be limited
* The reviewers agree that there is limited justification for the specific technical choices, such as the evaluation metrics
* The reviewers agree that DP example is cartoonish, does not represent state-of-the-art and does not provide significant value

This was a borderline paper that was discussed at an online meeting between the AC and all reviewers. All reviewers agreed about the merits and weaknesses of the paper listed above. None of the reviewers felt strongly that the paper should be accepted. In the end, the weaknesses were considered to outweigh the strengths and the contribution was considered insufficient for ICLR.

**Additional Comments On Reviewer Discussion:**

There was extensive discussion on the paper that ultimately came down to evaluating the weight of the contribution.
The paper was discussed at an online meeting between the AC and all the reviewers where the consensus opinion was formed.

---

### Decision · Program_Chairs · 2025-01-22

Reject